# Similarity in viral and host promoters couples viral reactivation with host cell migration

Kathrin Bohn-Wippert[1], Erin N. Tevonian[1], Melina R. Megaridis[1] & Roy D. Dar[1,2,3]

Viral–host interactomes map the complex architecture of an evolved arms race during host cell invasion. mRNA and protein interactomes reveal elaborate targeting schemes, yet evidence is lacking for genetic coupling that results in the co-regulation of promoters. Here we compare viral and human promoter sequences and expression to test whether genetic coupling exists and investigate its phenotypic consequences. We show that viral–host co-evolution is imprinted within promoter gene sequences before transcript or protein interactions. Co-regulation of human immunodeficiency virus (HIV) and human C-X-C chemokine receptor-4 (CXCR4) facilitates migration of infected cells. Upon infection, HIV can actively replicate or remain dormant. Migrating infected cells reactivate from dormancy more than non-migrating cells and exhibit differential migration–reactivation responses to drugs. Cells producing virus pose a risk for reinitiating infection within niches inaccessible to drugs, and tuning viral control of migration and reactivation improves strategies to eliminate latent HIV. Viral–host genetic coupling establishes a mechanism for synchronizing transcription and guiding potential therapies.

[1] Department of Bioengineering, University of Illinois at Urbana-Champaign, 1270 Digital Computer Lab, MC-278, 1304W Springfield Avenue, Urbana, Illinois 61801, USA. [2] Carl R. Woese Institute for Genomic Biology, University of Illinois at Urbana-Champaign, 1206W Gregory Drive, Urbana, Illinois 61801, USA. [3] Center for Biophysics and Quantitative Biology, University of Illinois at Urbana-Champaign, 1110 West Green Street, Urbana, Illinois 61801, USA. Correspondence and requests for materials should be addressed to R.D.D. (email: roydar@illinois.edu).

Within biological organisms, interactions and functions can be topologically represented as a network of interaction modules[1]. Investigations have integrated viral–host messenger RNA[2] and protein[3–5] interaction networks essential for our understanding of human disease[6]. These networks can lead to therapeutic outcomes as demonstrated by a study in which viral–host interactome screening identifies and targets host factors involved in influenza virus replication with antiviral drugs[7]. Typical interaction networks rely on direct mRNA and protein interactions, but long-range indirect coupling of viral and host pathways may exist at the DNA level.

Human host gene promoters hold clues for identifying pathways harnessed by viruses and viral–host relationships stem from millions of years of evolution in different viruses and hosts[8]. These interactions have potential insight into regulatory mechanisms and viral targeting strategies. To date, the study of endogenous retroviral elements has focused predominantly on their structure, evolution, distribution and regulation of, or interaction with, the host[9–12]. Reports have shown methodologies for quantifying the real-time activity of transposable elements in single bacterial cells[13], identified the rewiring of transcriptional circuitry in pluripotent stem cells by transposable elements[14] and revealed widespread similarity of enhancers of innate immune response genes in a number of viruses[15]. These studies highlight that investigating endogenous retroviral elements may provide a deeper understanding of viral–host regulatory relationships and have broad therapeutic potential by identifying novel drug targets of viral–host regulation and their interlaced phenotypes.

Genetic coupling constitutes an indirect co-regulator of viral and host promoters driven by high similarity in *cis* regulatory arrangement and represents an additional layer to viral–host interactomes. With a long evolutionary history of interactions, viral and host gene promoters may hold clues into regulatory mechanisms and host cell dependencies. In this scenario, the virus has converged to co-express with an array of host cell-coding promoters and pathways for a fitness advantage. Genetic coupling would provide a framework for synthetic gene programming strategies by exibiting robust functionality in diverse chromosomal landscapes and environments faced by competing viruses.

Implementing a comprehensive search for similarity between viral and human promoter sequences, we discover the existence of genetic coupling and co-regulation of highly similar promoters of human immunodeficiency virus (HIV), cytomegalovirus (CMV) and corresponding human genes, whereby co-expression of the HIV long terminal repeat (LTR) promoter and human C-X-C motif chemokine receptor-4 (CXCR4) facilitates migration and reactivation of dormant or latently infected cells. We show that both migration and reactivation can be differentially controlled using drug treatments. These findings further complicate leading strategies for eliminating latent reservoirs of HIV[16], as current approaches risk reactivating latent virus in migrating cells that can reach target-rich cell niches[17].

## Results

### A genome-wide search for viral–host promoter similarity.
To search for promoter similarity between human protein coding genes and the HIV-1 LTR promoter, we performed a logic-based search of the annotated human genome for conserved *cis* regulatory binding sites in the LTR across clades of HIV-1 (Fig. 1a). The search was strictly limited to promoters with binding sites for TBP (TATA-binding protein), specificity protein 1 (SP1) and nuclear factor kappa B (NFKB), and did not filter for site arrangement, exact distance from the transcription start site (TSS)

or number of sites. The presence of additional binding site types were ignored to de-constrain the search and resulted in 366 TBP–SP1–NFKB gene promoters (Supplementary Table 1). Human coagulation factor 3 (F3) and CXCR4 were found to exhibit the highest promoter similarity to the LTR (Fig. 1b and Supplementary Fig. 1). Promoters of HIV-1, HIV-2, simian immunodeficiency virus (SIV) and the CMV major immediate early promoter (MIEP) showed high conservation of TBP–SP1–NFKB arrangement in their core promoters[18–20] (Fig. 1b). Post-search curation revealed additional similarities between the promoters (AP1, nucleosome occupancy, P-TEFb, CPG islands and so on). In addition, similar to the LTR, the distance of NFKB and the most upstream SP1 site in the CXCR4 and F3 promoters is within a reported range required for cooperativity by both transcription factors in promoter activation[21]. Collectively, this binding site arrangement suggests a uniquely conserved viral–host promoter for gene regulation, cooperativity, expression dynamics and an elaborate control and coordination of viral–host gene expression.

### Promoters discovered with highest similarity co-express.
To test whether viral–host promoter similarity drives correlated gene expression, an extensive analysis of exogenous drug perturbations was performed (Fig. 2). Using genome-wide microarrays of diverse drug perturbations[22], CXCR4 and F3 displayed significant co-transcription across ~885 perturbations in a neutrophil HL60 cell line (z-score, $P < 0.04$, Fig. 2a). Of the three cell lines measured in the original study where the genome-wide microarrays were performed, the HL60 neutrophil cell line was chosen as the closest to the Jurkat T-lymphocyte cell line for evaluating co-expression (compared with MCF7 breast cancer and PC3 prostate cancer cell lines in the microarray data set). The co-expression slopes of genetically coupled promoters land high on distributions comparing over 10k human genes with either F3 or CXCR4 (Fig. 2b,c). Co-expression of the HIV LTR and human promoters was quantified using expression data from a recently reported drug screen on a clonal Jurkat T-cell population of the LTR driving an mCherry reporter[23,24]. Accounting for 262 treatments common to the microarray data set, co-expression slopes were quantified between changes to human mRNA transcription and fluorescent intensity of LTR–mCherry expression after applying a moving average (Fig. 2d). CXCR4 and F3 result in significantly high co-expression slopes on a distribution between the LTR and ~10k human genes (z-score, $P < 0.003$ for CXCR4, Fig. 2e). Genes with higher co-expression than CXCR4 or F3 to the LTR promoter exist and most likely occur through alternate modes of *cis*, *trans* and posttranscriptional regulation, and are not co-regulated through the *cis* regulatory binding elements used to determine genetic coupling in this study.

The result of high co-expression for F3 and LTR is consistent with reports showing increased F3 expression in monocytes and plasma under chronic HIV infection[25]. In contrast to CXCR4, a co-receptor in HIV-1 infection, the dominant co-receptor CCR5 having only a single NFKB-binding site and high dissimilarity in promoter architecture exhibited uncorrelated expression with both F3 and the LTR (Supplementary Fig. 2). Strong co-expression between HIV and CMV promoters was shown using flow cytometry of polyclonal cell populations harbouring LTR–GFP or MIEP–mCherry after 24 h treatment with Trichostatin A (TSA), Prostratin (Pro), and tumour necrosis factor (TNF)-α ($R^2 = 0.85$, Fig. 2f). Promoter coupling between distant viral families (RNA and DNA) that co-infect their host, including multiple hosts throughout their evolution (SIV and HIV), demonstrates that genetic coupling may elucidate a deeper understanding of co-evolution of multiple viruses and their hosts[8] along with their

co-infection dynamics. Collectively, the data show that genetically coupled viral and host promoters are co-regulated and co-expressed under diverse perturbations.

**CXCR4-LTR co-expression presents a risk for latency reversal.** Upon infection of CD4 + T-lymphocytes, HIV integrates into the host genome and exploits cellular resources to produce viral

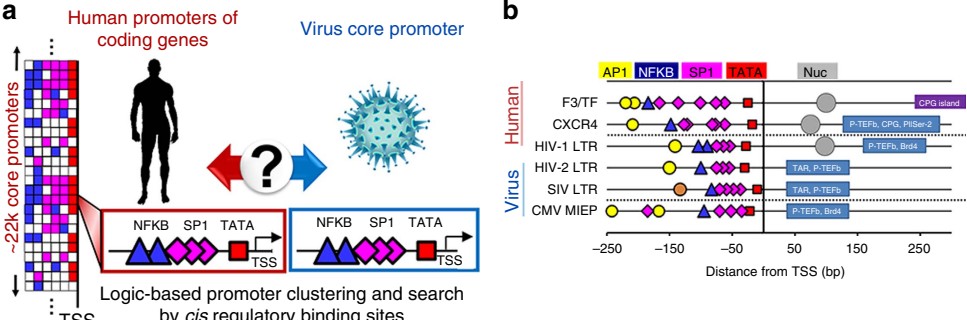

**Figure 1 | Viral and host promoters display high similarity in gene regulatory sequence.** (**a**) Logic-based search of human coding gene promoters for similarity to viral core promoters. The list of human promoters indicates a logic series of *cis*-binding elements (boxes) for each promoter (rows) comprising diverse patterns of TATA box (red), SP1 (pink) and NFKB (blue) binding sites. Conserved *cis*-binding sites in a virus (NFKB: blue triangles, SP1: pink diamonds and TATA: red squares) are searched for in all human coding genes. HIV and CMV promoters are curated and compared within ± 250 bp from the TSS of human promoters. Promoters are clustered regardless of number, arrangement or distance of sites from the TSS. Additional transcription factor-binding site types are ignored to de-constrain results of the search. (**b**) The genome-wide promoter search results in ~ 366 gene promoters with binding sites for TBP (TATA-binding protein, red squares), SP1 (pink diamonds) and NFKB (blue triangles), and F3/TF, CXCR4 and HIV-1 LTR show the highest similarity in *cis* regulatory arrangement. Upstream existence of AP1 sites (yellow circles) and downstream nucleosome occupancy (grey circles), binding of P-TEFb (ref. 69), BRD4 (ref. 69) and CPG islands (purple and navy rectangle regions) from the TSS are also consistent between promoters and curated after the initial genome-wide search. The brown circle in the SIV LTR promoter represents simian factor 1 or SF1, an activator protein 1 (AP1) related transcription factor in monkeys reported in SIV around − 135 to − 131 bp[76].

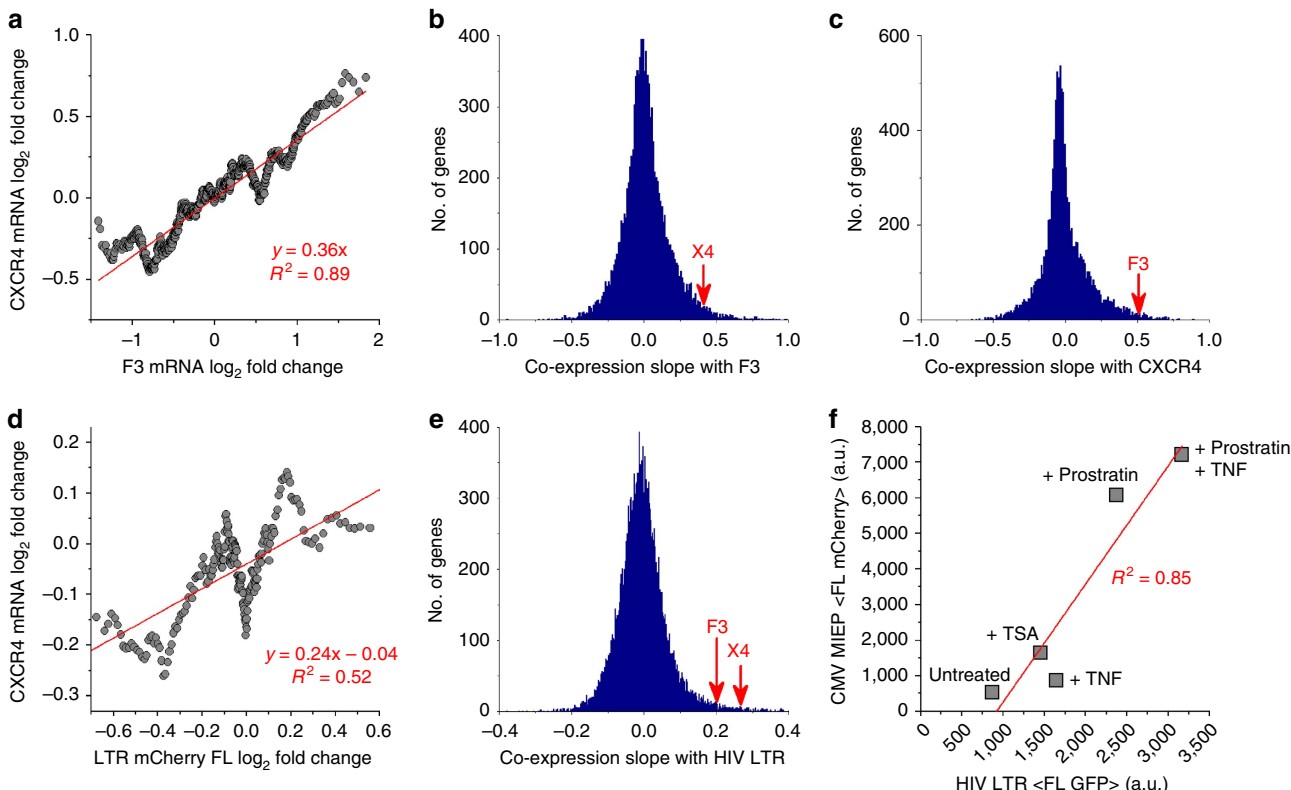

**Figure 2 | Genetically coupled promoters are co-expressed across hundreds of drug perturbations.** (**a**) Comparison of mRNA fold change for over 800 perturbations of CXCR4 and F3 transcription in HL60 cells after a moving average is applied. (**b,c**) Distribution of correlation slopes (slope of red trend in **a**) for over 10k genes compared with either F3 or CXCR4. Both are on the high end of the slope distributions. (**d**) High-throughput flow cytometry of an HIV LTR–mCherry Jurkat cell line[23] treated with 262 drug compounds common to the HL60 microarray data set[22] shows a positive correlation between CXCR4 mRNA levels and mean HIV LTR–mCherry fluorescence. (**e**) F3 and CXCR4 both have high co-expression slopes with the LTR compared with over 10k human genes. (**f**) Mean fluorescence of polyclonal LTR–GFP and MIEP–mCherry Jurkat populations reveals correlation under different drug treatments. All measurements were performed with flow cytometry in duplicate or triplicate.

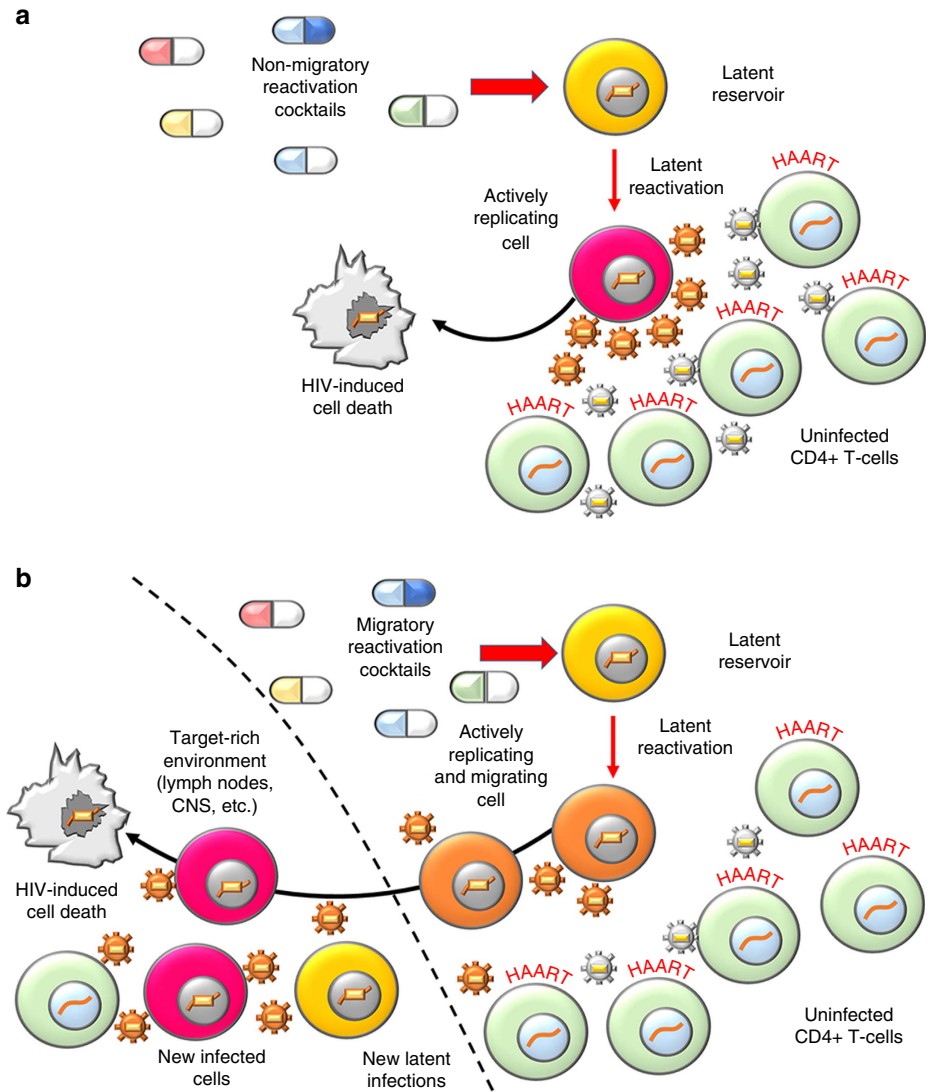

**Figure 3 | Migration challenges elimination of latent HIV reservoirs with drug treatments.** (**a**) An ideal non-migratory reactivation cocktail leads to active replication of latent cells and production of virus in a HAART-treated environment. Reactivated cells die without significant migration and the latent reservoir is cleared[16]. Bystander cells are protected by HAART from newly produced virus. (**b**) Migratory reactivation cocktails induce active replication and migration of latently infected cells. Migrating cells shed virus before cell death. Reactivated cells can reach target-rich cell niches high in CXCL12 ligand and unexposed to HAART, allowing new active and latent infections to occur[38].

progeny. Productive infection leads to active replication, whereas an alternate quiescent state termed proviral latency in which the integrated virus does not express is also established. Eliminating the latent cell reservoir has proven difficult[26] and is identified as the primary barrier for curing infected patients[27–29] as the latent virus re-establishes infection by stochastically switching into a replicating state[30,31]. The leading strategy for eliminating the latent reservoir termed 'shock and kill', entails complete reactivation of the latent reservoir while under treatment with highly active anti-retroviral therapy (HAART)[16]. HAART protects uninfected bystander cells from produced virus, while reactivated cells undergo cell death over time (Fig. 3a). To date, HIV cure research has made progress[23,26,29,32], but the possibility of correlated migration and reactivation of latent cells poses unaddressed challenges for 'shock and kill' therapy (Fig. 3b).

CXCR4 and its ligand CXCL12 constitute a major migration axis throughout the human body, responsible for directed trafficking of T cells and stem cells[33]. CXCL12-rich attractors for cells overexpressing CXCR4 include lymph nodes, the central nervous system, brain, heart and other organs[34–37]. CXCL12 abundant regions consisting of uninfected cells are target-rich for viral propagation and HIV-CXCR4 promotor coupling raises concerns for latency-reversal treatments. Migrating reactivated cells shedding virus can reach unprotected target-rich environments and cause new infections before their cell death[38] (Fig. 3b).

**HIV LTR expression correlates with T-cell migration**. To investigate how LTR-CXCR4 promoter coupling and co-expression affect reactivation and migration phenotypes, migration assays using chemotactic gradients for the CXCR4–CXCL12 migration axis were performed. Polyclonal Jurkat cell lines containing LTR–GFP or MIEP–mCherry at a low multiplicity of infection (that is, 1)[30,31], along with naive Jurkats, showed increased migration after TNF treatment (Fig. 4a and Supplementary Fig. 3). TNF activates NFKB signalling and subsequently the CXCR4 and viral promoters[39,40], revealing

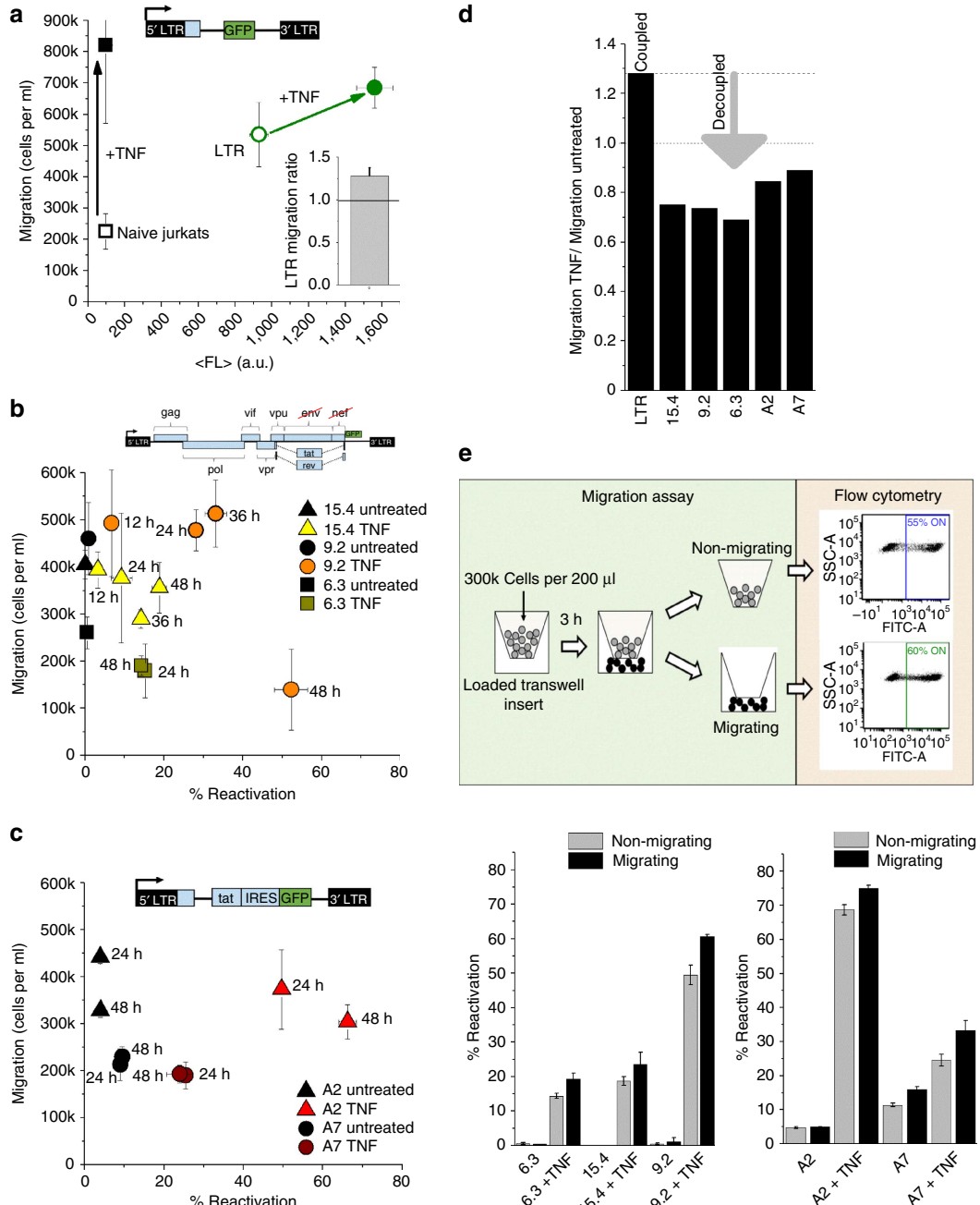

**Figure 4 | Migration in reactivated cell populations harbouring latent HIV. (a)** HIV LTR–GFP displays correlated gene expression as measured by mean fluorescence (<FL>) and migration after 24 h TNF. Naive Jurkat cells show only an increase in migration with TNF. (Inset) Correlated migration increase with TNF is alternatively shown using an average ratio of TNF-treated migration by untreated migration. This calculation accounts for 18 migration measurements of LTR–GFP Jurkats used as a positive control for each experiment. The s.e. is plotted as well. **(b)** Upper: to test migration of latent cells infected with full-length HIV, the Jurkat latency (JLat) model was used. Lower: migration and reactivation of three clonal JLat populations of latent HIV under 12–48 h TNF treatments. All three clones reveal decreased or constant migration with increasing reactivation. The untreated samples of the isoclones 9.2 and 15.4 were measured after 12 h, whereas the untreated sample of isoclone 6.3 was taken after 24 h. **(c)** To test migration and reactivation of Tat-expressing T-cells, a minimal viral circuit with HIV-1 LTR promoter driving Tat, internal ribosome entry site (IRES) and GFP was used (LTR-Tat-IRES–GFP or LTIG)[41]. Migration and reactivation of two clonal LTIG populations (A2 and A7) was measured after 24 and 48 h of TNF treatment. All clones show a decrease or constant migration with increasing reactivation compared to the promoter only case (**a**). This result is consistent with Tat decoupling of migration by inhibiting CXCR4 on the cell membrane for an inverse or constant migration–reactivation relationship. **(d)** Normalized migration of TNF treatment by untreated migration post 24 h shows a coupled phenotype with LTR–GFP and a decoupling caused by Tat reactivation of JLat and LTIG latent clones. **(e)** Upper: reactivation measurements of migrating and non-migrating cells for different JLat clones in the upper (non-migrating) and lower (migrating) wells of a transwell migration plate after 48 h TNF using flow cytometry. Lower: migrating cells consistently reactivate more compared with non-migrating cells after 48 h TNF treatment. All migration and reactivation experiments were carried out in duplicate or triplicate and independently on different days. The average values and s.e. for separate duplicates and triplicates are plotted.

a positive correlation between migration and mean green fluorescent protein (GFP) fluorescence (Fig. 4a).

**Latency reversal decouples LTR expression and cell migration**. To examine beyond the minimal promoter system, we performed migration assays in the context of full-length HIV-1, to determine whether the correlation between migration and viral expression is conserved. Isoclones of a previously generated Jurkat latency model (JLat)[41], a Jurkat cell line containing full-length HIV with a deletion of env and GFP replacing the nef reading frame, were used for migration assays. Cells were reactivated with TNF for 12–48 h followed by 3 h migration measurements and compared with untreated cell migration (Fig. 4b and Supplementary Figs 4 and 5). Counter to the minimal promoter system (Fig. 4a), all clones demonstrate a decoupling between migration and reactivation with either equal or lower levels of migration (Fig. 4b and Supplementary Fig. 6). Downregulation of CXCR4 from the cell surface has been reported for the viral *trans*-activator of transcription (Tat) protein expressed in these cells[42,43]. To investigate whether decreased migration with increased reactivation is controlled by Tat, identical experiments were performed using latent clonal cell lines harbouring minimal viral circuits expressing Tat[41] (LTR-Tat-IRES–GFP, Fig. 4c). Although still migrating, a decoupling of correlated migration and reactivation was preserved (Fig. 4d), suggesting that full-length HIV has evolved the capacity to tune host cell migration using two coordinated mechanisms: co-expression with the genetically coupled CXCR4 promoter and protein interactions at the cell surface between the viral product Tat inhibiting CXCR4 for decreased cell motility.

The observation of decreased migration of reactivated latent cells agrees with recent reports *in vivo*. In a study using humanized mice, Murooka *et al.*[44] report decreased velocity of T-cells productively infected with HIV. Deceleration of *in vivo* migration was conserved for HIV Δenv-infected cells, suggesting that other HIV factors contribute to decreased T-cell motility. Our *in vitro* findings, with Δenv JLat or minimal LTR-Tat-IRES–GFP vectors, suggest that Tat expression during reactivation is sufficient to reduce migration and is consistent with full-length HIV behaviour *in vivo* implicated in playing a role in pathogenesis within target-rich lymph nodes[44]. Strikingly, despite a consistent trend of reduced migration for HIV expressing cells[44,45], JLat and minimal Tat clones treated with TNF for 24 or 48 h reveal that sub-populations of migrating cells reactivate at higher rates compared with stationary (or non-migrating) cells (Fig. 4e, Supplementary Figs 7 and 8, and Supplementary Note 1). Viral control of migration at both genetic (LTR) and protein (Tat) levels motivates the potential for exogenous manipulation of viral–host migration and reactivation.

Regulatory control of latently infected cells with migration–reactivation drug cocktails has implications for latency reversal strategies in diverse scenarios. For example, stimulation of cell migration without reactivating latency may benefit the channelling of HIV-1 strains residing in the central nervous system[17]. A recent study in HIV-infected patients demonstrates that brain-derived HIV-1 strains are less responsive to latency-reversing agents due to polymorphisms occurring in the LTR regulatory SP1 sites[46]. This further supports a need for multi-modal and migratory control strategies addressing the spatial dependence and motility of the latent reservoir.

**Drug control of reactivation and migration of latent cells**. To investigate the ability of leading reactivation cocktails to exogenously and differentially control migration and reactivation,

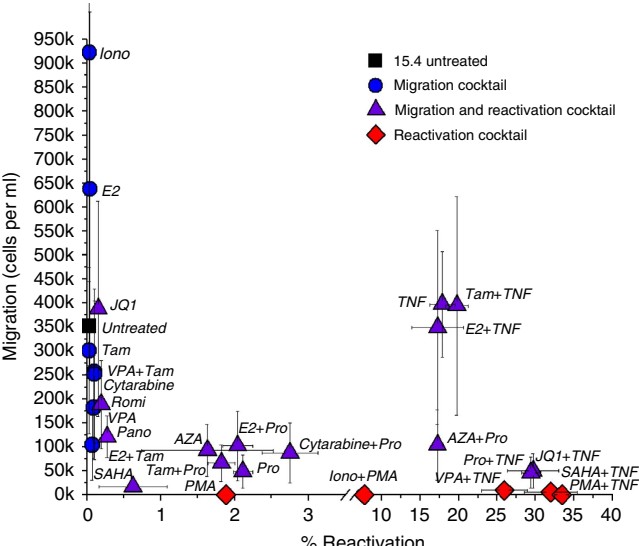

**Figure 5 | Drug cocktails differentially control migration and reactivation of latently infected T-cells.** Drug cocktail treatments control latently infected cell populations between states of migration (blue circles), migration and reactivation (purple triangles) or only reactivation (red diamonds). JLat 15.4 treated for 48 h with leading HIV reactivation cocktails maps diverse migration and reactivation behaviours. Migration and reactivation values represent an average of duplicate or triplicate experiments. AZA, 5-Aza-2-deoxycytidine; E2, 17β-Estradiol; Iono, Ionomycin; Pano, Panobinostat; PMA, Phorbol 12-myristate 13-acetate; Pro, Prostratin; Romi, Romidepsin; SAHA, suberoylanilide hydroxamic acid; Tam, Tamoxifen; TNF, tumour necrosis factor-α; VPA, valproic acid.

latently infected cells were treated with common modulators of HIV transcription (Supplementary Table 2). To exclude the possibility of cell toxicity from applied drug treatments, we used published concentrations (Supplementary Table 2) as well as propidium iodide staining with flow cytometry analysis to assess cell death (Supplementary Fig. 9). For a low reactivating latent clone[41] (JLat 15.4 from Fig. 4b), cell populations fall into three behaviours after 48 h treatments defining distinct classes of drug cocktails (Fig. 5): (1) migrating cells without reactivation (blue circles); (2) migrating and reactivating cells (purple triangles); and (3) reactivating cells without migration (red diamonds). Although TNF and Pro couple migration and reactivation, Phorbol 12-myristate 13-acetate (PMA) and combinations with PMA were found to be the strongest suppressors of migration that reactivate latency. Through a CXCR4 internalization mechanism at the cell surface[47], PMA demonstrates that a treatment strategy directly targeting receptors at the membrane may be more effective and faster than intracellular regulation of the CXCR4 promoter. The results motivate that drug cocktails directly affecting receptors at the cell surface for immobilization are favourable to avoid viral spread through additional infections (Fig. 3b). In addition, cancer treatments such as 17β-Estradiol (E2) and Cytarabine display migration of latently infected cells (Fig. 5) and may pose a risk in HIV + cancer patients treated with these drugs. Custom drug strategies for HIV + cancer patients may prove important given the overexpression of CXCR4 in more than 20 cancer types compared to non-cancerous cells[48–51].

**HIV-infected primary CD4+ T-cells behave similar to JLats.** To confirm that LTR-CXCR4 promoter coupling is conserved in primary human cells, migration assays and fluorescence

measurements were carried out using unsorted and sorted HIV-1 infected primary CD4+ T-cells (Fig. 6a). Despite consistent increases in their mean values, donors 1 and 2 of unsorted LTR–GFP infected CD4+ T-cells revealed no visible increase of migration and fluorescence after TNF treatment when s.e. bars

(measured in duplicate) are taken into account (Fig. 6b). Challenged by the high cell counts required of transwell migration assays, the unsorted LTR–GFP-infected populations provided only a duplicate of each measurement in at least two of three donors with 48 h TNF treatment (Fig. 6b). Here migration was estimated by measuring the percent of GFP+ cells that migrated post 3 h and multiplying this by the total migrated cells in the bottom of the transwell migration plate to calculate the total population of LTR-infected and migrated cells. To prove LTR and CXCR4 co-expression, as seen in polyclonal LTR–GFP Jurkats, we performed migration and fluorescence measurements of GFP+ sorted CD4+ T-cells. GFP+ sorted populations showed robust increases in both migration and reactivation (Fig. 6c). This result is consistent with polyclonal LTR–GFP Jurkats (Figs 6b,c and 4a).

After observing LTR–GFP-infected primary cells, ideal comparison of a latent primary cell model with the JLat results (Figs 4 and 5) requires infecting and sorting resting primary CD4+ T-cells with a JLat vector for the GFP−, or OFF population. Within the OFF population, the latently infected CD4+ cell population is very low compared with the uninfected cell population and, as a purely latent population is needed for the migration assay, it is currently unfeasible to perform with no biomarkers to sort for latency. Consequently, investigation of migration for a mixture of both infected and uninfected cells presents significant challenges (Supplementary Fig. 10). Despite these experimental challenges, activated primary CD4+ T-cells infected with full-length HIV-1 containing env and d2GFP instead of nef (JLatd2GFP) were GFP+ sorted and quantified for migration and fluorescence (Fig. 6d). This GFP+ infected and sorted population may represent actively replicating latent cells, that is, post reactivation. Although deficient in observing the transition of reactivation to a fully activated state, the sorted infected population allows for observation of modulated migration and viral expression phenotypes. These behaviours resemble those of active primary cells reactivated from a resting latent state.

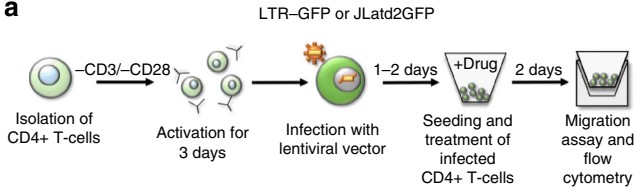

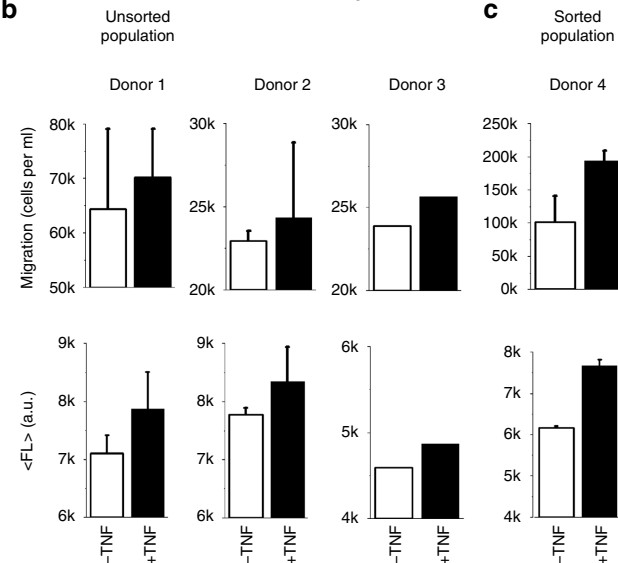

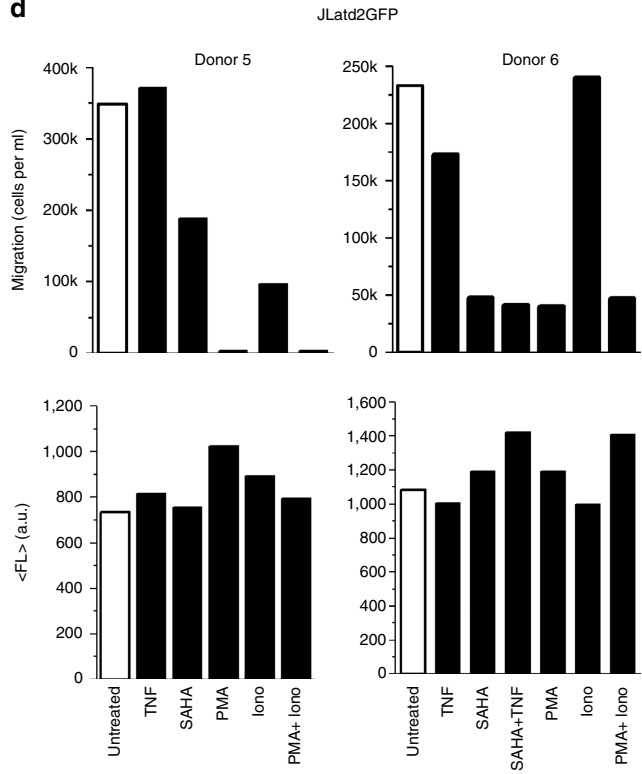

**Figure 6 | LTR-CXCR4 coupling and migratory behaviours of HIV-infected primary CD4+ T-cells.** (**a**) Primary CD4+ T-cells are isolated from fresh human whole blood and stimulated with anti-CD3/anti-CD28 antibodies on the same day. After 3 days of stimulation, the cells are infected with concentrated lentivirus containing either HIV-1 LTR-GFP or the full-length HIV-1 (JLatd2GFP) for 2 h by spinoculation. Next, CD4+ T-cells are sorted for GFP-positive cells 1 day after infection or unsorted cells are used 1–2 days after infection. Sorted or unsorted CD4+ T-cells are then seeded into V-bottom plates and treated with drugs for 2 days. Migration assays were carried out using a 96-well transwell chamber and expression of GFP was measured using flow cytometry. (**b**) 48 h TNF treatment of unsorted LTR-GFP infected CD4+ T-cells from three separate donors shows a correlated increase of both migration and mean fluorescence. Measurements using donor 1 and 2 were performed in duplicate and the average values and the s.e. are plotted. Only a single measurement was obtained from Donor 3. (**c**) GFP+ sorted LTR-GFP-infected CD4+ T-cells show co-expression of LTR and CXCR4-migration after 48 h of TNF treatment, consistent with the Jurkat cells. Error bars are s.e. (**d**) Sorted CD4+ T-cells infected with JLatd2GFP display a decoupling from increased migration after 48 h of TNF treatment. The decoupling of migration is relative to the primary CD4+ T-cells infected with only LTR-GFP in **b**,**c**. In addition, cells treated with SAHA reduce migration consistent with the effect of HDACis in the Jurkat model in Fig. 5. PMA also displays a strong reduction of migration, as well as eliminating migration when combined with Ionomycin (Iono) similar to Fig. 5. For the second donor (donor 6), combination of PMA and Iono reduces enhanced Iono migration and SAHA reduces TNF migration. Measurements were performed on two separate donors.

Using selected drugs from Fig. 5, the measurements show that TNF treatment does not increase migration (that is, decoupled from the LTR–GFP case in Fig. 6b,c), suberoylanilide hydroxamic acid (SAHA) decreases migration, and PMA and PMA+ Ionomycin (Iono) completely suppress migration (Fig. 6d). These results are consistent with treatments on JLats (Figs 4 and 5). Unlike the JLat clones, infected primary cell experiments represent a large range of the integration site landscape of the provirus. Both CXCR4-LTR co-expression and decoupling by full-length HIV are observed in the polyclonal primary cell experiments (Fig. 6).

Finally, to observe CXCR4 migratory response of uninfected primary CD4+ T-cells to diverse drug treatments from Fig. 5, a migration assay was performed (Supplementary Fig. 11). The results highlight the dominant effects of diverse drug families (histone deacetylase inhibitors (HDACis) and protein kinase C (PKC)) on the reduction of migration, irrespective of the infection state of the cell. Drug effects on migration of uninfected CD4+ T-cells show that PMA reduces Iono-treated cell migration, SAHA reduces TNF migration and Pro reduces enhanced migration by E2. Off-target effects increase the migratory challenge for affecting infected CD4+ T-cells with minimal influence on bystander cells using strategic drug therapies for migration in and out of target-rich niches without producing new infections (Fig. 3b).

## Discussion

The computational investigation of similarity between viral and human coding promoters and their *cis* regulatory arrangements guided the experimental investigation of a novel regulatory relationship between a virus and its host. CXCR4 is typically studied for its role as a mediator of HIV viral entry at later stages of infection[52]. Despite genetic coupling and co-expression observed between HIV and CXCR4, the dominant CCR5 tropism of HIV is completely uncoupled genetically and uncorrelated in response to drug perturbations (Supplementary Fig. 2). This results in a dual-role of HIV-CXCR4 with membrane interactions and promoter co-regulation for both infection and migration of the host cell. Co-expression between the LTR and CXCR4 in polyclonal LTR–GFP cell populations shows that the promoter coupling is largely invariant to differences in episodic transcriptional activity of LTR integration across the human genome[24]. Interestingly, the observed decoupling of migration in the minimal and full-length latency models that express Tat suggests that the virus has evolved to synchronize viral gene expression with host cell CXCR4 upregulation, to control or decrease CXCR4-mediated migration individually within each infected cell, potentially for a fitness increase[44]. In addition, the consistent difference in reactivation levels of stationary and migrating cells may be also under the influence of the viral Tat protein. Tat transactivation is heterogeneous in clonal populations with single-cell transients stochastically initiating from the inactive state when reactivated, which could lead to asynchronous coupling of steady state levels[30,31].

Best-in-class drug cocktails (Fig. 5) were used to quantify migration and reactivation, and to identify major migration–reactivation behaviours of Tat-expressing cells. Consistent with previous reports, HDACis and PMA reduce CXCR4-mediated migration. In addition, certain preferred synergistic compounds in reactivation cocktails such as JQ1 (ref. 53) and Iono may migrate treated cells when a migratory suppressor is not combined simultaneously. A subset of compounds combined with TNF provided the largest reactivation with reduced migration. Although consistent with a general trend of decreased migration with increasing reactivation from Fig. 4b,c, sole treatment and quenching of migration by SAHA and PMA suggests a consistency with their known internalization of CXCR4 at the membrane and a complete reduction of TNF migration levels, that is, membrane modification dominates intracellular NFKB activation (Figs 5 and 6d). Treatment with a combination of JQ1 and TNF, both migrating compounds when treated independently, results in high reactivation and a strong reduction of migration, whereas migrating compounds E2 and Tamoxifen conserve both migration and reactivation levels when combined with TNF compared with TNF alone (Fig. 5). Finally, anticancer drug treatments of E2 or Cytarabine raise additional risk in HIV+ cancer patients by migration of latently infected T-cells, which underlines the need for customized latency removal treatments individualized for each patient and their pre-existing medical conditions. In addition, migratory cocktails and their diverse mechanisms affecting both the latent reservoir and uninfected bystander T-cells will need to be accounted for in designing migration strategies (Figs 5 and 6, and Supplementary Fig. 11). These results set the stage for either advanced gene therapies utilizing synthetic biology or large-scale screening of small molecules that strategically modulate migration–reactivation behaviours.

Customized therapeutic strategies are further needed as recent studies contribute to an emerging picture of prolonged cell viability impeding eradication of HIV-infected cells. Reports using mathematical models to provide an understanding of *in vivo* replication kinetics of HIV estimate a generation time of ∼1–2 days for productively infected cells[54,55]. These theoretical models do not directly measure apoptotic death for productively infected cells or latency in patients on long-term HAART. Recent reports further extend the possibility of prolonged cell death through experimental investigations by showing that: (1) HIV-1 can acquire mutations to evade cytolytic T lymphocytes required for clearance of the latent reservoir[56], (2) latency reversal agents impair cytotoxic T-cell-mediated killing[57–59] and (3) clinical trials with HDACis and disulfiram fail to deplete the latent HIV reservoir in patients on HAART despite HIV reactivation[60]. Collectively, these studies provide evidence that T-cell killing of reactivated latent cells is difficult and may not occur at all. Migratory challenges to HIV 'shock and kill' presented in this study are only further exacerbated by this lack of T-cell death by providing a longer migration window accompanied by viral shedding of reactivated cells (Fig. 3b). Furthermore, in a latent population, migrating cells reactivate more than stationary cells (Fig. 4d and Supplementary Figs 7 and 8), presenting an additional risk for viral spread with 'shock and kill' (Fig. 3b).

Computational logic-based promoter searches for similarity in gene regulatory arrangements can elucidate additional genetic coupling and guide drug therapies. An unexpected genetic coupling of HIV-LTR and CMV-MIEP promoters was uncovered between two distinct and co-evolving viruses (Fig. 2f and Supplementary Fig. 3). CMV has been prominent in patients infected with HIV-1 and the relationship between both viruses is important for persistence under current HAART treatment[61]. LTR-MIEP coupling not only suggests that viral–host coupling can include co-evolution of multiple viruses, but may elucidate a more comprehensive viral–host network and treatment for common co-infection in patients[61].

Moreover, human F3 (tissue factor), another coupled promoter, is involved in inflammation, cellular homing, blood coagulation, hypercoagulation and thrombosis (Figs 1 and 2). Thromboembolic complication has been observed in HIV+ and AIDS patients[62–65], and along with inflammation is elevated in HIV-infected patients[65,66] with mixed reports regarding

those under HAART[63,65]. With a coagulation–inflammation–thrombosis circuit[67] affected by both HIV and CMV infection, the findings in this study may provide mechanistic insight for the elevated risks for thrombosis in patients and the design of novel treatments for hypercoagulation and inflammation.

Coupled transcription factor control of promoters has implications for advancing predictive DNA networks and understanding interactomes of human disease. Future studies of genetically coupled viral–host promoters may elucidate mechanisms of viral–host regulatory dynamics, viral fitness and innovative therapies. The discovery that HIV exploits transcriptional coupling with host cell pathways provides a novel paradigm for viral–host DNA interactions and the future engineering of synthetic systems. Although adding challenges to HIV therapies, this study reports the existence of a genotype to phenotype coupling of migration and reactivation of latent HIV. Improving regulatory control of migration–reactivation behaviours with drug cocktails may benefit migration-based disease phenotypes such as HIV and cancer.

## Methods

**Cell culture and growth conditions.** Jurkat cells were cultured in RPMI 1640 with L-glutamine and 25 mM HEPES (Thermo Scientific) supplemented with 10% fetal bovine serum (FBS) and 1% penicillin and streptomycin (Corning Cellgro). Cells were held in healthy and continuous growth conditions (5% $CO_2$ and 37 °C).

For isolation of primary CD4+ T-cells, healthy peripheral blood mononuclear cells were isolated from fresh human whole blood (Innovative Research, MI, USA and BioreclamationIVT, NY, USA) under the collection of buffy coat followed by Ficoll-Hypaque density gradient centrifugation. Total CD4+ T-cells were immediately isolated by negative selection using RossetteSep Human CD4+ T-Cell Enrichment Kit (Stemcell Technologies). Primary T-cell stimulation was carried out on the same day of isolation using $1 \times 10^6$ CD4+ T-cells per ml cultured in RPMI 1640 medium supplemented with 10% FBS, 1% penicillin/streptomycin, 30 U ml$^{-1}$ human interleukin 2 (Miltenyi Biotec Inc., CA, USA) and $1 \times 10^6$ Dynabeads Human T-Activator CD3/CD28 (Thermo Fisher Scientific).

**Cell lines and transfection.** Jurkat cells were obtained from ATCC (Naive Jurkats), NIH AIDS Reagent Program (JLats and latent LTR-Tat-IRES–GFP (LTIGs)) and the Weinberger Laboratory at the Gladstone Institutes at UCSF (LTR–GFP and MIEP–mCherry polyclonal). Briefly, naive Jurkats were infected with a vector consisting of the HIV-1 LTR promoter driving a stable GFP (CLG polyclonal cell line[30]) or the full-length CMV MIEP promoter driving mCherry tagged with a nuclear localization signal[68]. Cells were infected at a low multiplicity of infection (1) and sorted polyclonal populations were used for comparing co-expression of LTR and the MIEP. LTR–mCherry isoclone 20 used here (Fig. 2) has been previously described[23,24]. For HIV-1 latency studies, JLat isoclones 6.3, 9.2 and 15.4 consisting of full-length HIV with GFP replacing the nef reading frame and a deletion of env were selected from a previously generated library[41]. Minimal latent virus utilized isoclones A2 and A7 previously integrated with a LTIG vector[41].

The infection of CD4+ T-cells included the production of JLatd2GFP lentiviral supernatant. For this, HEK293 cells ($5 \times 10^5$) were co-transfected using FuGENE6 transfection reagent (Promega) according to the manufacturer's instruction. The transduced cells were harvested 24 h after media change and the viral supernatant was centrifugated at 500 g to remove all remaining cells. To concentrate the remaining lentiviral supernatant, a lentivirus concentration reagent (Takara Bio Inc., CA, USA) was used according manufacturer's instruction. After centrifugation at 1,500 g for 45 min at 4 °C, the off-white pellet containing the lentivirus was resuspended in 1/40 to 1/80th of the original volume using cold DMEM supplemented with 10% FBS and 1% penicillin/streptomycin.

For infection of primary CD4+ T-cells with lentiviral supernatant, $1 \times 10^6$ activated CD4+ T-cells per ml containing anti-CD3/anti-CD28 beads from 3 days of stimulation were collected as pellets by centrifugation at 500 g for 10 min at room temperature and resuspended in 60 µl of 1/40 to 1/80 concentrated lentiviral suspension consisting of either the HIV-1 LTR promoter driving a stable GFP (analogue to the CLG polyclonal Jurkat cell line) or the replication incompetent full-length HIV-1 genome with a deletion of the viral env protein and replacement of the nef reading frame by a destabilized d2GFP (Weinberger Laboratory at the Gladstone Institutes at UCSF) and topped with 40 µl RPMI 1640 medium supplemented with 10% FBS and 1% penicillin/streptomycin. Cells and virus were centrifugated at 1,200 g for 2 h at room temperature. After spinoculation, stimulation beads were removed using a magnet and $1 \times 10^6$ cells per ml were resuspended in RPMI 1640 medium containing 10% FBS, 1% penicillin/streptomycin and 30 U ml$^{-1}$ interleukin 2.

**Sorting of CD4+ T-cells.** To validate isolation of CD4+ T-cells, cells were stained with a primary anti-human CD4 monoclonal (SIM.4) antibody (NIH AIDS Reagent Program, MD, USA) and a secondary goat anti-mouse IgG-PE antibody (Santa Cruz Biotechnology, TX, USA). Cells were stained for 30 min at 4 °C and 20 min at 4 °C with the primary and secondary antibody, respectively. After washing twice with staining buffer, the cells were sorted for either GFP+ or GFP− populations using a BD FACS Aria II.

**Analysis of promoter gene sequences.** The annotated human genome sequence was searched for genes with promoters similar to the HIV-1 LTR promoter. A logic-based approach was implemented to group all genes that contain at least one of the most highly conserved cis regulatory binding sites (SP1, NFKB and TBP) of the LTR core promoter at a range of distances within the human promoters. Three hundred and sixty-six promoters were detected to have at least one site of all three binding site types, including SP1 detected in the core promoter, and NFKB and TBP detected within ±1 kbp from the TSS (Supplementary Table 1). Promoters were grouped regardless of the number, arrangement or location of binding sites on the promoter. All other transcription factor binding sites not included in this conserved set were ignored in the search and identification of promoter similarity. All 366 promoters in the TBP-SP1-NFKB group were manually scanned in a ±250 bp range from the TSS for similarity to the HIV-1 LTR (Fig. 1 and Supplementary Fig. 1). Finally, discovered promoters with highest similarity in the gene group were manually curated for promoter arrangements with closest similarity to the LTR including additional binding site types (Fig. 1b). Binding sites for AP1 and epigenetic features downstream of the TSS such as nucleosome occupancy, P-TEFb[69], BRD4 (ref. 69) and CPG islands were manually curated and obtained using the UCSC genome browser[70]. HIV-1, HIV-2, SIV and CMV MIEP promoters were obtained from openly available sequences and published literature[19,20,71–73]. Binding motifs for SP1, NFKB and TBP were obtained from Transfac[74]. Fimo was used to identify the locations of sequences that significantly match the motifs of interest ($P < 1E-4$) within specified ranges of all annotated TSSs.

**Perturbation microarrays and drug screening.** Genome-wide microarrays accounting for 885 treatments of an HL60 cell line were used to quantify co-expression between host and viral promoters[22]. Additional details on the microarrays can be found in the original paper from Lamb et al.[22]. In brief, genome-wide microarrays were performed on three cell types (HL60, MCF7 and PC3) exposed to thousands of diverse perturbagens[22]. Total RNA was isolated and synthesis of cRNA target, its hybridization to microarrays and scanning of those arrays was performed using Affymetrix GeneChip products and reagents. Total RNA from development batches was processed manually using HG-U133A cartridge arrays (part number 510681). Total RNA from production batches was amplified and labelled using the GeneChip Array Station and hybridized to HT_HG-U133A (early access version; part number 520276) High-Throughput Arrays, which were scanned using the HT Scanner. The data used in this publication are deposited in NCBI's Gene Expression Omnibus (GEO, http://www.ncbi.nlm.nih.gov/geo/) and is accessible through GEO series accession number GSE5258. Data are also available for download at http://portals.broadinstitute.org/cmap/.

Details on drug screening of the HIV LTR promoter with 1,600 bioactive and biodiverse small-molecule compounds (MicroSource Discovery Systems Inc.) can be found in Dar et al.[23]. In brief, a Jurkat isoclone cell line, isoclone 20, consisting of LTR-d2GFP and LTR–mCherry was treated with 10 µM for 24 h starting in log-growth phase. Unfixed 96-well plates consisting of the samples were run on a BD LSRII flow cytometer with high-throughput sampling module. Each plate consisted of untreated and TNF-treated positive control columns for post-analysis calculations between plates from different days of screening.

To calculate co-expression slopes, log$_2$ fold change microarray values across all treatments between any two promoters were processed using a moving average of 50 treatments followed by calculation of the slope for a linear fit between the responses of two genes. For the viral–host co-expression slope quantification (Fig. 2d,e), 262 treatments shared between the microarray data set[22] and drug screen of the isoclonal Jurkat population harboring LTR–mCherry with flow cytometry[23] were used. Co-expression slope calculations between viral and human promoters was identical to calculations made between two human promoters.

**Reagents for migration and reactivation assays.** For migration experiments, Jurkat cells were stimulated with TNF at a final concentration of 10 ng ml$^{-1}$. The anticancer drugs Romidepsin and Panobinostat were used at a final concentration of 5 and 15 nM, respectively. The histone deacetylase inhibitor TSA was applied at a final concentration of 400 nM and cells were treated with the chemotherapy agent Cytarabine at a final concentration of 0.35 µM. Iono and JQ1 were used at a final concentration of 1 µM, SAHA at 2.5 µM, Pro at 3 µM, 5-Aza-2-deoxycytidine at 5 µM, E2 and Tamoxifen at 10 µM and valproic acid at 1 mM. Cells were treated with PMA, a protein kinase C agonist, at a final concentration of 200 ng ml$^{-1}$. All chemicals, except for TSA (Sigma-Aldrich) and TNF (R&D Systems), were obtained from Cayman Chemicals. Primary CD4+ T-cells were stimulated with

TNF, PMA, SAHA, Iono and a combination of PMA and Iono using the same concentrations as used for Jurkat cells.

**Migration assay.** Migration assays were performed in 96- or 24-well transwell plates consisting of 5 μm pore polycarbonate membranes (Corning Inc., Products 3388 and 3421). Twenty-four-well transwell chambers were used to investigate reactivation of migrating (lower well) and non-migrating (upper well) populations of latent HIV-1 using Jurkat cell lines. The larger 24-well insert enabled collection of sufficient cells to run flow cytometry post migration (Fig. 4b–e). All other migration assays used the 96-well transwell plate system. Briefly, the cells were grown to a density of $\sim 1 \times 10^6$ cells per ml for the JLat cell lines and 1 ml of cell suspension was transferred to a 24-well plate and stimulated with a single reagent or a combination of reagents. Stimulated primary CD4+ T-cells were seeded at a concentration of $0.08 \times 10^6$ cell per ml into a 96-well V-bottom plate and treated with chemical compounds. All migration experiments included a polyclonal LTR–GFP Jurkat population with TNF as a positive control and untreated cells served as a negative control. Treated and untreated cells were incubated for 48 h in 37 °C and 5% $CO_2$. After incubation, all JLat cell lines were pelleted by centrifugation ($500\,g$) and resuspended in migration medium (RPMI 1640 with L-glutamine and 25 mM HEPES supplemented with 0.5% BSA) at a concentration of $0.2 \times 10^6$ cells per 100 μl for 96-well or $0.3 \times 10^6$ cells per 200 μl for 24-well plates in preparation for loading the migration chamber. For migration assays with primary CD4+ T-cells, cells were resuspended in migration media using a concentration of $0.05 \times 10^6$ cells per ml[75]. Human CXCL12 (or SDF-1, R&D Systems) was diluted to appropriate concentrations in migration medium ($25\,ng\,ml^{-1}$ for all migrations except for the dose response in Supplementary Fig. 3) and added with a volume of 0.6 ml (24-well plate) or 0.15 ml (96-well plate) to the lower chamber of the transwell. The upper wells were inserted on top and diluted cells were loaded into the upper chamber followed by inspection for and removal of air bubbles under each well. Cells were allowed to migrate for 3 h at 37 °C and 5% $CO_2$. For migration experiments using a 96-well plate, all cells in the lower well were counted using a MOXI Z cell counter (ORFLO). For experiments with unsorted HIV-1 LTR-infected CD4+ T-cells, the content of the bottom well was also used for measuring the percent of fluorescent cells using flow cytometry and for calculating the percentage of migrating GFP-expressing cells.

Total number of GFP+ migrated cells = (% GFP+ of unsorted infected CD4+ T-cells in the bottom of the transwell) × (total number of migrating cells in the bottom transwell).

Migrated treated cells were viable at the time of measurement as measured by propidium iodide staining (Supplementary Fig. 9) and spun down, pelleted and resuspended in migration media before seeding the transwell for migration measurements. For 24-well migration format, all cells from the upper and lower wells were used for flow cytometry and lower wells were counted. Except for sorted and infected JLatd2GFP primary CD4+ T-cells, all migration experiments were carried out in duplicate or triplicate on separate days and the average values are plotted.

**Flow cytometry analysis.** Flow cytometry was performed using a BD Fortessa flow cytometer whose performance is calibrated daily. All samples of 96-well plates were measured 48 h post treatment. Samples for migrating and non-migrating experiments of JLat cell lines were measured for 12–48 h TNF treatment. JLat cell lines and primary CD4+ T-cells were gated using side scatter versus GFP intensity to differentiate between GFP-positive and GFP-negative cells, compared with conservative gating determined by both naive Jurkat and untreated JLat populations, as well as uninfected, untreated primary CD4+ T-cells (see Flow cytometry gating strategy, Supplementary Fig. 5). Ten thousand cells in the live gate were collected for all measurements using FSC versus SSC. Gene expression of polyclonal Jurkat LTR MIEP and CD4+ T-cell populations was quantified by the mean fluorescent intensity of all live cells collected.

**Data availability.** Mircroarray data that support the findings of this study have been deposited in GEO with the primary accession code GSE5258. These data are also available for download at http://portals.broadinstitute.org/cmap/. Other data that support the findings of this study are available from the corresponding author upon request.

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

## Acknowledgements

We are grateful to Michael Simpson, Melanie Ott, Katherine Pollard, Brandon Razooky, Nina Hosmane, the Leor Weinberger Laboratory, J.J. Miranda and Christopher Brooke for fruitful discussions. We thank the Weinberger Laboratory for the kind gift of plasmids. We thank Alisha Holloway, Sean Thomas and Kristen Eilertson for bioinformatics support at the Gladstone Institutes Bioinformatics Core. We thank Marielle Cavrois and Marianne Gesner for support at the Gladstone Flow Cytometry Core; Barbara Pilas and Angela Kouris at the UIUC Flow Cytometry Facility; and Susan McKenna for assistance with graphics. This work was partially funded by a UCSF CTSI-SOS RAP Core Award. ENT and MRM acknowledge support provided by the Cancer Scholars Program at the University of Illinois at Urbana-Champaign. R.D.D. was supported by the NIH Ruth L. Kirschstein National Research Service Award (AI104380) and an NIH NIAID Career Transition Award (AI120746). The following reagents were obtained through the NIH AIDS Reagent Program, Division of AIDS, NIAID, NIH: J-Lat Full Length Clones (clone 6.3, 9.2 and 15.4) and J-Lat Tat–GFP Clones (A2 and A7) from Dr Eric Verdin.

## Author contributions

R.D.D. conceived the experimental, analytical and computational work. K.B.-W., E.N.T., M.R.M. and R.D.D. designed and performed the migration experiments, flow cytometry measurements and drug treatments. K.B.-W. performed the primary cell experiments. R.D.D. and K.B.-W. analysed and interpreted the experimental data. R.D.D., K.B.-W., E.N.T. and M.R.M. wrote the manuscript.

## Additional information

**Competing financial interests:** The authors declare no competing financial interests.

