## [Peer Review File · Nature Communications]

Reviewers' comments:

Reviewer #1 (Remarks to the Author):

The authors present an interesting study exploring co-regulation of viral and human gene promoters. They compare transcription factor binding sites in HIV-1 to roughly 22,000 human gene promoters. They identify high similarity between HIV (and other viral promoters) to CXCR4, F3 and other human gene promoters. They subsequently show using drug perturbation microarrays that human CXCR4 and F3 promoters have similar co-expression with HIV promoter in cell lines. Reactivation of latent HIV promoter and CXCR4 migration was then found to be co-regulated using TNF stimuli in cell lines, with the interesting caveat that HIV Tat expression impairs cell migration likely due to Tat inhibition of CXCR4 (reported elsewhere). As the authors find a small increase in HIV reactivation in migrating versus non-migrating cell lines, they suggest reactivation drugs that also promote CXCR4-mediated migration could direct reactivated cells to tissue sites with poor penetration of anti-retroviral therapy (ART) and cause unwanted spreading infection. Although, as their study also shows that HIV Tat impairs cell migration, one could speculate that potent reactivation of latent HIV and Tat generation would prevent cell migration and avoid this issue. Finally, the authors identify HIV latency reversing agents (LRAs) that potentially reactivate HIV but limit cell migration in a cell line, and thus may have more benefit in reactivating latent HIV in patients while avoiding spreading infection in tissue sites.

Overall, this is a well written, intriguing study unveiling a previously unknown coupling of viral and human gene promoters, where HIV-1 and CXCR4 promoter co-expression might impact cell migration during reactivation of latent HIV. This will be of interest not only to the HIV latency field but also the broader virus-host factor field given the initial findings reveal co-regulation of HIV or CMV promoter with other host genes. Further studies on HIV or CMV expression with these other host genes seem likely to follow.

However, the experiments exploring reactivation of latent HIV and CXCR4 migration were performed in cell lines using modified HIV constructs. These are not strong models of HIV latency in patients on ART, where latency commonly arises in resting CD4+ T-cells (not cancerous dividing cells) and HIV DNA has integrated into diverse human sites rather than a single integration site in many clonal cell lines used here. For broader interest to the HIV latency field and to have more influence upon thinking in the field, the key findings of co-regulation of HIV and CXCR4 promoters, and impact of LRAs on reactivation of latency and cell migration need to be performed in primary human resting CD4 T-cells that are either latently infected in vitro or from HIV individuals on ART ex vivo. While this will take time, especially if the lab is not equipped to perform such studies, this could be readily achieved by collaboration with multiple investigators currently performing such patient primary T-cell studies in the field. This additional work would help strengthen these latter findings of the paper by demonstrating that these results also apply to latently-infected, primary resting CD4 T-cells and thus are important for researchers developing new LRAs to consider.

There are also some further technical aspects and questions to consider:

1. In Fig 4, please clarify whether the average results in each panel derive from one independent experiment performed in duplicate or triplicate, or from 2-3 independent experiments performed on separate days. Error bars should also be added to data points in Fig. 4a-c so it is clear results are consistent across independent experiments when error is taken into account. In Fig 4d, are the small reactivation increases in migrating versus non-migrating cells maintained in 3 independent experiments? In Fig 4c, are the small migration changes in A2 clone at 24hr or A7 clone at 48hr still seen if the error is taken into account? Also for Fig 4b, all time points should be shown with error bars for both untreated and TNF-treated cells on the same graph given the migration of untreated cells vary over time, especially for clone 9.2 in Supp Fig 4. With all time points on the same graph, clone 9.2 may show little difference between TNF treated and untreated cells even at 48hr given the migration of untreated cells is <100K. Additionally, are declines in TNF-treated

clones 15.4 and 6.3 still present across time points when error is accounted for? Finally in the Fig 4c legend, the authors did not specifically show that Tat was stripping CXCR4 off the cell membrane so this cannot be concluded. The authors should either show this data using CXCR4 staining and FACS analysis, or reword this sentence to say that the results were "consistent with" Tat-related deceleration of migration by inhibiting CXCR4 on the cell membrane etc.

2. If Tat prevents CXCR4-mediated cell migration (Fig 4b-c), why are the cells with higher reactivation in Fig 4d the migrating cells rather than non-migrating cells? Wouldn't the Tat argument predict that as HIV reactivation increases, HIV Tat increases, thus CXCR4 migration declines and non-migrating cells should have more reactivation? This should be addressed in the manuscript.

3. For Fig 5, was the toxicity of the various drugs tested on J-Lat 15.4 cell line eg. MTS metabolic assay or Live/Dead flow stain? This should be tested and included as supplementary data as some reductions in migration might arise from toxic effects of the drugs rather than direct effects on CXCR4 function.

4. The authors should discuss how results in Fig 5 relate to Fig 4 given there are many examples in Fig 5 where more potent reactivation by LRA combinations correspond with impaired migration, complementing Fig 4b-c results.

5. The manuscript is succinct and discusses the results in context with relevant literature. Although the statement and presumption that "reactivated cells die through apoptosis over a 1-2 day period" (line 121, Fig 3 model) over simplifies a more complex matter. The 2 cited references estimate a 1-2 day life span of productively infected cells in chronically infected patients after initiation of 1 or more antiretroviral drugs. However these references do not specifically show that productively-infected cells die by apoptosis. Secondly, these papers do not estimate the life span of reactivated, latently infected cells from patients on longer term ART, which may vary according to the reactivation stimulus and potency eg. T-cell activation versus less potent LRAs. How well patient cytotoxic T-cells recognise reactivated cells will also influence the life span of reactivated cells, especially as cytotoxic T-cell function is compromised in patients that initiate ART during chronic infection due to CTL escape mutations in latent viruses (Deng 2015 Nature 517:381-5). Certain LRAs also impair cytotoxic T-cell mediated killing (Shan 2012 Immunity 36:491-501; Jones 2041 Plos Path 10:e1004287; Jones 2016 Plos Path 12:e1005545). Clinical trials with HDACis or disulfiram also failed to decrease the HIV reservoir in patients on ART despite reactivating HIV RNA (Rasmussen 2016 Curr Opin HIV AIDS 11:394-401) again questioning the link between reactivation and cell death. Therefore the life span of reactivated cells and how they die is unclear, and this part of the paper stated as fact needs to be modified for accuracy.

Overall, this manuscript is an intriguing study with novel findings of viral-host promoter co-regulation likely to be of interest to the HIV latency and broader HIV-host factor fields. However, results from cell lines showing co-regulation of HIV and CXCR4 promoter, and the impact of LRAs on HIV reactivation and cell migration, would be more convincing if key experiments were performed in latently infected primary human T-cells that better model HIV latency in patients.

Minor comments:

1. Fig. 1: In panel A, it would be helpful to add the gene name to the list on the left side for clarity. The magnified human gene promoter does not match the item in the gene promoter list on the left, which has 2 NFkB, 3 SP1 and 1 TATA box. Shouldn't this match? Please add what the boxes signify to the figure legend: NFkB blue, SP1 pink, TATA red, white box absent site. In panel B, what is the brown circle in the SIV LTR? Please define F3/TF in the legend.

2. Supplementary Fig 1: For clarity, the same symbols and colours for SP1 and NF-KB should be

used in both Fig 1 and Supp Fig 1. HV-1 LTR, HIV-2 LTR and CXCR4 should be included in Supp Fig 1 for easy comparison.

3. Fig 5 & Supp Table 1: Why is Prostratin alone in the "Migration and Reactivation" classification when it has less migration (48550 cells/ml) than JQ1+TNF (50350 cells/ml) that was put in the reactivation only classification? This should be consistent. In Supp Table 1, please add untreated cells at the top of the table for comparison so it is clear which drugs decrease migration compared to untreated cells. Instead of the phenotype column, it would be more meaningful to make this column "Transcription activities" and list the known activities of these drugs modulating transcription eg. Prostratin - activates PKC and NF-KB. When the untreated cells are added to the table, the migration and reactivation results speak for themselves without needing the phenotype column.

4. Lines 126-129: Please provide references for CXCL12 being enriched in the tissue sites listed and attracting cells that overexpress CXCR4.

Reviewer #2 (Remarks to the Author):

A. Bohn-Wippert et al have conducted an investigation into the similarities between human protein coding genes and the HIV-1 LTR promoter. They found that human coagulation factor 3 (F3) and the chemokine receptor CXCR4 exhibit the greatest promoter similarity to LTR. Microarray analysis using exogenous drug perturbations revealed that F3 and CXCR4 exhibited a correlated co-expression with the LTR. However, the dominant co-receptor CCR5 exhibited an uncorrelated co-expression with F3 and the LTR. Using a flow cytometry method Bohn-Wippert and colleagues also found a strong co-relation of expression between the HIV and CMV promoters which may indicate co-evolution of viral families able to cause asymptomatic latent infections. Some very elegant migration assays are used to determine if HIV latency reversal agents reactivate migrating cells.

B. Although this is a unique study there are several issues which should be addressed by the authors concerning their methods, results and conclusions.

C. The format of this manuscript was a little difficult to follow without section headings being included or numbers on the figures.

The meaning of one sentence is unclear. This sentence begins on line 106: "The finding of promoter coupling between distant viral families able to cause asymptomatic latent infections,.... ends on line 109

More description of the microarray test and analyses is needed in the methods section.

D. The findings of control perturbations for the microarray assay are not described in the methods section. In addition, the effects of the cell lines upon the microarray results should be described. Overall more description of the microarray assay is required.

E. The authors do not address the issue as to why the structurally related chemokine receptors, CXCR4 and CCR5, have different correlations of co-expression with F3 and the LTR. The authors should provide the different expression levels of CCR5 and CXCR4 during the exogenous drug perturbations and an explanation as to why the CCR5 receptor is not co-expressed with the LTR. Due to the fact that all primary HIV strains can use the CCR5 receptor to infect cells, the impact of their study showing CXCR4 and LTR promoters exhibit correlated co-expression patterns is a little unclear.

It is unclear what the genetic coupling of HIV and CMV promoters during diverse drug perturbations really mean for latent viral infections. A more definitive description of how co-expression between HIV and CMV promoters provides a deeper understanding of co-infection and latent viral infection is needed. The particular drug perturbations which revealed this co-expression

of viral promoters should be described in more detail.

F. The authors found that polyclonal LTR-GFP Jurkat cells treated with the latency reversing agent SAHA alone or in combination with TNF or PMA resulted in an inverse reactivation-migration relationship indicating cells reactivated by these compounds do not seem to migrate. This is an important finding but the authors should have included more potent latency reversal agents such as panobinostat and/or romidepsin in their analysis of migration versus reactivation analyses.

G. References are sufficient.

H. The authors should describe in more detail the importance of the compound mixtures which enhanced cell migration, especially in regards to the treatment of HIV-infected individuals with these drug combinations. Furthermore, would their results be the same if resting memory T cells containing latent proviruses were used for this migration assay. The concluding paragraph describing what their findings mean for the treatment of HIV disease should be expanded. The manuscript was a little difficult to follow as it did not include an abstract and the results were not separated into sections with titles.

Reviewer #3 (Remarks to the Author):

Review for Nature Communications Manuscript COMMS-16-15460-T

“Genetic coupling of viral-host gene expression presents migratory challenges in HIV therapies”

Kathrin Bohn-Wippert, Erin N. Tevonian, Melina R. Megaridis, Roy D. Dar

Summary: In this manuscript, the authors present evidence that the HIV LTR and CMV MIEP promoters are genetically coupled to host promoters with similar architecture. The authors first use a computational approach to identify a set of 285 host promoters with LTR/MIEP-like architecture. Following manual curation and extensive drug perturbation microarray analysis to identify correlated gene expression, the authors focus on two host genes, F3 and CXCR4, for which promoter architecture and drug perturbation analysis suggests that the expression of these genes is genetically coupled to viral gene expression in HIV and CMV infections. The authors further show that this genetic coupling between HIV and CXCR4 can impact CD4+ T-cell migration of HIV-infected cells. This coupling could lead to reactivation of HIV in niches where HAART cannot penetrate, and can result in new, unwanted infections. Thus, genetic coupling of HIV and CXCR4 has important implications for the design of “shock and kill” anti-latency therapies.

Overall Critique: The authors put forth a very interesting idea of genetic coupling between a host and pathogen based on promoter architecture. The implications of this work are of great interest to the HIV field in general, and more specifically to the HIV “Cure” researchers. Furthermore, with HAART, HIV+ individuals live for very long periods, and many are now dealing with other diseases associated with aging. The authors note that certain cancer therapy could inadvertently induce migration and reactivation of the latent HIV+ CD4+ T-cell pool. While this manuscript is very interesting, there are several points at which it could be improved for the broader Nature Communications readership:

1. The authors note in the Methods section that ~285 promoters had at least one NFkB, Sp1, and TBP site. For completeness, the list of genes that met the automated search cutoff should be included as supplementary information.

2. The authors also show 13 host promoters in Supp Figure 1. How were these chosen? Were these the promoters that also met the “manual curation” threshold? The authors note that the genes associated with these 13 promoters have important functions in viral pathogenesis. For the larger set of 285 genes that met computational cutoff, was there significant enrichment in specific

cellular processes, as suggested by this more refined set of 13 genes?

3. In Figure 1b, 2b,c and d, only F3 and CXCR4 are highlighted. Where do the other 11 genes (from Supp Figure 1) fall on the distribution in 2e?
4. What are the other genes that fall to the right of CXCR4 in Figure 2e, if they are not the 9 genes in Supp Fig 1? The authors should address why other genes with different promoter architecture might be highly correlated with HIV LTR expression. How many of the hits are probably due to non-transcriptional effects of drug treatment driving correlated expression? Based on the promoter architecture of the genes that fall to the right of CXCR4, how many correlations are due to pleiotropic effects of drug perturbations resulting in perceived correlation?
5. Many observations from immortalized cell lines do not hold in primary cell models. While reactivation cannot be reasonably tested in a primary CD4+ T-cell model, it would be powerful to show that LTR-GFP expression and F3/CXCR4 expression are correlated in primary human CD4+ T-cells using a few drugs from the panel used in Figure 2. Can the authors also demonstrate that primary human CD4+ T-cells migrate similarly to data in Figure 5b in response to different "shock and kill" combinations.
6. Figure 5a doesn't add a significant amount of understanding beyond what 5b accomplishes.
7. The sentence in lines 107-109 of the main text needs to be revised for clarity.
8. Some discussion of why TNF treatment results in a smaller impact on cell migration for MIEP and LTR vs naïve Jurkats would be useful (Figure 4a and Supp Figure 3b). Can the authors also comment on why the basal level of cell migration is higher in MIEP and LTR Jurkats vs naïve Jurkats? Does infection create some sort of "on" state for NFkB-, Sp1-, and TBP-driven gene expression of CXCR4? The authors should quantify CXCR4 expression to determine if it is higher in LG and MIEP-cherry cells.
9. Since Tat expression decouples LTR/CXCR4 expression, and in fact reverses the impact on cell migration due to secondary interactions between Tat and CXCR4, the kinetics of reactivation vs migration are key to determining how important this phenomenon will be to "shock and kill" design. How far can cells travel in the first 48hrs after reactivation? The authors present volumetric migration data. Can they quantify cell speed? The data presented in Figure 4d might also address concern, but the implications should be strengthened in the text. The migrating cells have slightly higher rates of reactivation, meaning the cells that do migrate to HAART-inaccessible niches are slightly more likely to be actively producing virus. Perhaps there is more of a dichotomy at early points (less than 48hrs post treatment)? The data in Figures 4b and c suggests 24 or 36hrs might provide robust migration + reactivation, where migrating cells may have an even higher proportion of reactivating cells.

In summary, this manuscript raises important questions about therapy design by demonstrating genetic coupling between HIV and human promoters. With some revisions, the claims of the manuscript would be strengthened, and would be of great interest to the Nature Communications readership. This manuscript should be considered for publication provided these revisions are addressed.

Reviewers' comments:

Reviewer #1 (Remarks to the Author):

The authors present an interesting study exploring co-regulation of viral and human gene promoters. They compare transcription factor binding sites in HIV-1 to roughly 22,000 human gene promoters. They identify high similarity between HIV (and other viral promoters) to CXCR4, F3 and other human gene promoters. They subsequently show using drug perturbation microarrays that human CXCR4 and F3 promoters have similar co-expression with HIV promoter in cell lines. Reactivation of latent HIV promoter and CXCR4 migration was then found to be co-regulated using TNF stimuli in cell lines, with the interesting caveat that HIV Tat expression impairs cell migration likely due to Tat inhibition of CXCR4 (reported elsewhere). As the authors find a small increase in HIV reactivation in migrating versus non-migrating cell lines, they suggest reactivation drugs that also promote CXCR4-mediated migration could direct reactivated cells to tissue sites with poor penetration of anti-retroviral therapy (ART) and cause unwanted spreading infection. Although, as their study also shows that HIV Tat impairs cell migration, one could speculate that potent reactivation of latent HIV and Tat generation would prevent cell migration and avoid this issue. Finally, the authors identify HIV latency reversing agents (LRAs) that potently reactivate HIV but limit cell migration in a cell line, and thus may have more benefit in reactivating latent HIV in patients while avoiding spreading infection in tissue sites.

Overall, this is a well written, intriguing study unveiling a previously unknown coupling of viral and human gene promoters, where HIV-1 and CXCR4 promoter co-expression might impact cell migration during reactivation of latent HIV. This will be of interest not only to the HIV latency field but also the broader virus-host factor field given the initial findings reveal co-regulation of HIV or CMV promoter with other host genes. Further studies on HIV or CMV expression with these other host genes seem likely to follow.

However, the experiments exploring reactivation of latent HIV and CXCR4 migration were performed in cell lines using modified HIV constructs. These are not strong models of HIV latency in patients on ART, where latency commonly arises in resting CD4+ T-cells (not cancerous dividing cells) and HIV DNA has integrated into diverse human sites rather than a single integration site in many clonal cell lines used here.

We agree that primary CD4+ T-cells are more relevant than transformed cell lines and that polyclonal cell populations are closer to the diverse integration landscape of the virus as our previous studies have detailed the structure of episodic transcriptional bursts of the integrated LTR across the human genome (Dar 2012 PNAS 109:17454-9). In the original submission LTR-GFP and MIEP-mCherry polyclonal Jurkat populations were used in Figure 2e and 4a. Indeed, the use of clonal JLats or LTR-Tat-IRES-GFP does not cover hundreds or thousands of integration sites, but a consistent trend of decoupling migration and reactivation was observed when measuring several isoclones. We have now added new experiments consisting of polyclonal infected primary CD4+ T-cells harboring either LTR-GFP or the full-length HIV JLatd2GFP constructs presented in the new Figure 6 and show that the observed behaviors are conserved for polyclonal CD4+ T-cell populations. Below we expand on this observation to note that the results show that the promoter coupling is largely invariant to integration sites across the human genome.

We have added the following text in discussing Figure 6 regarding integration sites (line 239 in the results):

“Unlike the JLat clones, infected primary cell experiments represent a large range of the integration site landscape of the provirus. Both CXCR4-LTR co-expression and decoupling by full-length HIV are observed in the polyclonal primary cell experiments (Fig. 6).”

In the discussion we continue to state (line 276):

“Co-expression between the LTR and CXCR4 in polyclonal LTR-GFP cell populations shows that the promoter coupling is largely invariant to differences in episodic transcriptional activity of LTR integration across the human genome¹⁶.”

For broader interest to the HIV latency field and to have more influence upon thinking in the field, the key findings of co-regulation of HIV and CXCR4 promoters, and impact of LRAs on reactivation of latency and cell migration need to be performed in primary human resting CD4 T-cells that are either latently infected *in vitro* or from HIV individuals on ART *ex vivo*. While this will take time, especially if the lab is not equipped to perform such studies, this could be readily achieved by collaboration with multiple investigators currently performing such patient primary T-cell studies in the field. This additional work would help strengthen these latter findings of the paper by demonstrating that these results also apply to latently-infected, primary resting CD4 T-cells and thus are important for researchers developing new LRAs to consider.

We fully agree that demonstrating the findings in primary patient T-cells on ART *ex vivo* or by latently infecting primary human resting CD4+ T-cells *in vitro* would further emphasize the relevance of this study to the HIV latency and cure research fields.

To address this concern in the current revision we pursued migration experiments of latent infection of full-length HIV (JLatd2GFP) in resting primary CD4+ T-cells after 48 hours of treatment to find that mixed cell populations (infected and uninfected) of primary cells failed to recapitulate the results demonstrated by the purely latent JLat clonal populations (Supplementary Fig. 8, also inserted below). In this experiment, infection rate was increased by infecting active CD4+ T-cells. The unsorted JLatd2GFP population (~35% GFP+) which expresses Tat was not capable of decoupling increased migration with TNF treatment. For the preparation and isolation of latently infected cells, we followed a conventional protocol for *in vitro* latency (Lassen 2012 PLoS One 7:e30176). After using a GFP- or OFF-sort of an infected cell population for migration assays, we observed that even this population contains a very small subpopulation of reactivating cells. Additionally, the approach becomes more complicated when a specific number of latently infected cells are required for the migration experiment. For cell migration, 50k CD4+ T-cells/100µl are necessary in the upper transwell chamber which requires ~200k sorted cells for each drug treatment. At least ~200k cells were needed to ensure seeding of the 50k cells/100µl after all washing, centrifugation and pelleting steps which are necessary to replace the culture media with migration media. Since there are currently no existing biomarkers for latently infected cells available, it is impossible to sort the large and required purely latent population of cells needed to perform the migration measurements and carry out experiments similar to the Jurkat latency model in which all cells of the population are infected, latent, and can potentially reactivate (compared to a ~1-3% OFF subpopulation that may reactivate).

To account for this limitation we performed migration measurements on both LTR-GFP and JLatd2GFP for GFP+ and ON sorted populations of stimulated and infected primary CD4+ T-cells (Fig. 6, shown below), which would represent the behavior of fully reactivated and active T-cells after the transition from a resting latent cell state. The results show that migration of infected primary CD4+ T-cells is consistent with the migration shifts of TNF treated LTR-GFP and diverse drug treatments of JLat 15.4 in Figures 4 and 5. The results presented in Figure 6, and the main text are that infected primary cells change their migratory behavior depending on diverse drug treatments, and that Tat expressed in full-length HIV shows decreased migration with TNF treatment (Fig. 6d) or a decoupling compared to the LTR-GFP construct (Fig. 6b-c), consistent with the results seen in Jurkats (Figs. 4 and 5).

“Supplementary Figure 8: Unsorted and OFF sorted HIV infection of activated primary CD4+ T-cells is insufficient to detect decoupling of migration.

Activated primary CD4+ T-cells were infected with the JLatd2GFP construct using spinoculation. For the unsorted cell population, the decoupling of increased migration post-48h TNF treatment is no longer discernible due to the dominating number of uninfected CD4+ T-cells which obscure the behavior of HIV infected cells (**upper left**). Mean expression of the total population remains fairly constant with uninfected cells dominating fluorescence levels (**lower left**). PMA treatment shows quenching of CXCR4-mediated migration consistent with Figures 5 and 6, suggesting off-target migratory suppression by PMA seen in Supplementary Figure 8. Similarly, OFF, GFP- sorted CD4+ T-cells infected with JLatd2GFP show similar behavior of a strong increase in migration with 48h TNF treatment, and any potential latent minority of cells are unable to decouple the migration increase caused by drug treatment (**upper and lower right**). These results present a challenge in running migration experiments for non-pure latent OFF-sorted primary cell populations, typically consisting of a very small minority of latent cells selected by sorting the GFP- infected population and which is dominated by uninfected cell behavior. For the 2 donors above, infection of CD4+ T-cells was quantified at ~35% GFP+ cells.”

“Figure 6: LTR-CXCR4 coupling and migratory behaviors of HIV-infected primary CD4⁺ T-cells. (a) Primary CD4⁺ T-cells are isolated from fresh human whole blood and stimulated with anti-CD3/anti-CD28 antibodies on the same day. After 3d of stimulation, the cells are infected with concentrated lentivirus containing either HIV-1 LTR-GFP or the full-length HIV-1 (JLatd2GFP) for 2h by spinoculation. Next, CD4⁺ T-cells are sorted for GFP positive cells one day after infection or unsorted cells are used one to two days after infection. Sorted or unsorted CD4⁺ T-cells are then seeded into V-bottom plates and treated with drugs for 2d. Migration assays were carried out using a 96-well transwell chamber and expression of GFP was measured using flow cytometry. (b) 48h TNF treatment of unsorted LTR-GFP infected CD4⁺ T-cells from three separate donors shows a correlated increase of both migration and mean fluorescence. (c) GFP⁺ sorted LTR-GFP infected CD4⁺ T-cells show co-expression of LTR and CXCR4-migration after 48h of TNF treatment consistent with the Jurkat cells. (d) Sorted CD4⁺ T-cells infected with JLatd2GFP display a decoupling from increased migration after 48h of TNF treatment. The decoupling of migration is relative to the primary CD4⁺ T-cells infected with only LTR-GFP in panels b-c. In addition, cells treated with SAHA reduce migration consistent with the HDACi effect in the Jurkat model in Figure 5. PMA also displays a strong reduction of migration as well as eliminating migration when combined with Ionomycin (Iono) similar to Figure 5. For the second donor (donor 6), combination of PMA and Iono reduces enhanced Iono migration and SAHA reduces TNF migration. Measurements were performed on two separate donors.”

There are also some further technical aspects and questions to consider:

1. In Fig 4, please clarify whether the average results in each panel derive from one independent experiment performed in duplicate or triplicate, or from 2-3 independent experiments performed on separate days.

We have now clarified in the main text that besides measurements performed in duplicate, experiments were also performed independently on different days and then averages of 1-3 independent duplicates were taken to create the plots. In all cases cell flasks were grown in parallel to allow reduced measurement error and day-to-day variability. For this reason, data represents independent measurements on different days. In a few individual cases a third measurement was performed on a separate day.

The independent measurements have now been clarified (line 525 in the methods):

“Except for sorted and infected JLatd2G primary CD4+ T-cells, all migration experiments were carried out in duplicate or triplicate on separate days, and the average values are plotted.”

Error bars should also be added to data points in Fig. 4a-c so it is clear results are consistent across independent experiments when error is taken into account.

Individual standard error bars have now been added to each data point in Figure 4a-c. For discussion on the consistency please see the next two responses.

In Fig 4d, are the small reactivation increases in migrating versus non-migrating cells maintained in 3 independent experiments?

Yes. Three independent experiments were performed and carried out on different days for all of the measurements represented in the two panels of Figure 4d (JLats and latent LTIGs, now updated to Fig. 4e and expanded in Supplementary Fig. 6).

In Fig 4c, are the small migration changes in A2 clone at 24hr or A7 clone at 48hr still seen if the error is taken into account?

We appreciate this observation as it further clarifies the general purpose of Figure 4 and the decoupling caused by Tat expressing vectors. With error bars several JLat clones in Figure 4b and LTIG clones in Figure 4c can now be looked at as having comparable migration after TNF treatment (although even with some bar overlap the trends shift downwards). We note that comparison for reduced migration is done with the untreated controls in Figure 4b and 4c. Even if migration remains constant with TNF addition, this constitutes a decoupling as the increased migration seen in CXCR4-LTR promoter co-expression of Figure 4a is eliminated. This indicates that Tat is mediating reduced migration. To emphasize this point we have now included an additional bar plot summary in an updated Figure 4d, shown below, presenting the 24h TNF treated normalized migration of LTR-GFP, JLat, and LTIG populations +/-TNF, i.e. $\text{Migration}_{+\text{TNF}}/\text{Migration}_{-\text{TNF}}$, to emphasize that the full-length and minimal-HIV Tat constructs decouple TNF-increased migration observed with LTR-GFP cells (From ~1.3 to ~0.75). We have clarified this in lines 152 and 159 of the results:

“Counter to the minimal promoter-system results (Fig. 4a), all clones demonstrate a decoupling between migration and reactivation with either equal or lower levels of migration (Fig. 4b and Supplementary Fig. 5).”

“Although still migrating, a decoupling of correlated migration and reactivation was preserved (Fig. 4d), suggesting that full-length HIV has evolved the capacity to tune host-cell migration using two coordinated mechanisms: co-expression with the genetically coupled CXCR4 promoter, and protein interactions at the cell surface between the viral product Tat inhibiting CXCR4 for decreased cell motility.”

Figure 4d caption: “Normalized migration of TNF treatment by untreated migration post-24h shows a coupled phenotype with LTR-GFP and a decoupling caused by Tat reactivation of JLat and LTIG latent clones.”

Also for Fig 4b, all time points should be shown with error bars for both untreated and TNF-treated cells on the same graph given the migration of untreated cells vary over time, especially for clone 9.2 in Supp Fig 4. With all time points on the same graph, clone 9.2 may show little difference between TNF treated and untreated cells even at 48hr given the migration of untreated cells is <100K.

Additionally, are declines in TNF-treated clones 15.4 and 6.3 still present across time points when error is accounted for?

We originally had Figure 4b as suggested by the reviewer but decided it was too cluttered as the various untreated points all overlap on the y-axis. For this reason, we now present the comparison requested (both migration vs. reactivation, and bar plots comparing the migration) for the full dataset of Figure 4b in a new Supplementary Figure 5. These graphs show individual comparisons at each time point and either a conserved migration or inverse relationship with JLat clone reactivation at 12, 24, 36, and 48 hours +/-TNF treatment. We have also refined the observation and description of the decoupling of CXCR4-LTR co-expression to describe the trend of migration as either “conserved” or “reduced”. We specify this on line 152 of the results section:

“Counter to the minimal promoter-system results (Fig. 4a), all clones demonstrate a decoupling between migration and reactivation with either equal or lower levels of migration (Fig. 4b and Supplementary Fig. 5).

Finally in the Fig 4c legend, the authors did not specifically show that Tat was stripping CXCR4 off the cell membrane so this cannot be concluded. The authors should either show this data using CXCR4 staining and FACS analysis, or reword this sentence to say that the results were "consistent with" Tat-related deceleration of migration by inhibiting CXCR4 on the cell membrane etc.

We have now updated the caption of Figure 4c to emphasize that the reduction of TNF-induced migration is “consistent with” Tat-mediated inhibition of CXCR4 on the cell membrane.

The last sentence of the caption of Figure 4c now reads as follows:

“This result is consistent with Tat decoupling of migration by inhibiting CXCR4 on the cell membrane for an inverse or constant migration-reactivation relationship.”

2. If Tat prevents CXCR4-mediated cell migration (Fig 4b-c), why are the cells with higher reactivation in Fig 4d the migrating cells rather than non-migrating cells? Wouldn't the Tat argument predict that as HIV reactivation increases, HIV Tat increases, thus CXCR4 migration declines and non-migrating cells should have more reactivation? This should be addressed in the manuscript.

This is an interesting and valid observation. We formulated several hypotheses that explain the consistent difference in reactivation levels: (1) The ratio of CXCR4 and Tat levels is different in migrating from non-migrating cells, (2) Tat transactivation, even in a clonal population is heterogeneous and has transients continuously starting from the inactive state when reactivated (leading to a delayed coupling at steady state levels) (Weinberger 2005 Cell 122:169-82; Weinberger, Dar, Simpson 2008 Nature Genetics 40:466-470), and (3) Migrating cells have a higher reactivation rate than non-migrating cells (as proposed in the main text).

To address these related explanations, we have now included a new section in the Supplementary Information (shown below) where we have calculated the reactivation rates and total predicted reactivation levels after 3 hours of migration. Taking into account the reactivation level of the total population (Fig. 4b), reactivation at the start of migration, and accumulated migration at the end of migration, the *null hypothesis* assumes that migrating and non-migrating cells reactivate at the same rate. If the *null hypothesis* holds then our calculations would show that the percentage of reactivated cells for both non-migrating and migrating cells, before and after the migration experiment would be equal. Instead, for the migrating cells we observe a higher percent of reactivation, suggesting that the *null hypothesis* is negated and cannot explain the increase in reactivation, supporting the conclusion that migrating cells have a higher reactivation rate than non-migrating cells.

The section from the Supplementary Information has been included on the next page:

Reactivation rate from latency of migrating cells is higher than non-migrating cells

Consistent results in Figure 4e suggest that reactivation is higher for migrating than non-migrating cells for latent full-length JLat and minimal LTIG constructs.

To approximate if this is true, the following calculation was performed to disprove the *null hypothesis* that reactivation rates are equal between migrating and non-migrating cells.

1. Using average reactivation values for three JLat clones in Figure 4b, reactivation is linear over 12-48h TNF treatment and the rate was calculated using a linear fit (**upper panel, figure at right**). Here reactivation was measured with flow cytometry before the migration experiment.

2. (**lower panel**) Measured reactivation or %ON of JLats post-48h TNF treatment was used for both migrating and non-migrating cells and compared to an expected and calculated %ON taking into account the total number of migrated cells in each experiment and the constant reactivation rates calculated in the previous step for the cells that migrate during the 3h migration experiment.

Formulas used for calculation:

Calculated %ON post-3h migration = %ON at start of migration + % of cells that turn on in the migrated population

where,

%ON at start of migration = (%ON for Non-migrating population treated for 48h TNF measured at the end of the 3h of migration – % of cells that turned on in the non-migrating population during the 3h experiment)

3. The results show that the measured reactivation of migrating cells is higher than the predicted calculation using a constant reactivation rate ($y=x$, **lower panel**), therefore negating the *null hypothesis*. The non-migrating reactivation matches the calculated value, landing on $y=x$, and represents the constant reactivation rate assumption.

Figure: (upper) Calculation of reactivation rates for each of three JLat clones. **(lower)** Comparison of reactivated cells versus their expected amount of reactivation using a constant reactivation rate for migrating and non-migrating populations (*Null hypothesis*, blue line). Migrating cells land above the line with increased reactivation than expected while the non-migrating population represents the constant rate assumption during the migration experiment.

This result has been included when describing Figure 4e in the main text (line 172 of the results):

“Strikingly, despite a consistent trend of reduced migration for HIV expressing cells^{38,39}, JLat and minimal Tat clones treated with TNF for 24h or 48h reveal that sub-populations of migrating cells reactivate at higher rates compared to stationary (or non-migrating) cells (Fig. 4e, Supplementary Fig. 6, and Supplementary Information).”

3. For Fig 5, was the toxicity of the various drugs tested on J-Lat 15.4 cell line eg. MTS metabolic assay or Live/Dead flow stain? This should be tested and included as supplementary data as some reductions in migration might arise from toxic effects of the drugs rather than direct effects on CXCR4 function.

To determine if drug treatment causes cell death, we have performed Propidium Iodide (PI) staining for primary CD4+ T-cell populations showing minimal drug toxicity after 48h treatments. This is now included in the Supplementary Information and referred to in the main text as Supplementary Figure 7.

“Supplementary Figure 7: Propidium iodide staining of treated activated and uninfected primary CD4+ T-cell populations shows minimal drug toxicity post-48h. Cell death staining post-48h drug treatments on activated and uninfected primary CD4+ T-cells was performed to confirm that cell death was minimal for treatments in Figures 4-6. Propidium iodide (PI) staining, using drug concentrations listed in Supplementary Table 2, shows that within the live gated cell population of the flow cytometer (left), all cells remain below a PI+ value of 2% (right) indicating that a large majority of cells in the live gate are indeed live. This is consistent with the drug concentrations selected based on previous studies⁵⁻¹³ as well as previous PI staining of treated cells³.”

In addition the selected drug concentrations tested have been used in previous reports (Spina 2013 PLoS pathogens 9:e1003834; Kubarek 2007 FEBS Lett 581:1441-8; Kubarek 2009 Biomed Pharmacother 63:586-91; Huang 2014 PLoS One 9:e115249; Hezareh 2004 Antivir Chem Chemother 15:207-22; Crazzolara 2002 Br J Haematol 119:965-9; Han 2001 J Clin Invest 108:425-35; Dar 2014 Science; Mandawat 2010 Blood 116:5306-15; Clift 2014 Mol Pharmacol 85:542-52, Ierano 2013 Cancer Biol 14:175-83). An additional PI stain of diverse drugs previously showed that all cells outside the FSC-SSC LIVE gate is a strong indicator of cell death (Dar 2014 Science).

In the present study, the cell populations measured by FACS prior to migration displayed high %LIVE in the live gate compared to the untreated control. Migration and fluorescence assays were only carried out for live cell populations by spinning down pretreated cells and aspirating or removal of dead cells in the supernatant before

each migration experiment. This suggests that a majority of cells in the migration assays were alive. The same population of migrated cells was used to measure the fluorescence after 48h drug treatment. For Figure 5, the lowest treatment displayed ~21.5% of cells in the live gate (JQ1+TNF) which would translate into about one third of the untreated cell population, with a majority of treatments showing only slight reduction compared to untreated cells (Supplementary Table 2). We have now included an additional column in Supplementary Table 2 which reports the percentage of cells in the live gate to support that the differences quantified in CXCR4 migration are not due to cell toxicity of the treatments.

4. The authors should discuss how results in Fig 5 relate to Fig 4 given there are many examples in Fig 5 where more potent reactivation by LRA combinations correspond with impaired migration, complementing Fig 4b-c results.

As suggested, we have now elaborated the discussion section and include the following text in line 288 of the discussion section to clarify and strengthen the relationship between Figures 4 and 5:

“A subset of compounds combined with TNF provided the largest reactivation with reduced migration. Although consistent with a general trend of decreased migration with increasing reactivation from Figure 4b and 4c, sole treatment and quenching of migration by SAHA and PMA suggests a consistency with their known internalization of CXCR4 at the membrane and a complete reduction of TNF migration levels, i.e. membrane modification dominates intracellular NFκB activation (Fig. 5 and 6d). Treatment with a combination of JQ1 and TNF, both migrating compounds when treated independently, results in high reactivation and a strong reduction of migration, while migrating compounds E2 and Tam conserve both migration and reactivation levels when combined with TNF compared to TNF alone (Fig. 5).”

5. The manuscript is succinct and discusses the results in context with relevant literature. Although the statement and presumption that “reactivated cells die through apoptosis over a 1-2 day period” (line 121, Fig 3 model) over simplifies a more complex matter. The 2 cited references estimate a 1-2 day life span of productively infected cells in chronically infected patients after initiation of 1 or more antiretroviral drugs. However these references do not specifically show that productively-infected cells die by apoptosis. Secondly, these papers do not estimate the life span of reactivated, latently infected cells from patients on longer term ART, which may vary according to the reactivation stimulus and potency eg. T-cell activation versus less potent LRAs. How well patient cytotoxic T-cells recognise reactivated cells will also influence the life span of reactivated cells, especially as cytotoxic T-cell function is compromised in patients that initiate ART during chronic infection due to CTL escape mutations in latent viruses (Deng 2015 Nature 517:381-5). Certain LRAs also impair cytotoxic T-cell mediated killing (Shan 2012 Immunity 36:491-501; Jones 2011 Plos Path 10:e1004287; Jones 2016 Plos Path 12:e1005545). Clinical trials with HDACis or disulfiram also failed to decrease the HIV reservoir in patients on ART despite reactivating HIV RNA (Rasmussen 2016 Curr Opin HIV AIDS 11:394-401) again questioning the link between reactivation and cell death. Therefore the life span of reactivated cells and how they die is unclear, and this part of the paper stated as fact needs to be modified for accuracy.

We thank the reviewer for insight into this important issue that has seen recent advancements noted from HIV research. We recognize the recent reports in the literature presented which support that complete cell death is unclear and not an immediate outcome from reactivation of latently infected cells. Prolonged viability of reactivated cells only reinforces and supports the migratory challenge presented in the current study and adds additional concerns to the accumulating studies showing LRA ineffectiveness at T-cell killing. We appreciate

these in-depth comments and instead of stating fact regarding cell death we have now elaborated upon these amassing challenges from reported “shock and kill” research in the discussion and how the migratory challenge only adds to these reported challenges (line 306 of the discussion):

“Customized therapeutic strategies are further needed as recent studies contribute to an emerging picture of prolonged cell viability impeding eradication of HIV infected cells. Reports using mathematical models to provide an understanding of in vivo replication kinetics of HIV estimate a generation time of approximately 1-2 days for productively infected cells^{25,26}. These theoretical models do not directly measure apoptotic death for productively-infected cells or latency in patients on long term HAART. Recent reports further extend the possibility of prolonged cell death through experimental investigations by showing that: (1) HIV-1 can acquire mutations to evade cytolytic T lymphocytes required for clearance of the latent reservoir⁵⁵, (2) latency reversal agents impair cytotoxic T-cell mediated killing⁵⁶⁻⁵⁸, and (3) clinical trials with histone deacetylase inhibitors (HDACis) and disulfiram fail to deplete the latent HIV reservoir in patients on HAART despite HIV reactivation⁵⁹. Collectively, these studies provide evidence that T-cell killing of reactivated latent cells is difficult, and may not occur at all. Migratory challenges to HIV “shock and kill” presented in this study are only further exacerbated by this lack of T-cell death by providing a longer migration window accompanied by viral shedding of reactivated cells (Fig. 3b).”

Overall, this manuscript is an intriguing study with novel findings of viral-host promoter co-regulation likely to be of interest to the HIV latency and broader HIV-host factor fields. However, results from cell lines showing co-regulation of HIV and CXCR4 promoter, and the impact of LRAs on HIV reactivation and cell migration, would be more convincing if key experiments were performed in latently infected primary human T-cells that better model HIV latency in patients.

We appreciate the support of the reviewer for the novel findings presented in this study and its interest to the HIV latency and HIV-host factor fields. We believe that the consistency of the observed coupling and decoupling of LTR-CXCR4 for minimal LTR-GFP and full-length HIV (JLatd2GFP) infected primary cells of multiple donors now presented in Figure 6, along with migratory behaviors with combination drug treatments, provide a more convincing picture of the co-regulation in infected primary human T-cells.

Minor comments:

1. Fig. 1: In panel A, it would be helpful to add the gene name to the list on the left side for clarity. The magnified human gene promoter does not match the item in the gene promoter list on the left, which has 2 NFkB, 3 SP1 and 1 TATA box. Shouldn't this match? Please add what the boxes signify to the figure legend: NFkB blue, SP1 pink, TATA red, white box absent site. In panel B, what is the brown circle in the SIV LTR? Please define F3/TF in the legend.

We have now improved Figure 1a for clarity and consistency with the promoter list of human coding genes. We have also emphasized that Figure 1a is a schematic of the approach taken for the promoter search and does not represent real human promoters such as in Figure 1b. We have also clarified the figure legend regarding the color coding of the *cis* elements. The brown circle in Figure 1b represents Simian-Factor-1 or SF1, an AP-1 related transcription factor in Monkeys and reported in the SIV LTR promoter at ~ -135bp (Murakami and Li, Biomed Sci 1996). We have noted this in the legend for Figure 1 and expanded the main text to emphasize viral-host evolution and conservation of the genetic coupling architecture among different hosts (line 111 of the results).

This provides an additional indicator of how highly conserved the promoter coupling arrangement is as CMV along with all herpesviruses have co-evolved with their hosts for millions of years to establish lifelong infections into hosts which HIV also co-infects (McGeoch 2000 J Virol 74:10401-6).

The updated Figure 1 legend includes all of the above concerns and now reads as follows:

“Logic-based search of human coding gene promoters for similarity to viral core promoters. The list of human promoters indicates a logic series of cis binding elements (boxes) for each promoter (rows) comprised of diverse patterns of TATA box (red), SP1 (pink), and NFkB (blue) binding sites. Conserved cis binding sites in a virus (NFkB- blue triangles, SP1- pink diamonds, and TATA- red squares) are searched for in all human coding genes. HIV and CMV promoters are curated and compared within +/- 250 bp from the transcription start site (TSS) of human promoters. Promoters are clustered regardless of number, arrangement, or distance of sites from the TSS. Additional transcription factor binding site types are ignored to de-constrain results of the search. (b) The genome-wide promoter search results in ~366 gene promoters with binding sites for TBP (TATA, red squares), SP1 (pink diamonds), and NFkB (blue triangles), and F3/TF, CXCR4, and HIV-1 LTR show the highest similarity in cis regulatory arrangement. Upstream existence of AP1 sites (yellow circles) and downstream nucleosome occupancy (gray circles), binding of P-TEFb⁶⁸, BRD4⁶⁸, and CPG islands (purple and navy rectangle regions) from the TSS are also consistent between promoters and curated after the initial genome-wide search. The brown circle in the SIV LTR promoter represents Simian-Factor-1 or SF1, an AP-1 related transcription factor in Monkeys reported in SIV around -135 to -131bp⁷⁵.”

Line 111 of the results section reads:

“Promoter coupling between distant viral families (RNA and DNA) that co-infect their host, including multiple hosts throughout their evolution (SIV and HIV), demonstrates that genetic coupling may elucidate a deeper understanding of co-evolution of multiple viruses and their hosts¹⁸ along with their co-infection dynamics.”

2. Supplementary Fig 1: For clarity, the same symbols and colours for SP1 and NF-KB should be used in both Fig 1 and Supp Fig 1. HV-1 LTR, HIV-2 LTR and CXCR4 should be included in Supp Fig 1 for easy comparison.

Supplementary Figure 1 has now been updated with the same symbol and color convention as Figure 1. We have also highlighted CXCR4 and F3 which were previously in the list in Supplementary Figure 1 to ease comparison.

3. Fig 5 & Supp Table 1: Why is Prostratin alone in the “Migration and Reactivation” classification when it has less migration (48550 cells/ml) than JQ1+TNF (50350 cells/ml) that was put in the reactivation only classification? This should be consistent. In Supp Table 1, please add untreated cells at the top of the table for comparison so it is clear which drugs decrease migration compared to untreated cells. Instead of the phenotype column, it would be more meaningful to make this column “Transcription activities” and list the known activities of these drugs modulating transcription eg. Prostratin - activates PKC and NF-KB. When the untreated cells are added to the table, the migration and reactivation results speak for themselves without needing the phenotype column.

In reevaluating Figure 5 we have redrawn the behavior classification to better describe the observed data. Any points that deviate from strict adherence to the migration or reactivation axes have now been grouped into the “Migration and Reactivation Cocktail” classification. This means, drugs with reactivation rates lower than 0.10% are considered as only migratory drugs and drugs with migration rates lower than 10000 cells/ml are classified as

only reactivating drugs. All other measurements are classified as “Migration and Reactivation Cocktails” and to simplify the figure the background shading regions have now been removed. The new Figure 5 has been included below:

Regarding Supplementary Table 2, we have changed the “phenotype” column to a more general description “bioactivities concerning migration and reactivation” in order to address a broad range of mechanisms without relating to specific transcription or translation pathways of the cell. We have updated all known bioactivities of the drugs in this column along with supporting references.

4. Lines 126-129: Please provide references for CXCL12 being enriched in the tissue sites listed and attracting cells that overexpress CXCR4.

We have cited the following references supporting overexpression of CXCL12 mRNA and protein in the tissue sites listed in lines 130-133 of the results:

- Wilhelm et al., Mass-spectrometry-based draft of the human proteome. Nature. 2014 May 29;509(7502):582-7
- Su AI, Wiltshire T, Batalov S, Lapp H, Ching KA, Block D, Zhang J, Soden R, Hayakawa M, Kreiman G, Cooke MP, Walker JR, Hogenesch JB (2004) A gene atlas of the mouse and human protein-encoding transcriptomes. Proc Natl Acad Sci U S A. 101(16):6062-7.
- Wu C, Jin X, Tsueng G, Afrasiabi C, and Su AI (2016) BioGPS: building your own mash-up of gene annotations and expression profiles. Nucl. Acids Res. 44(D1): D313-D316. (*Database Issue*)
- GeneCards: a novel functional genomics compendium with automated data mining and query reformulation support. Bioinformatics. 1998;8:656–664

We have also cited a review of CXCR4 physiological and pathological functions:

- Domanska, U. M., R. C. Kruijzinga, W. B. Nagengast, H. Timmer-Bosscha, G. Huls, E. G. de Vries and A. M. Walenkamp (2013). "A review on CXCR4/CXCL12 axis in oncology: no place to hide." Eur J Cancer 49(1):

Reviewer #2 (Remarks to the Author):

A. Bohn-Wippert et al have conducted an investigation into the similarities between human protein coding genes and the HIV-1 LTR promoter. They found that human coagulation factor 3 (F3) and the chemokine receptor CXCR4 exhibit the greatest promoter similarity to LTR. Microarray analysis using exogenous drug perturbations revealed that F3 and CXCR4 exhibited a correlated co-expression with the LTR. However, the dominant co-receptor CCR5 exhibited an uncorrelated co-expression with F3 and the LTR. Using a flow cytometry method Bohn-Wippert and colleagues also found a strong co-relation of expression between the HIV and CMV promoters which may indicate co-evolution of viral families able to cause asymptomatic latent infections. Some very elegant migration assays are used to determine if HIV latency reversal agents reactivate migrating cells.

C. The format of this manuscript was a little difficult to follow without section headings being included or numbers on the figures.

The manuscript has now been reformatted to adhere to *Nature Communications* guidelines with section headings, and the final uploaded figures will be merged in the correct order using the file upload system of the journal. We hope this makes the paper easier to follow.

The meaning of one sentence is unclear. This sentence begins on line 106: “The finding of promoter coupling between distant viral families able to cause asymptomatic latent infections,.... ends on line 109

This sentence was unclear to more than one reviewer. We have now reworded the sentence in the result section (line 111):

“Promoter coupling between distant viral families (RNA and DNA) that co-infect their host, including multiple hosts throughout their evolution (SIV and HIV), demonstrates that genetic coupling may elucidate a deeper understanding of co-evolution of multiple viruses and their hosts¹⁸ along with their co-infection dynamics.”

More description of the microarray test and analyses is needed in the methods section.

Detailed description of the microarray tests used are readily available from the original paper from Lamb et al. (Lamb 2006 Science 313:1929-35). Although we referenced this paper, we have now added details on the cell types, perturbagens, and genome-wide microarray platform used in the study in the methods section.

The section “Perturbation microarrays and drug screening” in the methods (lines 441-472), supplies an expanded description of the genome-wide microarray screening performed by Lamb et al., 2006:

“Genome-wide microarrays accounting for 885 treatments of an HL60 cell line were used to quantify co-expression between host and viral promoters¹. Additional details on the microarrays can be found in the original paper from Lamb et al. (2006)¹. In brief¹, genome-wide microarrays were performed on three cell types (HL60, MCF7, and PC3) exposed to thousands of diverse perturbagens. Total RNA was isolated and synthesis of cRNA target, its hybridization to microarrays and scanning of those arrays was performed using Affymetrix GeneChip

products and reagents. Total RNA from development batches was processed manually using HG-U133A cartridge arrays (part number 510681). Total RNA from production batches was amplified and labeled using the GeneChip Array Station (GCAS) and hybridized to HT_HG-U133A (early access version; part number 520276) High-Throughput Arrays (HTA) which were scanned using the HT Scanner. The data used in this publication is deposited in NCBI's Gene Expression Omnibus (GEO, <http://www.ncbi.nlm.nih.gov/geo/>) and is accessible through GEO series accession number GSE5258. Data are also available for download at <http://portals.broadinstitute.org/cmap/>."

D. The findings of control perturbations for the microarray assay are not described in the methods section. In addition, the effects of the cell lines upon the microarray results should be described. Overall more description of the microarray assay is required.

The referenced paper by Lamb et al. (Lamb 2006 Science 313:1929-35), describes all the vehicle controls performed on the microarray assay. We have now included a few brief descriptors of the assay performed for the reader (see previous response). For cell line comparison, we have added the following in the results section (line 91):

"Of the three cell lines measured in the original study where the genome-wide microarrays were performed, the HL60 neutrophil cell line was chosen as the closest to the Jurkat T-Lymphocyte cell line for evaluating co-expression (compared to MCF7 breast cancer and PC3 prostate cancer cell lines in the microarray dataset)."

E. The authors do not address the issue as to why the structurally related chemokine receptors, CXCR4 and CCR5, have different correlations of co-expression with F3 and the LTR. The authors should provide the different expression levels of CCR5 and CXCR4 during the exogenous drug perturbations and an explanation as to why the CCR5 receptor is not co-expressed with the LTR. Due to the fact that all primary HIV strains can use the CCR5 receptor to infect cells, the impact of their study showing CXCR4 and LTR promoters exhibit correlated co-expression patterns is a little unclear.

We thank the reviewer for the comment and highlighting the need for clarification as through discussions many have found it particularly interesting why CXCR4 and not CCR5 shows co-regulation with the LTR. We have now added a lower panel to Supplementary Figure 2 to show the significant promoter structure dissimilarity between CCR5 and CXCR4 to complement the co-expression profiles that were previously presented showing no co-expression between CCR5 and genetically coupled promoters presented in Figure 1b (CCR5 and F3 as well as CCR5 and LTR; calculated using expression levels available in the genome-wide microarray datasets described above). To emphasize this divergence as to which tropism the viral promoter has assimilated, we have expanded the discussion section to include the dual role of CXCR4 in the HIV-host relationship in both late-stage viral infection at the membrane, as well as co-expression and coordination of viral expression with cell migration.

Line 105 of the results now reads as follows:

"In contrast to CXCR4, a co-receptor in HIV-1 infection, the dominant co-receptor CCR5 having only a single NFkB binding site and high dissimilarity in promoter architecture exhibited uncorrelated expression with both F3 and the LTR (Supplementary Fig. 2)."

And Line 270 of the discussion:

“CXCR4 is typically studied for its role as a mediator of HIV viral entry at later stages of infection⁵³. Despite genetic coupling and co-expression observed between HIV and CXCR4, the dominant CCR5 tropism of HIV is completely uncoupled genetically and uncorrelated in response to drug perturbations (Supplementary Fig. 2). This results in a dual-role of HIV-CXCR4 with membrane interactions and promoter co-regulation for both infection and migration of the host-cell.”

It is unclear what the genetic coupling of HIV and CMV promoters during diverse drug perturbations really mean for latent viral infections. A more definitive description of how co-expression between HIV and CMV promoters provides a deeper understanding of co-infection and latent viral infection is needed. The particular drug perturbations which revealed this co-expression of viral promoters should be described in more detail.

We agree with the reviewer that the importance of these findings should be elaborated upon in the revision. We have now added the following text in the results and discussion sections emphasizing the potential importance of these findings for promoter coupling in understanding co-evolution between multiple viruses and CMV co-infection in AIDS patients:

Line 111 of the results section (similar to the response to “C” above):

“Promoter coupling between distant viral families (RNA and DNA) that co-infect their host, including multiple hosts throughout their evolution (SIV and HIV), demonstrates that genetic coupling may elucidate a deeper understanding of co-evolution of multiple viruses and their hosts¹⁸ along with their co-infection dynamics.”

Line 324 of the discussion section:

“Computational logic-based promoter searches for similarity in gene regulatory arrangements can elucidate additional genetic coupling and guide drug therapies. An unexpected genetic coupling of HIV-LTR and CMV-MIEP promoters was uncovered between two distinct and co-evolving viruses (Fig. 2f and Supplementary Fig. 3). CMV has been prominent in patients infected with HIV-1 and the relationship between both viruses is important for persistence under current HAART treatment⁶⁰. LTR-MIEP coupling not only suggests that viral-host coupling can include co-evolution of multiple viruses, but may elucidate a more comprehensive viral-host network and treatment for common co-infection in patients⁶⁰.

Moreover, human F3 (tissue factor), another coupled promoter, is involved in inflammation, cellular homing, blood coagulation, hypercoagulation, and thrombosis (Figs. 1 and 2). Thromboembolic complication has been observed in HIV+ and AIDS patients⁶¹⁻⁶⁴ and along with inflammation is elevated in HIV infected patients^{64,65} with mixed reports regarding those under HAART^{62,64}. With a coagulation-inflammation-thrombosis circuit⁶⁶ affected by both HIV and CMV infection, the findings in this study may provide mechanistic insight for the elevated risks for thrombosis in patients and the design of novel treatments for hypercoagulation and inflammation.”

The particular drug perturbations used for co-expression of Figure 2f have now been specified in the main text (Line 108):

“Strong co-expression between HIV and CMV promoters was shown using flow cytometry of polyclonal cell populations harboring LTR-GFP or MIEP-mCherry after 24h treatment with Trichostatin A (TSA), Prostratin, and tumor necrosis factor alpha (TNF) ($R^2 = 0.85$, Fig. 2f).”

F. The authors found that polyclonal LTR-GFP Jurkat cells treated with the latency reversing agent SAHA alone or in combination with TNF or PMA resulted in an inverse reactivation-migration relationship indicating cells reactivated by these compounds do not seem to migrate. This is an important finding but the authors should have included more potent latency reversal agents such as panobinostat and/or romidepsin in their analysis of migration versus reactivation analyses.

As suggested, we have carried out 48h treatments of the two anticancer drugs panobinostat (Pano) and romidepsin (Rom) using JLat 15.4. The experiments resulted in less potent reactivation than SAHA for this specific JLat clone (more on this and dose response shown below) while displaying a migrating cell phenotype. The results for Pano and Rom have been added to Figure 5 of the main text (figure also included below). The used drug concentrations and cell viability were added to Supplementary Table 2 and also described in the methods under “Reagents for migration and reactivation assays”.

In addition, we performed dose response curves for both drugs using 24h and 48h of treatment on JLat 15.4 from Figure 5. The results for Rom and Pano are shown in the figure below. Although, a 24h treatment shows higher reactivation levels, migration of the treated cell, and increased cell viability compared to 48h treatments, this result is not related to the 48h drug treatment in Figure 5 and therefore was not used. The result does confirm that the 24h treatment reactivates JLat 15.4 more than 48h, and that these compounds consistently migrate cells.

We used drug concentrations below 80nM since higher concentrations cause massive cell death (Jones 2014 PLoS Pathog 10:e1004287). The occurrence of cell death is also supported using extended treatment times. We will also note that Pano and Rom can show extremely varied reactivation for different JLats (Bouchat 2016 EMBO Mol. Med. 8:117-138) and that in certain *ex vivo* studies using blood from HIV+ patients treated with ART, Pano alone did not induce significant viral reactivation (Laird 2015 J Clin Invest 5:1901-1912).

The finding of migration with these two potent LRAs is still important and shows examples of HDACis that do not completely reduce CXCR4 migration.

Figure: Dose response curves for 24h and 48h treatment with romidepsin and panobinostat on JLat 15.4. For each panel the % reactivation is shown in blue with the primary y-axis, and the %LIVE in the live gate from flow cytometry is the orange trend and values displayed on the secondary y-axis to the right. As only 48h treatments were used in Figure 5, the reactivation and viability of mid-concentration levels for 24h treatment were not included in the manuscript but were performed for more rigorous investigation into the reactivation of this clone and the potency of the new treatments used.

H. The authors should describe in more detail the importance of the compound mixtures which enhanced cell migration, especially in regards to the treatment of HIV-infected individuals with these drug combinations.

We expand upon previous discussion of the increase of migration and reactivation by combinations with TNF in the results and discussion section of the main text:

Results section line 203:

“Additionally, cancer treatments such as 17β-Estradiol (E2) and Cytarabine enhanced migration of latently infected cells (Fig. 5). Custom drug strategies for HIV+ cancer patients may prove important given the overexpression of CXCR4 in more than 20 cancer types compared to non-cancerous cells⁴²⁻⁴⁵.”

Discussion section line 297:

“Finally, anticancer drug treatments of E2 or Cytarabine raise additional risk in HIV+ cancer patients by migration of latently infected T-cells which underlines the need for customized latency removal treatments individualized for each patient and their pre-existing medical conditions. In addition, migratory cocktails and their diverse mechanisms affecting both the latent reservoir and uninfected bystander T-cells will need to be accounted for in designing migration strategies (Figs. 5 and 6, and Supplementary Fig. 9).”

Furthermore, would their results be the same if resting memory T cells containing latent proviruses were used for this migration assay.

We have now performed new experiments presented in Figure 6 that show consistency of correlated CXCR4-LTR migration and expression in infected and GFP+ sorted primary CD4+ T-cells. We also show that diverse drug treatments of primary T-cells infected with full-length HIV (JLatd2GFP) also have modulated migration levels consistent with the Jurkats in Figures 4 and 5 (TNF addition decouples migration, SAHA reduces migration, and PMA or PMA+Ionomycin completely remove migration).

We have elaborated on the challenges of performing migration experiments on a partial sub-population of latent provirus in primary T-cells which cannot be sorted to seed the migration assay in Supplementary Figure 8. In short, as there is no way to isolate a pure population of latently infected cells, an unsorted population consisting primarily of uninfected cells results in inconsistent results and migration experiments that do not reflect the migratory behavior of a reactivating latent phenotype. For this reason, we found latent cell migration experiments to not be feasible and noted this in the main text and supplementary information. To perform primary cell experiments we instead sorted infected GFP+ primary cells which would be more similar to a reactivated cell state with elevated Tat levels already present within the cell.

The concluding paragraph describing what their findings mean for the treatment of HIV disease should be expanded. The manuscript was a little difficult to follow as it did not include an abstract and the results were not separated into sections with titles.

We agree that the discussion needs elaboration and have now provided an expanded discussion section providing more in-depth aspects of the findings and implications of the research. In addition we have expanded the concluding paragraphs to elaborate on the findings for HIV treatment including inflammation, coagulation, and the coupling of both tissue factor and CMV promoters (Deeks 2011 Annu Rev Med. 62:141-155).

Please find the updated discussion section in the revised manuscript. We have purposely avoided including the whole discussion in this letter due to its length.

Reviewer #3 (Remarks to the Author):

Review for Nature Communications Manuscript COMMS-16-15460-T “Genetic coupling of viral-host gene expression presents migratory challenges in HIV therapies”

Kathrin Bohn-Wippert, Erin N. Tevonian, Melina R. Megaridis, Roy D. Dar

Summary: In this manuscript, the authors present evidence that the HIV LTR and CMV MIEP promoters are genetically coupled to host promoters with similar architecture. The authors first use a computational approach to identify a set of 285 host promoters with LTR/MIEP-like architecture. Following manually curation and extensive drug perturbation microarray analysis to identify correlated gene expression, the authors focus on two host genes, F3 and CXCR4, for which promoter architecture and drug perturbation

analysis suggests that the expression of these genes is genetically coupled to viral gene expression in HIV and CMV infections. The authors further show that this genetic coupling between HIV and CXCR4 can impact CD4+ T-cell migration of HIV-infected cells. This coupling could lead to reactivation of HIV in niches where HAART cannot penetrate, and can result in new, unwanted infections. Thus, genetic coupling of HIV and CXCR4 has important implications for the design of “shock and kill” anti-latency therapies.

Overall Critique: The authors put forth a very interesting idea of genetic coupling between a host and pathogen based on promoter architecture. The implications of this work are of great interest to the HIV field in general, and more specifically to the HIV “Cure” researchers. Furthermore, with HAART, HIV+ individuals live for very long periods, and many are now dealing with other diseases associated with aging. The authors note that certain cancer therapy could inadvertently induce migration and reactivation of the latent HIV+ CD4+ T-cell pool. While this manuscript is very interesting, there are several points at which it could be improved for the broader Nature Communications readership:

1. The authors note in the Methods section that ~285 promoters had at least one NFkB, Sp1, and TBP site. For completeness, the list of genes that met the automated search cutoff should be included as supplementary information.

A table with a list of genes meeting the NFkB, Sp1, and TBP search criteria has now been included in Supplementary Table 1. We will note that the total number of genes in the table is larger than 285. This difference is explained in the methods section (“Analysis of promoter gene sequences”, line 418-439) detailing the promoter search, and in no way affects the main focus of the current study, the highly coupled promoters selected for further investigation, and subsequent findings.

We have updated the results section mentioning the total number of 366 promoters and included the supplementary table (line 73):

“The presence of additional binding site types were ignored to de-constrain the search and resulted in 366 TBP-SP1-NFKB gene promoters (Supplementary Table 1).”

2. The authors also show 13 host promoters in Supp Figure 1. How were these chosen? Were these the promoters that also met the “manual curation” threshold? The authors note that the genes associated with these 13 promoters have important functions in viral pathogenesis. For the larger set of 285 genes that met computational cutoff, was there significant enrichment in specific cellular processes, as suggested by this more refined set of 13 genes?

The promoters in Supplementary Figure 1 were chosen by “manual curation” by scanning the results of the promoter search in the +/-250 bp window for similarity (very few promoters showed the regulatory arrangement needed in the +/-250bp region from the TSS making manual scanning of the 366 search results fairly straightforward). The figure represents promoters with the highest similarity towards the top, and some with less similarity in the arrangement, distances, and site numbers as the list descends (described in the figure legend).

The question regarding enrichment of cellular processes in the total NFkB-SP1-TATA gene group is an interesting one which we lightly touched upon in the original submission. We have now used DAVID Bioinformatics Resources, (<https://david.ncifcrf.gov/home.jsp>, *Nature Protocols* 2009; 4(1):44 & *Nucleic Acids Res.* 2009;37(1):1) to cluster the gene group to list the most enriched processes and localizations. Although some

gene ontologies displayed more enrichment than others (e.g. localized in the nucleus and involved in transcriptional regulation), the large assortment of gene ontologies represented below lead us to believe there is not a significant enrichment to describe the gene set with promoters from our *cis* regulatory binding site search (i.e. NFKB-SP1-TBP group). Gene ontology analysis was performed on all 366 promoters included in the new Supplementary Table 1. We believe there are several follow-on studies to the computational portion of the paper, but at this time would merit a larger effort to tease out if there is indeed an effect. For now we have provided the annotated ontology chart and clustering below and decided not to include them in the Supplementary Information. The notation of the promoters of Supplementary Figure 1 have been included to point out that CXCR4 and F3 are integral to the cell membrane.

Functional Annotation Chart

Sublist	Category	Term	RT	Genes	Count	%	P-Value	Benjamini
[ ]	UP_SEQ_FEATURE	splice variant	RT		143	43.3	1.6E-2	8.0E-1
[ ]	SP_PIR_KEYWORDS	alternative splicing	RT		143	43.3	1.6E-2	7.0E-1
[ ]	SP_PIR_KEYWORDS	phosphoprotein	RT		140	42.4	1.3E-2	7.0E-1
[ ]	SP_PIR_KEYWORDS	nucleus	RT		86	26.1	2.9E-2	5.9E-1
[ ]	GOTERM_MF_FAT	ion binding	RT		86	26.1	4.8E-2	1.0E0
[ ]	GOTERM_MF_FAT	cation binding	RT		85	25.8	4.6E-2	1.0E0
[ ]	GOTERM_MF_FAT	metal ion binding	RT		83	25.2	6.7E-2	1.0E0
[ ]	SP_PIR_KEYWORDS	metal-binding	RT		66	20.0	8.6E-3	7.9E-1
[ ]	GOTERM_BP_FAT	regulation of transcription	RT		61	18.5	1.9E-2	6.7E-1
[ ]	GOTERM_MF_FAT	transition metal ion binding	RT		58	17.6	7.9E-2	9.9E-1
[ ]	SP_PIR_KEYWORDS	acetylation	RT		57	17.3	2.5E-2	6.0E-1
[ ]	UP_SEQ_FEATURE	mutagenesis site	RT		49	14.8	7.6E-3	6.7E-1
[ ]	SP_PIR_KEYWORDS	Transcription	RT		47	14.2	2.1E-2	5.8E-1
[ ]	GOTERM_BP_FAT	transcription	RT		47	14.2	8.1E-2	8.0E-1
[ ]	GOTERM_BP_FAT	regulation of RNA metabolic process	RT		46	13.9	1.3E-2	6.0E-1
[ ]	SP_PIR_KEYWORDS	zinc	RT		46	13.9	6.7E-2	7.0E-1
[ ]	GOTERM_BP_FAT	regulation of transcription, DNA-dependent	RT		45	13.6	1.4E-2	6.0E-1
[ ]	SP_PIR_KEYWORDS	transcription regulation	RT		45	13.6	3.3E-2	5.9E-1
[ ]	SP_PIR_KEYWORDS	coiled coil	RT		44	13.3	4.6E-2	6.2E-1
[ ]	SP_PIR_KEYWORDS	dna-binding	RT		42	12.7	3.4E-2	5.7E-1
[ ]	GOTERM_CC_FAT	organelle membrane	RT		28	8.5	7.2E-3	8.7E-1
[ ]	GOTERM_CC_FAT	mitochondrion	RT		25	7.6	3.6E-2	8.2E-1
[ ]	GOTERM_BP_FAT	cell cycle	RT		23	7.0	2.2E-2	7.0E-1
[ ]	GOTERM_BP_FAT	regulation of cell proliferation	RT		23	7.0	2.5E-2	7.3E-1
[ ]	GOTERM_CC_FAT	Golgi apparatus	RT		22	6.7	2.2E-2	9.5E-1
[ ]	GOTERM_BP_FAT	regulation of cell death	RT		22	6.7	5.9E-2	7.8E-1
[ ]	GOTERM_MF_FAT	identical protein binding	RT		21	6.4	7.1E-3	9.6E-1
[ ]	SP_PIR_KEYWORDS	developmental protein	RT		21	6.4	3.3E-2	6.1E-1
[ ]	INTERPRO	Zinc finger, C2H2-like	RT		21	6.4	5.2E-2	1.0E0
[ ]	SMART	ZnF_C2H2	RT		21	6.4	5.4E-2	9.3E-1
[ ]	GOTERM_BP_FAT	regulation of programmed cell death	RT		21	6.4	9.1E-2	8.2E-1
[ ]	GOTERM_BP_FAT	positive regulation of molecular function	RT		20	6.1	9.1E-3	6.8E-1
[ ]	GOTERM_CC_FAT	endomembrane system	RT		20	6.1	2.6E-2	9.2E-1
[ ]	GOTERM_BP_FAT	cell death	RT		20	6.1	5.7E-2	7.8E-1
[ ]	GOTERM_BP_FAT	death	RT		20	6.1	6.0E-2	7.7E-1
[ ]	INTERPRO	Zinc finger, C2H2-type	RT		20	6.1	7.5E-2	1.0E0

Functional Annotation Clustering (Top 4 clusters)

Annotation Cluster 1		Enrichment Score: 1.81			Count	P_Value	Benjamini
[ ]	GOTERM_BP_FAT	positive regulation of protein kinase activity	RT		13	6.9E-4	4.5E-1
[ ]	GOTERM_BP_FAT	positive regulation of kinase activity	RT		13	9.3E-4	3.3E-1
[ ]	GOTERM_BP_FAT	positive regulation of transferase activity	RT		13	1.3E-3	3.1E-1
[ ]	GOTERM_BP_FAT	activation of protein kinase activity	RT		8	4.4E-3	5.7E-1
[ ]	GOTERM_BP_FAT	regulation of kinase activity	RT		15	5.0E-3	5.5E-1
[ ]	GOTERM_BP_FAT	regulation of transferase activity	RT		15	7.2E-3	6.5E-1
[ ]	GOTERM_BP_FAT	positive regulation of molecular function	RT		20	9.1E-3	6.8E-1
[ ]	GOTERM_BP_FAT	regulation of protein kinase activity	RT		14	9.3E-3	6.6E-1
[ ]	GOTERM_BP_FAT	regulation of phosphorylation	RT		17	9.7E-3	6.5E-1
[ ]	GOTERM_BP_FAT	positive regulation of catalytic activity	RT		18	1.2E-2	6.7E-1
[ ]	GOTERM_BP_FAT	regulation of phosphate metabolic process	RT		17	1.4E-2	6.0E-1
[ ]	GOTERM_BP_FAT	regulation of phosphorus metabolic process	RT		17	1.4E-2	6.0E-1
[ ]	GOTERM_BP_FAT	activation of MAPKK activity	RT		3	7.3E-2	8.1E-1
[ ]	GOTERM_BP_FAT	MAPKKK cascade	RT		6	2.3E-1	9.2E-1
[ ]	GOTERM_BP_FAT	JNK cascade	RT		3	2.8E-1	9.4E-1
[ ]	GOTERM_BP_FAT	stress-activated protein kinase signaling pathway	RT		3	3.1E-1	9.5E-1
[ ]	GOTERM_BP_FAT	protein kinase cascade	RT		8	4.9E-1	9.8E-1
Annotation Cluster 2		Enrichment Score: 1.62			Count	P_Value	Benjamini
[ ]	GOTERM_BP_FAT	cell morphogenesis involved in differentiation	RT		12	4.5E-3	5.4E-1
[ ]	GOTERM_BP_FAT	axonogenesis	RT		10	8.0E-3	6.5E-1
[ ]	GOTERM_BP_FAT	cell morphogenesis	RT		14	1.2E-2	7.0E-1
[ ]	GOTERM_BP_FAT	cellular component morphogenesis	RT		15	1.2E-2	6.6E-1
[ ]	GOTERM_BP_FAT	neuron differentiation	RT		16	1.2E-2	6.4E-1
[ ]	GOTERM_BP_FAT	cell morphogenesis involved in neuron differentiation	RT		10	1.3E-2	6.2E-1
[ ]	GOTERM_BP_FAT	neuron projection morphogenesis	RT		10	1.5E-2	5.9E-1
[ ]	GOTERM_BP_FAT	cell projection morphogenesis	RT		10	3.2E-2	7.6E-1
[ ]	GOTERM_BP_FAT	cell part morphogenesis	RT		10	4.1E-2	7.8E-1
[ ]	GOTERM_BP_FAT	neuron projection development	RT		10	4.1E-2	7.8E-1
[ ]	GOTERM_BP_FAT	neuron development	RT		12	4.2E-2	7.8E-1
[ ]	GOTERM_BP_FAT	axon guidance	RT		6	4.3E-2	7.9E-1
[ ]	GOTERM_BP_FAT	cell projection organization	RT		11	1.2E-1	8.5E-1
[ ]	GOTERM_BP_FAT	cell motion	RT		13	1.4E-1	8.6E-1

Annotation Cluster 3		Enrichment Score: 1.61			Count	P_Value	Benjamini
[ ]	GOTERM_BP_FAT	branching morphogenesis of a tube	RT		7	1.1E-3	3.0E-1
[ ]	GOTERM_BP_FAT	tube morphogenesis	RT		9	2.0E-3	3.9E-1
[ ]	GOTERM_BP_FAT	morphogenesis of a branching structure	RT		7	2.1E-3	3.6E-1
[ ]	GOTERM_BP_FAT	angiogenesis	RT		7	5.0E-2	7.8E-1
[ ]	GOTERM_BP_FAT	blood vessel morphogenesis	RT		8	8.5E-2	8.2E-1
[ ]	GOTERM_BP_FAT	blood vessel development	RT		8	1.5E-1	8.7E-1
[ ]	GOTERM_BP_FAT	vasculature development	RT		8	1.6E-1	8.9E-1
[ ]	SP_PIR_KEYWORDS	angiogenesis	RT		3	3.0E-1	9.4E-1
Annotation Cluster 4		Enrichment Score: 1.61			Count	P_Value	Benjamini
[ ]	UP_SEQ_FEATURE	zinc finger region:C2H2-type 7	RT		19	5.3E-4	4.2E-1
[ ]	UP_SEQ_FEATURE	domain:KRAB	RT		15	1.3E-3	4.8E-1
[ ]	UP_SEQ_FEATURE	zinc finger region:C2H2-type 6	RT		19	1.3E-3	3.6E-1
[ ]	INTERPRO	Krueppel-associated box	RT		15	3.0E-3	8.4E-1
[ ]	SMART	KRAB	RT		15	3.1E-3	3.6E-1
[ ]	UP_SEQ_FEATURE	zinc finger region:C2H2-type 5	RT		19	3.7E-3	6.2E-1
[ ]	UP_SEQ_FEATURE	zinc finger region:C2H2-type 8	RT		16	4.1E-3	5.7E-1
[ ]	PIR_SUPERFAMILY	PIRSF005559:zinc finger protein ZFP-36	RT		8	9.9E-3	7.8E-1
[ ]	GOTERM_BP_FAT	regulation of RNA metabolic process	RT		46	1.3E-2	6.0E-1
[ ]	GOTERM_BP_FAT	regulation of transcription, DNA-dependent	RT		45	1.4E-2	6.0E-1
[ ]	UP_SEQ_FEATURE	zinc finger region:C2H2-type 4	RT		18	1.5E-2	8.6E-1
[ ]	UP_SEQ_FEATURE	zinc finger region:C2H2-type 3	RT		19	1.5E-2	8.3E-1
[ ]	GOTERM_BP_FAT	regulation of transcription	RT		61	1.9E-2	6.7E-1
[ ]	INTERPRO	Zinc finger, C2H2-type/integrase, DNA-binding	RT		19	1.9E-2	1.0E0
[ ]	SP_PIR_KEYWORDS	Transcription	RT		47	2.1E-2	5.8E-1
[ ]	UP_SEQ_FEATURE	zinc finger region:C2H2-type 9	RT		13	2.3E-2	8.8E-1
[ ]	UP_SEQ_FEATURE	zinc finger region:C2H2-type 10	RT		12	2.3E-2	8.7E-1
[ ]	UP_SEQ_FEATURE	zinc finger region:C2H2-type 2	RT		18	2.4E-2	8.6E-1
[ ]	SP_PIR_KEYWORDS	nucleus	RT		86	2.9E-2	5.9E-1
[ ]	SP_PIR_KEYWORDS	transcription regulation	RT		45	3.3E-2	5.9E-1
[ ]	SP_PIR_KEYWORDS	dna-binding	RT		42	3.4E-2	5.7E-1
[ ]	INTERPRO	Zinc finger, C2H2-like	RT		21	5.2E-2	1.0E0
[ ]	SMART	ZnF_C2H2	RT		21	5.4E-2	9.3E-1
[ ]	SP_PIR_KEYWORDS	zinc	RT		46	6.7E-2	7.0E-1
[ ]	INTERPRO	Zinc finger, C2H2-type	RT		20	7.5E-2	1.0E0
[ ]	GOTERM_BP_FAT	transcription	RT		47	8.1E-2	8.1E-1
[ ]	UP_SEQ_FEATURE	zinc finger region:C2H2-type 1	RT		14	1.1E-1	1.0E0
[ ]	UP_SEQ_FEATURE	zinc finger region:C2H2-type 13	RT		7	1.2E-1	1.0E0
[ ]	UP_SEQ_FEATURE	zinc finger region:C2H2-type 12	RT		8	1.2E-1	1.0E0
[ ]	GOTERM_MF_FAT	DNA binding	RT		48	1.3E-1	1.0E0

3. In Figure 1b, 2b,c and d, only F3 and CXCR4 are highlighted. Where do the other 11 genes (from Supp Figure 1) fall on the distribution in 2e?

To address this comment, we have now added the distribution of Figure 2e into an updated lower panel of Supplementary Figure 1 to include the additional co-expression slopes with the LTR co-expression distribution with available microarray data reported in the microarray dataset.

The additional gene promoters show reduced structural similarity from F3 and CXCR4 (X4) along with a broad range of co-expression slopes suggesting that the criteria of having all *cis* element types (TATA-SP1-NFKB) in the promoter is insufficient for high co-expression and co-regulation. This has been expanded upon in the caption of Supplementary Figure 1:

*“Despite having regulatory similarity with regards to the presence of TATA-SP1-NFKB sites (without necessary similarity in arrangement, number of sites, and distances from the TSS), unlike CXCR4 and F3 which have high co-expression slopes, these promoters displayed a range of co-expression slopes when compared with the HIV LTR as in Figure 2. Genes labeled on the histogram are all of the genes from the promoter subgroup that exist in the Lamb et al., 2006 microarray dataset¹. The range of resulting co-expression slopes supports that existence of the three *cis* element types is insufficient for high co-expression seen with CXCR4 and F3, and further supports their selection as genetically coupled, co-regulated, and among the most co-expressed promoters with the HIV LTR for phenotypic investigation in the study.”*

4. What are the other genes that fall to the right of CXCR4 in Figure 2e, if they are not the 9 genes in Supp Fig 1? The authors should address why other genes with different promoter architecture might be highly correlated with HIV LTR expression. How many of the hits are probably due to non-transcriptional effects of drug treatment driving correlated expression? Based on the promoter architecture of the genes that fall to the right of CXCR4, how many correlations are due to pleiotropic effects of drug perturbations resulting in perceived correlation?

CXCR4 is ranked with the 23rd highest co-expression slope with the LTR of the ~10k human genes in the distribution. Other than CXCR4, no other genes with higher co-expression slopes pass the criteria of having NFKB-SP1-TBP in their promoter further motivating the selection of CXCR4 as both similar and highly co-expressed. As a different sub-set of transcription factors may be controlling co-expression it is difficult to discern transcriptional from non-transcriptional effects resulting in perceived correlation without expanding the logic-based search of promoters to include additional iterative searches with *cis* regulatory binding site types and scanning a large number of site types (e.g. in groups of three transcription factor species at a time). This advanced study requires a larger and extensive computational effort.

Assuming these higher co-expressed genes are non-transcriptional (i.e. based on the *cis* element search) we would find that a large majority of genes with higher co-expression are either post-transcriptionally regulated or have additional protein-protein, mRNA regulation, and indirect regulatory network interactions (i.e. pleiotropic effects) that drive this co-expression. Although of great computational interest for future investigations to infer drug interactions and regulatory relationships, as this is not the main focus of this study, we chose not to address these highly co-expressed genes in the current manuscript.

5. Many observations from immortalized cell lines do not hold in primary cell models. While reactivation cannot be reasonably tested in a primary CD4+ T-cell model, it would be powerful to show that LTR-GFP expression and F3/CXCR4 expression are correlated in primary human CD4+ T-cells using a few drugs from the panel used in Figure 2.

We have now performed new primary CD4+ T-cell experiments with 24h treatments using a subset of the drug panel from Figure 2d between Jurkat and primary cells infected with HIV-1 LTR-GFP (see result below). The GFP+ cells of the unsorted population of the primary CD4+ T-cells were used to determine the mean fluorescence. By comparing LTR infected Jurkat cells already measured with 10 μ M of these compounds to primary cells, the experiment aimed to show correlated response between Jurkat and primary cells and subsequently a correlation which is exhibited by Jurkat LTR-mCherry with the CXCR4 and F3 promoters in HL60 cells with microarray. Unfortunately, our single measurements below are inconclusive and do not provide the correlated signature. Compounds were selected from a broad range (\pm x1 LOG₂ Fold Change) of LTR-mCherry fluorescence change (from Dar 2014 Science) in order to cover a large correlated expression range with minimal treatments. Figure 2d compares microarray data using CXCR4 mRNA and the fluorescence of the mCherry protein of HIV-1 LTR-mCherry infected Jurkats from a drug screen allowing the comparison of 262 measurements. To quantify co-expression slopes for all genes in Figure 2, including CXCR4, a 50 drug moving average was applied to the data sorted in descending order by the Jurkat LTR-mCherry Fold Change. This means that every point on the co-expression trend quantified represents 50 drug response measurements and minimizes deviations by any one treatment/measurement. Limited in throughput of isolated and infected primary CD4+ T-cells, without a full drug screen/assay we were only able to measure 6 drug treatments carried out in duplicate from a single donor -- far less than those used to conclude co-expression in Figure 2d. Despite not being able to prove co-expression for the compound subset in primary cells the new experiments of LTR expression and migration in primary cells shown in Figure 6 are consistent with the LTR-CXCR4 coupling, JLat2GFP decoupling, and drug-modulated migration shown in Jurkat T-cells.

Figure: Comparison of expression with treatments of CD4+ LTR-GFP and Jurkat LTR-mCherry.

We performed primary CD4+ T-cell experiments using only the HIV-1 LTR promoter to show that correlated migration and viral expression still exists in primary human T-cells. At first, migration assays were carried out using unsorted cell populations to use the number of cells required for seeding each migration measurement, maximize the number of drug measurements, and provide the reader with duplicate results (Fig. 6b). We report co-expression of LTR and CXCR4 in primary human T-cells using 48h TNF treatment on three separate donors (Fig. 6b).

To confirm that the behavior observed exists on a purely sorted infected cell population, we sorted infected cells that were GFP+ using FACS at our flow cytometry facility. Sorted CD4+ cells infected with LTR-GFP showed a strong correlation between <GFP FL> and increased migration with TNF treatment along with Figure 6b of unsorted cells. This measurement demonstrates coupled expression for a purely sorted population of cells without the need to assume equal migration rates between LTR-GFP infected and uninfected cells (Fig. 6c).

Can the authors also demonstrate that primary human CD4+ T-cells migrate similarly to data in Figure 5b in response to different “shock and kill” combinations.

To address this comment, we have carried out additional cell migration experiments using isolated primary CD4+ T-cells from whole blood infected with full-length HIV (JLatd2GFP, Δenv and d2GFP in place of nef) (Fig. 6d). Cells were sorted using FACS to get a GFP+ population of infected T-cells. An array of potent drugs for migration and reactivation were used from Figure 5 to investigate the migratory behavior in infected primary T-cells. The new Figure 6d in the main text shows that primary cells infected with full-length HIV behave similarly to the JLats (Fig. 5) and that elevated levels of Tat, enriched in cells sorted GFP+, is sufficient to suppress increases of migration after 48h TNF treatment. In addition, we find that SAHA alone and combined with TNF reduces migration and PMA and PMA+Ionomycin completely stop migration in these cells. Results of these experiments are described in line 235 of the result section:

“Using selected drugs from Figure 5, the measurements show that TNF treatment does not increase migration (i.e. decoupled from the LTR-GFP case in panels b and c), SAHA decreases migration, and PMA and PMA+Ionomycin completely suppress migration (Fig. 6d). These results are consistent with treatments on JLats (Figs. 4 and 5).”

We note that although we specifically set out to perform experiments on the reactivation of latently infected primary cell populations, it was quickly understood that after sorting the GFP- or OFF population to isolate a mixture of low numbers of latent cells with a large majority of uninfected cells, that the dominating determinant of migratory behavior would be the uninfected cells. We show and explain this challenge for any future work on assaying migrating latent cells in a new Supplementary Figure 8 in which unsorted and GFP-/OFF-sorted JLat2GFP infected primary cells are not able to decouple and reduce TNF-treated cell populations from increasing their migration. Currently without a biomarker to identify and sort latent cells for a full latent population, along with the experimental obstacle of the required amount of ~200k sorted CD4+ cells for each drug treatment measurement which severely limited our throughput for each blood donation, the latent infected cell migration experiments will not be able to be performed. Therefore, we resorted to performing experiments on primary cells that were fully infected, activated, and sorted for showing migratory shifts of treated cells.

6. Figure 5a doesn't add a significant amount of understanding beyond what 5b accomplishes.

We thank and agree with the reviewer on this observation and have now condensed Figure 5 to focus on the main findings of Figure 5b. Figure 5 has also had the background shading removed to clearly present the data.

7. The sentence in lines 107-109 of the main text needs to be revised for clarity.

The meaning of this sentence was unclear to more than one reviewer. We have now clarified lines 111 in the revised text to read:

“Promoter coupling between distant viral families (RNA and DNA) that co-infect their host, including multiple hosts throughout their evolution (SIV and HIV), demonstrates that genetic coupling may elucidate a deeper understanding of co-evolution of multiple viruses and their hosts¹⁸ along with their co-infection dynamics.”

8. Some discussion of why TNF treatment results in a smaller impact on cell migration for MIEP and LTR vs naïve Jurkats would be useful (Figure 4a and Supp Figure 3b).

The reviewer raises an interesting observation. However, the evaluations of our results have shown that there is no clear trend for increased migration after the infection of Jurkat cells. This means that migration values for untreated cell populations are subject to variability and we observed higher migration values for Naïve Jurkats and lower migration values for Jurkats infected with LTR-GFP (as shown in the table below). Another negation of this trend is demonstrated by the isoclone JLat 6.3 which shows migration comparable to naïve Jurkats, and smaller levels of migration compared to the JLat 15.4 and 9.2 (Fig. 4a and 4b).

96 well plate	Migration (Cells/ml)		
	48h treatment		
CLG poly Untreated	338000		
CLG poly TNF-treated	476000		
	24h treatment		
CLG poly Untreated	580000	196000	197000
CLG poly TNF-treated	899000	253000	233000
	24h treatment		
Naïve Jurkats Untreated	523000	153000	92800
Naïve Jurkats TNF-treated	555000	190000	153000

Table: Comparison of migration for uninfected and infected Jurkats cells after 24h and 48h of treatment using a 96-well migration plate. Migration can be different for the same cell line and high as well as low migration can be observed. However, the TNF-treated cell population shows increased migration than the untreated counterpart.

Variability in migration can be caused by cellular factors like the concentration of calcium in the cell (Valente 2001, J Virol 75:439-447), or the ZAP-7-dependent tyrosine kinase which is necessary for CXCR4 chemokine receptor signaling in human T cells (Ottoson 2001 J Immunol 167:1857-1861). Notwithstanding these variations, it is clear that all TNF-treated Jurkats, uninfected or infected with the LTR-GFP or MIEP-mCherry, show an increase of migration after 24h-48h treatment.

Can the authors also comment on why the basal level of cell migration is higher in MIEP and LTR Jurkats vs naïve Jurkats? Does infection create some sort of “on” state for NFkB-, Sp1-, and TBP-driven gene expression of CXCR4? The authors should quantify CXCR4 expression to determine if it is higher in LG and MIEP-cherry cells.

We have performed CXCR4 antibody (Ab) staining to investigate if untreated naïve Jurkats truly have lower levels of basal CXCR4 on the surface pre-treatment compared to MIEP and LTR infected Jurkats. The results below show that levels of CXCR4 can be lower for uninfected compared to infected Jurkat cells. In addition, the CMV promoter infected cells showed comparable CXCR4 Ab <PE FL> to the LTR-GFP case. This is consistent with lower CXCR4 in uninfected cells.

Figure: Staining of the CXCR4 receptor of naïve Jurkats as well as Jurkats infected with either HIV-1 LTR-GFP or CMV MIEP-mCherry using a PE-labeled CXCR4 antibody.

Collectively, our investigations do not support a consistent trend between infected and uninfected cell migration. We emphasize that all of the main conclusions in the current study are based on general behaviors relative to consistent internal controls of the cell line being observed. To limit potential variations of CXCR4 receptors on the cell surface caused by cell age and cell culturing, we consistently used cells that were pre-cultured for 2 weeks and the same batch of cells to carry out migration assays. In addition, we expanded all cultivated cells on the same day.

9. Since Tat expression decouples LTR/CXCR4 expression, and in fact reverses the impact on cell migration due to secondary interactions between Tat and CXCR4, the kinetics of reactivation vs migration are key to determining how important this phenomenon will be to “shock and kill” design. How far can cells travel in the first 48hrs after reactivation? The authors present volumetric migration data. Can they quantify cell speed?

The current volumetric migration data suggests that even with decreases of migration with Tat, reactivated cells continue to migrate at 2-days into reactivation from latency. With slight differences in reported calculations, the heart pumps 5L of blood per minute through our circulatory system (basically all of our blood). Approximately a quarter of this blood are red blood cells that circulate throughout the body in 20 sec. Our measurements show that when migration is not suppressed, we are seeing minimal migration rates of 100k cells per ml per hour (in volumetric transwell plates), or ~1.7 cells per ml per min (~2k cells/L of blood per minute). This back of the envelope calculation gets further complicated by additional migratory axes encountered throughout *in vivo* migration by other chemotactic pathways, and the differences in reaching lymph node and other distal regions. Despite these considerations we believe that with the extended migratory window allowed by questionable cell death post 48 hours of treatment/reactivation (see extended discussion, Line 308), the observation of ongoing migration over the 2 day period (e.g. assuming cases where cells turn on in the first hour or in the 47th hour), would reach a majority of the target-rich niches of the body described in the paper.

We also note experimental evidence provided by Marooka et al. (Marooka 2012 Nature 490: 283-7) in which mice were infected with HIV in the foot, and monitoring circulation of productive HIV infected cells with and without an Env deletion led to detection of modulated infected cell formation of viral synapse in the mouse lymph nodes – suggesting that in mice, although quantified to decelerate migration, Tat producing cells do not stop the cells from reaching target-rich regions.

Referencing the stochastic nature of single-cell HIV reactivation from latency (Weinberger, Dar, Simpson 2008 Nat. Genetics 40: 466-470, Weinberger 2005 Cell122:169-82, and Dar 2014 Science), the above suggests that a distribution of turn-on times in a latent cell population favors cells that reactivate at a later time would most likely make it to lymph nodes and other targeted regions in the body.

The dual role of both genetic coupling to synchronize co-expression in individual cells in order to reduce over-expression of CXCR4 on the cell surface by secondary interactions between Tat and CXCR4 is an intriguing finding at the core of this study – a viral-host relationship at the gene sequence level to control a host cell phenotype. Further studies of kinetics at the single-cell level to understand time-dependent dynamics between reactivation and migration is of great importance and will be pursued in an advanced study as it requires a completely different and extensive experimental platform, analyses, and modeling (e.g. microscopy imaging of single-cell 2D migration/chemotaxis), and quickly deviates from the focus of the present study. The modeling and experimentation of dynamics of co-regulated expression at the single-cell level will provide additional insights into the implications and control of this viral-host relationship.

The data presented in Figure 4d might also address concern, but the implications should be strengthened in the text. The migrating cells have slightly higher rates of reactivation, meaning the cells that do migrate to HAART-inaccessible niches are slightly more likely to be actively producing virus. Perhaps there is more of a dichotomy at early points (less than 48hrs post treatment)? The data in Figures 4b and c suggests 24 or 36hrs might provide robust migration + reactivation, where migrating cells may have an even higher proportion of reactivating cells.

In estimating the reactivation rate of latently infected cells we find a strong linear fit of %ON versus duration of TNF treatment for the various JLat clones (new Supplementary Information and found in response to Reviewer 1 above). We have now performed two new experiments for JLat 9.3 and 15.4 with TNF treatment of 24h to test if earlier time points with higher migration may show a larger migration/non-migrating reactivation difference. Our results (Supplementary Fig. 6, inserted below for convenience) show that the proportion of reactivating cells in the migrating cell population is gradually building up, consistent with a constant reactivation rate, and does not

show any accentuated dichotomy or higher proportion between the migrating and non-migrating populations. We have extended the text regarding this observation in the results and the discussion:

Line 172 of the results section:

“Strikingly, despite a consistent trend of reduced migration for HIV expressing cells^{2,3}, JLat and minimal Tat clones treated with TNF for 24h or 48h reveal that sub-populations of migrating cells reactivate at higher rates compared to stationary (or non-migrating) cells (Fig. 4e, Supplementary Fig. 6, and Supplementary Information).”

Line 318 of the discussion section:

“Migratory challenges to HIV “shock and kill” presented in this study are only further exacerbated by this lack of T-cell death, providing a longer migration window accompanied by viral shedding of reactivated cells (Fig. 3b). Furthermore, in a latent population, migrating cells reactivate more than stationary cells (Fig. 4d and Supplementary Fig. 6), presenting an additional risk for viral spread with “shock and kill” (Fig. 3b).”

“Supplementary Figure 6: Reactivation of migrating and non-migrating cells of JLat isoclone 9.2 and 15.4 after 24h and 48h of TNF-treatment.

To test for any time dependence in the effect seen in Figure 4e, with migrating cells reactivating more than non-migrating cells, measurements after 24h treatment were performed for comparison to 48h. The results support a constant reactivation rate with TNF treatment of JLat clones over time. The results reveal that migrating cells consistently reactivate more compared to non-migrating cells with higher reactivation rate as the percentage of “ON” cells is consistently higher over time (see section on reactivation rate calculation above). Migration assays were performed in duplicate and triplicate for 24h and 48h respectively using a 24-well migration plate, and average values are plotted with standard-error.”

In summary, this manuscript raises important questions about therapy design by demonstrating genetic coupling between HIV and human promoters. With some revisions, the claims of the manuscript would be strengthened, and would be of great interest to the Nature Communications readership. This manuscript should be considered for publication provided these revisions are addressed.

We thank the reviewer for their comments and appreciate their strong support of the manuscript for publication provided the revisions are addressed.

REVIEWERS' COMMENTS:

Reviewer #1 (Remarks to the Author):

The authors present a revised manuscript describing the novel co-regulation of viral and human gene promoters that is primarily focused on co-regulation of HIV-1 LTR and CMV-MIEP promoters with human CXCR4 or F3 promoters. Additionally, this study indicates that HIV-1 and CXCR4 promoter co-expression might impact cell migration during reactivation of latent HIV, and that latency reversing agents (LRAs) can have variable effects on reactivation and cell migration. This will be of interest to the HIV latency field and also the broader virus-host factor field given that the initial data revealed co-regulation of HIV or CMV promoter with a range of other host genes.

The authors are to be congratulated for their considered responses and additional experiments to comments raised by the reviewers in the original manuscript. Overall, the revised manuscript is strengthened by the inclusion of the following new items:

- a) Supplementary Table 1 that lists all 366 human promoters with similarity in key transcription factor binding sites to HIV-1.
- b) Providing a clearer justification in Supplementary Figure 1 why studies centred on coregulation of HIV-1 LTR with CXCR4 and F3 due to their high co-expression slopes, rather than other human genes like DERA or BCL2L10 that also had high promoter similarity but lower coexpression slopes.
- c) Including error bars to Fig 4a-c, Fig 5 and Supplementary Figure 3, to help gauge error between various values.
- d) New Fig 4d panel to help demonstrate that TNF-mediated migration is coupled in HIV-LTR expressing cells but de-coupled in Tat expressing JLat or LTIG cell clones.
- e) Supplementary information to try demonstrate that the reactivation rate of HIV-1 LTR from latency in the JLat 6.3, 9.2 and 15.4 clones is higher in migrating cells versus non-migrating cells.
- f) Supplementary Fig 5 to more clearly demonstrate migration and reactivation changes between untreated and TNF treated JLat clones for 12-48 hr timepoints with error bars.
- g) Supplementary Fig 6 to show reactivation of HIV-1 LTR increases in migrating versus non-migrating JLat clones after 48hr TNF.
- h) Providing new details in Supplementary Table 2 for the biological activities of the various latency reversing agent (LRA) compounds tested, % live cells and the untreated cells for comparison.
- i) Fig 6 to examine coregulation of the HIV-1 LTR promoter with cell migration in biologically relevant, activated primary human CD4+ T-cells after infection with minimal LTR-GFP vector or near full length JLatd2EGFP HIV.
- j) Supplementary Figure 8 to provide additional data addressing why the impact of TNF on reactivation and migration of latently infected primary human CD4+ T-cells was unable to be determined ie. due to issues of large numbers of uninfected T-cells in the unsorted or GFP- sorted T-cell populations making it difficult to observe TNF effects on the low numbers of latently infected T-cells in the culture.
- k) Supplementary Fig 7 to demonstrate that for activated uninfected CD4+ T-cells treated with various drug concentrations, the cells within the live cell gate for flow analysis were in fact live given their low levels of propidium iodide staining (dead cell marker).
- l) Supplementary Fig 9 to show how various LRAs impact the migration of uninfected primary human CD4+ T-cells, in line with their activities on the migration of HIV infected cells in Figures 5 and 6d.
- m) A new paragraph in the discussion describing the issue that the turnover of reactivated, latently infected cells maybe prolonged in HIV infected individuals. This issue combined with LRAs that enable migration of reactivated cells may foster reseeding of HIV-1 infection in tissue sites with ineffective HAART and should be avoided.

In addition to these new additions that strengthen the revised manuscript, these new data also raise additional comments for further consideration as follows.

Additional Comments for revised manuscript:

1. Now that error bars are added to Fig 4a and Supplementary Fig 3b, the error bars in migration measurements for LTR-GFP infected Jurkat cells overlap between untreated and TNF-treated cells. Therefore despite a trend for TNF increasing migration as well as transcription, the migration increase does not appear significant when error is taken into account and this should be noted in the results (text lines 141-144).

2. In Supplementary Fig 5, for the 24hr graph, as the error bars for the untreated and TNF treated JLat 9.2 completely overlap, the minor reduction in migration of JLat 9.2 with TNF is not convincing at this time point and the text "Reduced migration values also exist for JLat 9.2 after 24hr" should be removed or modified to include the overlapping error in the measurements.

3. In Supplementary Fig 6, for the 24hr time point with TNF for JLat clones 15.4 and 9.2, the error bars for the % reactivation between migrating and non-migrating cells overlap. Despite the trend of an increase in reactivation in migrating versus non-migrating cells, this change does not look significant due to the overlapping error bars. In contrast, the change looks more convincing at 48 hours for both JLat clones as the error bars don't overlap. Therefore the text in the figure legend should be reworded to account for the error bars, or focus on the increase at 48 hours as this is most convincing.

4. Supplementary information for "Reactivation rate from latency of migrating cells is higher than non-migrating cells" is hard to follow. Section 1 and top graph – is a linear equation really appropriate to calculate the JLat 6.3 reactivation rate as the %ON is similar for 24 and 48 hours for this clone? Section 2: This is hard to follow and please consider rewording, perhaps adding an example with numbers, to make it easier to follow. Section 3: The error bar for JLat 15.4 migrating cells hits the non-migrating cells – doesn't this suggest that their reactivation is similar when error is taken into account?

5. Fig 6 legend and text lines 216-128. Please clarify exactly how the number of GFP+ migrating cells was measured and what is shown in Fig 6 panels b-e. Presumably migration reflects the number of cells counted in the bottom of the transwell. Does FL (a.u) reflect the GFP MFI or number of GFP+ cells in the total cell population or just the migrating cells in the bottom of the transwell? Please reword the sentence on text lines 216-218 – have you given the calculation to measure the number of GFP+ migrating cells? Finally in the methods section lines 523-528, shouldn't the number of GFP+ migrating cells be calculated by multiplying the % GFP+ cells in the bottom of the transwell by the total number of migrating cells in the bottom transwell? Using the % GFP+ in the unsorted infected CD4+ T-cells containing both migrating and non-migrating cells is less relevant if there is more reactivation in the migrating versus non-migrating subsets? Please clarify exactly what was used in the above calculations.

6. Similar to point 1 above for LTR-GFP in Jurkats, the new Fig 6b data showing LTR-GFP infection of unsorted activated human CD4+ T-cells has overlapping error bars for TNF minus and plus samples for Donors 1 and 2 in both migration and GFP graphs. This indicates that these minor increases are not significant. Why are there no error bars for donor 3? These data are not convincing that there is a significant increase in migration for LTR-GFP in unsorted CD4+ T-cells with TNF, similar to Fig 4a in Jurkats (where overlapping migration error bars were also seen minus or plus TNF). The only significant change in migration and reactivation for the activated CD4+ T-cells appears to be for the sorted productive population in Fig 6c, where the error bars do not overlap. The text in lines 209-220 should be altered to reflect the error in these results as outlined above, and emphasise the results in the sorted productive cells in Fig 6c as this is where LTR reactivation and migration is most convincingly coupled.

7. Supplementary Fig 8: For the left bottom graph (GFP in unsorted cells), is there any difference

in the reactivation levels if the % of GFP+ cells is shown for the samples rather than mean GFP fluorescence (where uninfected cells dominate the calculation)? Separately, for the GFP- sorted cells in the top right panel, please show error bars if results from 2 donors are shown. Also, was GFP measured in the GFP- sorted fraction after TNF treatment like the unsorted fraction on the left bottom graph? Please include this actual data if measured, even if very low GFP levels, to show that the technique did not generate enough GFP+ cells to accurately measure differences in GFP reactivation with and without TNF.

8. Supplementary Fig 9 legend: Due to overlapping error bars between the untreated sample and TNF or ionomycin, these changes are unlikely to be significant. Therefore the conclusions that TNF increases migration and ionomycin decreases migration in this single donor are not clear and should be removed or reworded to account for the overlapping error bars.

Minor comments:

9. Fig 1b versus Supplementary Fig 1: F3 has 5 SP1 sites in Fig 1 but 4 SP1 sites Supp Fig 1. Additionally, F3 has all transcription factor binding sites located a negative distance from the TSS in Fig 1b but these sites span negative and positive distances from the TSS in Supp Fig 1. Please ensure these are the same for F3 in Fig 1b and Supp Fig 1.

10. Fig 3, Fig 3 legend, plus text lines 128 and 138 still refer to reactivated latently infected cells dying in 1-2 days. As discussed in reviewer comments for the original manuscript, the author rebuttal and their new paragraph in the discussion, this is a complex issue. The 1-2 day estimated lifespan referenced is for productively infected cells on patients beginning ART. It is not clear how long it will take for reactivated, latently infected cells to die and may depend on the reactivation stimulus and immune cell status. Therefore it is suggested to remove the "1-2 day" time frame from Fig 3, Fig 3 legend and text lines 128 and 138. The main point is if reactivated cells can migrate to tissues with inefficient HAART penetration before they die, they can reseed infection irrespective of the time frame in which they die, which is currently unknown and could very well be longer than 1-2 days depending on the LRA.

11. Text line 137 and Fig 3 legend: please provide references for the target-rich tissue sites with inefficient HAART penetration (eg. LN) that could allow migrating reactivated cells to reseed new infections.

12. In Fig 4b legend, please mention that the untreated JLat samples were measured at 12hr for JLat clones 9.2 and 15.4 but 24hr for clone 6.3 as no 12hr time point was taken. In Supplementary Fig 4 legend, please add that no 12hr or 36hr time point was collected for JLat clone 6.3. In Supplementary Fig 5 legend, please include that the values for the untreated JLats can be found in Supplementary Fig 4 as well as Fig 4b for the TNF-treated JLats.

13. Please add to the discussion the possible reasons given in the author rebuttal on page 8 (Reviewer 1, question 2) as to why if Tat decreases CXCR4 expression from the cell surface and reduces the migration of HIV infected cells, why the migrating Tat expressing JLat and LTIG clones still have higher reactivation than the non-migrating cells. This is nicely explained in the rebuttal and would be helpful to readers if added to the discussion.

14. Text lines 205-208: The text result that Cytarabine "enhanced" migration of latently infected cells is confusing as this sample had less migration than the untreated cells in Fig 5. Is it meant that these drugs still allowed migration of latently infected cells and thus may pose an issue if used to treat cancer in HIV+ patients? Please reword the text for clarity here.

15. Supplementary Table 2: Please add the bioactivities for: cytarabine (cytosine arabinoside incorporated into human DNA and kills cells), Valproic acid (HDACi), Panobinostat (also add that it is a HDACi), JQ1 (bromodomain inhibitor that reactivates HIV transcription) and 5-Aza-2-

deoxycytidine (DNA methyltransferase inhibitor that can reactivate latent HIV).

16. Results text lines 231-233: Line 231-The GFP+ infected and sorted population used in this study actually represents cells initially productively infected that are further stimulated by drugs, which is totally different to latent HIV being reactivated by drugs. Therefore please add "may" to the sentence "This GFP infected and sorted population may represent actively replicating latent cells ie. post reactivation". For clarity on line 233, please also change "transient of reactivation to a fully activated state" to "transition of reactivation to a fully activated state".

17. Methods: Text line 401 - Please change "transfection" to "infection" of primary CD4+ T-cells with lentiviral supernatant. Text line 545 in the Flow analysis section - please define how the live gate was determined ie. FSC vs SSC.

In summary, the authors are to be congratulated on the additional data provided in the revised manuscript and their efforts to address all reviewer questions. The primary human CD4+ T-cell experiments in particular were difficult but have provided interesting new data despite being unable to test the impact of various LRAs on reactivation and migration of latently infected primary T-cells due to technical limitations. However, this new primary T-cell data confirms in productively infected, primary human CD4+ T-cells that HIV-1 LTR transcription and cell migration are both enhanced by TNF in the absence of Tat, and confirm that this trend is lost when Tat-expressing, full length HIV is used for infection similar to the cell line data. They also confirm in productively infected, activated human CD4+ T-cells that certain LRAs enhance HIV expression but importantly impair cell migration. Such LRAs maybe beneficial to reactivating latently infected cells in patients while minimising spread of these reactivated cells to tissue sites with inefficient HAART penetration to prevent reseeding of new infections. Given the novelty of these findings, their interest to the HIV latency field, virus-host factor field and opportunities to follow up other viral-human gene co-regulated promoters, the paper is recommended for publication provided the additional comments above are addressed.

Reviewer #2 (Remarks to the Author):

Bohn-Wippert et al have expanded their research study including new experiments using primary cells and more potent latency reversing compounds. These authors were extremely responsive to the reviewers comments and suggestions for improvement of their research study.

Reviewer #3 (Remarks to the Author):

The authors have satisfactorily revised this manuscript to address my own comments and the comments of the other reviewers. This has substantially improved the manuscript, especially with regards to including some analysis of primary CD4+ T cells. A few minor details:

1. Enrichment analysis was done using DAVID, which is not updated very often. Although there was a recent update in October 2016. Can the authors confirm that all of the enrichment analysis was done using the most recent update?

2. Regarding comment 4, reviewer 3, the authors bring up several possible explanations of why there are some genes that rank higher than CXCR4 in figure 2e. While it is outside the scope of this manuscript, some acknowledgement of these genes in the text would be helpful, including the possible explanations as to how they are regulated given their differing promoter architectures. The main text leaves this issue unaddressed and even just a sentence will add clarity for many readers as to why the other hits are not addressed.

If these minor issues are addressed I believe this manuscript should be accepted for publication.

REVIEWERS' COMMENTS:

Reviewer #1 (Remarks to the Author):

The authors present a revised manuscript describing the novel co-regulation of viral and human gene promoters that is primarily focused on co-regulation of HIV-1 LTR and CMV-MIEP promoters with human CXCR4 or F3 promoters. Additionally, this study indicates that HIV-1 and CXCR4 promoter co-expression might impact cell migration during reactivation of latent HIV, and that latency reversing agents (LRAs) can have variable effects on reactivation and cell migration. This will be of interest to the HIV latency field and also the broader virus-host factor field given that the initial data revealed co-regulation of HIV or CMV promoter with a range of other host genes.

The authors are to be congratulated for their considered responses and additional experiments to comments raised by the reviewers in the original manuscript. Overall, the revised manuscript is strengthened by the inclusion of the following new items:

- a) Supplementary Table 1 that lists all 366 human promoters with similarity in key transcription factor binding sites to HIV-1.
- b) Providing a clearer justification in Supplementary Figure 1 why studies centred on coregulation of HIV-1 LTR with CXCR4 and F3 due to their high co-expression slopes, rather than other human genes like DERA or BCL2L10 that also had high promoter similarity but lower coexpression slopes.
- c) Including error bars to Fig 4a-c, Fig 5 and Supplementary Figure 3, to help gauge error between various values.
- d) New Fig 4d panel to help demonstrate that TNF-mediated migration is coupled in HIV-LTR expressing cells but de-coupled in Tat expressing JLat or LTIG cell clones.
- e) Supplementary information to try demonstrate that the reactivation rate of HIV-1 LTR from latency in the JLat 6.3, 9.2 and 15.4 clones is higher in migrating cells versus non-migrating cells.
- f) Supplementary Fig 5 to more clearly demonstrate migration and reactivation changes between untreated and TNF treated JLat clones for 12-48 hr timepoints with error bars.
- g) Supplementary Fig 6 to show reactivation of HIV-1 LTR increases in migrating versus non-migrating JLat clones after 48hr TNF.
- h) Providing new details in Supplementary Table 2 for the biological activities of the various latency reversing agent (LRA) compounds tested, % live cells and the untreated cells for comparison.
- i) Fig 6 to examine coregulation of the HIV-1 LTR promoter with cell migration in biologically relevant, activated primary human CD4+ T-cells after infection with minimal LTR-GFP vector or near full length JLatd2EGFP HIV.
- j) Supplementary Figure 8 to provide additional data addressing why the impact of TNF on reactivation and migration of latently infected primary human CD4+ T-cells was unable to be determined ie. due to issues of large numbers of uninfected T-cells in the unsorted or GFP- sorted T-cell populations making it difficult to observe TNF effects on the low numbers of latently infected T-cells in the culture.
- k) Supplementary Fig 7 to demonstrate that for activated uninfected CD4+ T-cells treated with various drug concentrations, the cells within the live cell gate for flow analysis were in fact live given their low levels of propidium iodide staining (dead cell marker).

- l) Supplementary Fig 9 to show how various LRAs impact the migration of uninfected primary human CD4+ T-cells, in line with their activities on the migration of HIV infected cells in Figures 5 and 6d.**
- m) A new paragraph in the discussion describing the issue that the turnover of reactivated, latently infected cells maybe prolonged in HIV infected individuals. This issue combined with LRAs that enable migration of reactivated cells may foster reseeding of HIV-1 infection in tissue sites with ineffective HAART and should be avoided.**

In addition to these new additions that strengthen the revised manuscript, these new data also raise additional comments for further consideration as follows.

Additional Comments for revised manuscript:

- 1. Now that error bars are added to Fig 4a and Supplementary Fig 3b, the error bars in migration measurements for LTR-GFP infected Jurkat cells overlap between untreated and TNF-treated cells. Therefore despite a trend for TNF increasing migration as well as transcription, the migration increase does not appear significant when error is taken into account and this should be noted in the results (text lines 141-144).**

Thank you for this comment. We understand your concern about the attenuated effect of co-expression of the CXCR4 and the LTR promoters when the stand-error is taken into account. Since Figure 4a only describes a result of a duplicate, we created an inset representing 18 additional migration measurements to strengthen the effect when more than two measurements are accounted for. The inset contains all positive controls, 18 in total, (HIV LTR-GFP construct, +/-TNF) which we used in each single migration measurement and describes the average value of the ratio of migration of TNF treated samples divided by the migration of the untreated samples (i.e. $\text{Migration}_{+TNF}/\text{Migration}_{-TNF}$). The standard-error is plotted as well. The inset reveals the clear effect of coupling for human CXCR4 and the HIV LTR promoter. We have updated Figure 4a and explained the additional inset in the caption as follows:

“(Inset) Correlated migration increase with TNF is alternatively shown using an average ratio of TNF treated migration by untreated migration. This calculation accounts for 18 migration measurements of LTR-GFP Jurkats used as a positive control for each experiment. The standard-error is plotted as well.”

- 2. In Supplementary Fig 5, for the 24hr graph, as the error bars for the untreated and TNF treated JLat 9.2 completely overlap, the minor reduction in migration of JLat 9.2 with TNF is not convincing at this time point and the text “Reduced migration values also exist for JLat 9.2 after 24hr” should be removed or modified to include the overlapping error in the measurements.**

We appreciate your observation and therefore we removed the sentence “Reduced migration values also exist for JLat 9.2 after 24h.” from the caption of Supplementary Figure 5.

3. In Supplementary Fig 6, for the 24hr time point with TNF for JLat clones 15.4 and 9.2, the error bars for the % reactivation between migrating and non-migrating cells overlap. Despite the trend of an increase in reactivation in migrating versus non-migrating cells, this change does not look significant due to the overlapping error bars. In contrast, the change looks more convincing at 48 hours for both JLat clones as the error bars don't overlap. Therefore the text in the figure legend should be reworded to account for the error bars, or focus on the increase at 48 hours as this is most convincing.

We have reworded the following text in the caption of Supplementary Figure 6 as follows:

“Consistent increases in reactivation were observed for both JLat isoclonal cells despite having overlapping standard-error bars. The increase of reactivation is further accentuated for 48h TNF.”

4. Supplementary information for “Reactivation rate from latency of migrating cells is higher than non-migrating cells” is hard to follow. Section 1 and top graph – is a linear equation really appropriate to calculate the JLat 6.3 reactivation rate as the %ON is similar for 24 and 48 hours for this clone? Section 2: This is hard to follow and please consider rewording, perhaps adding an example with numbers, to make it easier to follow. Section 3: The error bar for JLat 15.4 migrating cells hits the non-migrating cells – doesn't this suggest that their reactivation is similar when error is taken into account?

We thank the reviewer for these details which increase the clarity of the section.

For section 1: As the linear fit had a very high R^2 for the two other JLat clones we did not find it necessary to change the functional form for the fit, even if the low reactivating JLat 6.3 does show a plateau of reactivation. In this case we would be underestimating the reactivation rate for that specific clone, but despite this the reactivation lands above the null hypothesis line.

For section 2 we have now included both of the suggested changes. We have reworded and added some details to clarify any misunderstanding of the calculation. In addition we have included an example for the full calculation of JLat 9.2 by defining and providing the values of each quantity in the equation followed by the actual calculation.

For Section 3: Indeed, of the three JLat clones 15.4 had the largest standard-error for its measurements. We will note 2 observations. The first is that the general trend shows all three JLat clones individually landing above the null-hypothesis ($y=x$) trend. The second, is that although not plotted, figure 4e in the main text shows LTIG clones A2 and A7 with similar behavior and smaller standard errors suggesting that they too are reactivating with higher rates.

These changes can be seen and are reflected in an updated version for the Supplementary Information section “Reactivation rate from latency of migrating cells is higher than non-migrating cells” to make it

more understandable to the reader. We have also provided the requested calculated example with real values for a better understanding. We have inserted the text here for convenience:

“2. (lower panel) Measured reactivation or %ON of JLAts post-48h TNF treatment was used for both migrating and non-migrating cells and compared to an expected or calculated %ON taking into account the total number of migrated cells in each experiment, the reactivation level at the start of migration, and the constant reactivation rates calculated in the previous step for the cells that migrate over a 3h migration experiment.

Formulas used for calculation:

Calculated %ON post-3h migration = %ON at start of migration + % of cells that turn on in the migrated population after 3h

Where because the measurement of reactivation for both migrating and non-migrating cells were performed after migration of cells treated with TNF for 48h, the following is used to back-calculate the initial reactivation levels for both migrating and non-migrating cells at the beginning of the migration experiment.

%ON at start of migration = (%ON for Non-migrating population treated for 48h TNF measured at the end of the 3h of migration – % of cells that turned on in the non-migrating population during the 3h experiment)

Example of expected rate calculation for JLat 9.2 :

% ON of migrating population measured in Figure 4e: 60.71%

% ON of non-migrating population measured in Figure 4e: 49.48%

Reactivation rate increase per hour (upper panel above based on Figure 4b): 1.03%

Migration rate/hr based off of total # of cells migrated post-48hrs: 27.9k/hr

*Newly reactivating cells per hour within migrating cell population: $27.9k * 1.03\%/hr = 287.38$ cells/hr*

*Calculated or expected %ON post-3h migration = $(49.48\% - (3 * 1.03\% * (300k \text{ cells} - 3 * 27.9k \text{ cells}) / 300k \text{ cells})) + (3 * 287.38) / (3 * 27.9k) = 48.28\%$*

5. Fig 6 legend and text lines 216-128. Please clarify exactly how the number of GFP+ migrating cells was measured and what is shown in Fig 6 panels b-e. Presumably migration reflects the number of cells counted in the bottom of the transwell. Does FL (a.u) reflect the GFP MFI or number of GFP+ cells in the total cell population or just the migrating cells in the bottom of the transwell? Please reword the sentence on text lines 216-218 – have you given the calculation to measure the number of GFP+ migrating cells? Finally in the methods section lines 523-528, shouldn't the number of GFP+ migrating cells be calculated by multiplying the % GFP+ cells in the bottom of the transwell by the total number of migrating cells in the bottom transwell? Using the % GFP+ in the unsorted infected CD4+ T-cells containing both migrating and non-migrating cells is less relevant if there is more reactivation in the migrating versus non-migrating subsets? Please clarify exactly what was used in the above calculations.

Thanks for this comment. We have clarified our explanation about the calculation of the # of GFP+ migrating cells by rewording line 241-244 as well as the calculation of this cell population in the method section.

Line 241-244:

“Here migration was estimated by measuring the percent of GFP+ cells that migrated post-3h and multiplying this by the total migrated cells in the bottom of the transwell migration plate to calculate the total population of LTR infected and migrated cells.”

Calculation in method section:

“Total # of GFP+ migrated cells = (% GFP+ of unsorted infected CD4+ T-cells in the bottom of the transwell) x (total # of migrating cells in the bottom transwell)”

6. Similar to point 1 above for LTR-GFP in Jurkats, the new Fig 6b data showing LTR-GFP infection of unsorted activated human CD4+ T-cells has overlapping error bars for TNF minus and plus samples for Donors 1 and 2 in both migration and GFP graphs. This indicates that these minor increases are not significant. Why are there no error bars for donor 3? These data are not convincing that there is a significant increase in migration for LTR-GFP in unsorted CD4+ T-cells with TNF, similar to Fig 4a in Jurkats (where overlapping migration error bars were also seen minus or plus TNF). The only significant change in migration and reactivation for the activated CD4+ T-cells appears to be for the sorted productive population in Fig 6c, where the error bars do not overlap. The text in lines 209-220 should be altered to reflect the error in these results as outlined above, and emphasise the results in the sorted productive cells in Fig 6c as this is where LTR reactivation and migration is most convincingly coupled.

We have changed the entire paragraph (line 233-248) about the results for HIV LTR-infected primary CD4+ T-cells (Figure 6b-c) to emphasize that GFP+ sorted cell population shows a robust increase of migration and fluorescence intensity compared to unsorted CD4+ T-cells.

“To confirm that LTR-CXCR4 promoter coupling is conserved in primary human cells, migration assays and fluorescence measurements were carried out using unsorted and sorted HIV-1 infected primary CD4+ T-cells (Fig. 6a). Despite consistent increases in their mean values, donors 1 and 2 of unsorted LTR-GFP infected CD4+ T-cells revealed no visible increase of migration and fluorescence after TNF treatment when standard-error bars (measured in duplicate) are taken into account (Fig. 6b). Challenged by the high cell counts required of transwell migration assays, the unsorted LTR-GFP infected populations provided only a duplicate of each measurement in at least 2 of 3 donors with 48h TNF treatment (Fig. 6b). Here migration was estimated by measuring the percent of GFP+ cells that migrated post-3h and multiplying this by the total migrated cells in the bottom of the transwell migration plate to calculate the total population of LTR infected and migrated cells. To prove LTR and CXCR4 co-expression, as seen in polyclonal LTR-GFP Jurkats, we performed migration and fluorescence measurements of GFP+ sorted CD4+ T-cells. GFP+ sorted populations showed robust increases in both migration and reactivation (Fig. 6c). This result is consistent with polyclonal LTR-GFP Jurkats (Figs. 6b-c and 4a).”

7. Supplementary Fig 8: For the left bottom graph (GFP in unsorted cells), is there any difference in the reactivation levels if the % of GFP+ cells is shown for the samples rather than mean GFP fluorescence (where uninfected cells dominate the calculation)? Separately, for the GFP- sorted cells in the top right panel, please show error bars if results from 2 donors are shown. Also, was GFP measured in the GFP- sorted fraction after TNF treatment like the unsorted fraction on the left bottom graph? Please include this actual data if measured, even if very low GFP levels, to show that the technique did not generate enough GFP+ cells to accurately measure differences in GFP reactivation with and without TNF.

Thank you for these comments. We have now updated Supplementary Figure 8 to show that the mean fluorescence level of GFP for the OFF sorted cell population cannot be used to measure differences of mean GFP between treated and untreated cells. Additionally, we have updated the caption of this figure and described how many donors were used and how many replicates were performed of each experiment. We have also described the % of reactivation in the GFP+ cell population to show that no difference between untreated and treated cells exists.

The mentioned points in the updated Supplementary Figure 8 caption are described as follows:

*“Mean fluorescence of the OFF, GFP- sorted cell population is uniform and no difference is viewable among untreated and TNF-treated cell populations (**lower right**). These results present a challenge in running migration experiments for non-pure latent OFF-sorted primary cell populations, typically consisting of a very small minority of latent cells selected by sorting the GFP- infected population and which is dominated by uninfected cell behavior (**upper right**). All experiments with unsorted primary CD4+ T-cells were performed as a single measurement using 2 donors and the average values are plotted with standard-error. However, experiments with OFF-sorted primary CD4+ T-cells were carried out as a single measurement using a single donor. Infection of CD4+ T-cells was quantified at ~35% GFP+ cells whereby the rate of reactivation was similar between untreated and TNF-treated cells.”*

8. Supplementary Fig 9 legend: Due to overlapping error bars between the untreated sample and TNF or ionomycin, these changes are unlikely to be significant. Therefore the conclusions that TNF increases migration and ionomycin decreases migration in this single donor are not clear and should be removed or reworded to account for the overlapping error bars.

To address this comment, we reworded the caption section describing the increase of migration under TNF-treatment. The updated sentence reads as follows:

“17 β -Estradiol (E2) shows increases in migration compared to the untreated population, PMA reduces migration alone and with Ionomycin addition.”

We did not mention a decrease for migration after the single Ionomycin treatment.

Minor comments:

9. Fig 1b versus Supplementary Fig 1: F3 has 5 SP1 sites in Fig 1 but 4 SP1 sites Supp Fig 1. Additionally, F3 has all transcription factor binding sites located a negative distance from the TSS in Fig 1b but these sites span negative and positive distances from the TSS in Supp Fig 1. Please ensure these are the same for F3 in Fig 1b and Supp Fig 1.

This is a good observation that needs clarification for the reader.

As additional binding site curation was performed (AP1 sites, CPG islands, etc.) after the initial *cis* regulatory binding site search and comparison (of NFkB, SP1, and TATA), promoters of interest were found in the literature to have a majority of these binding sites already investigated. For the case of F3, the following paper shows evidence of 5 identified SP1 sites (H = human, upper row, Oeth et al., 1997).

[1] Oeth, P., Parry, G.C. & Mackman, N. Regulation of the tissue factor gene in human monocytic cells. Role of AP-1, NF-kappa B/Rel, and Sp1 proteins in uninduced and lipopolysaccharide-induced expression. *Arteriosclerosis, thrombosis, and vascular biology* **17**, 365-74 (1997).

Supplementary figure 1 shows results from the promoter binding site search of the human genome with 4 SP1 sites. To clarify we have now added the following to descriptions of Supplementary Figure 1, in addition to citing previous research on the F3 promoter displayed in Figure 1b:

“F3 and CXCR4 promoter arrangements are compared to HIV in Figure 1b where the F3 promoter in Figure 1 has been published in an extensive promoter study in the literature (Oeth et al., 1997) and closely matches the above version from our search.”

Regarding the span of distances from the TSS: The results of Supplementary Figure 1 are displayed before any alignment by the TATA box or alternate start sites of the promoter. In this case each promoter is plotted in the location that the TSS is expected by the annotated genome, and the TATA box is a very good indication of the offset that is needed for alignment.

We have now added the following to the caption of Supplementary Figure 1:

“The binding sites displayed are the raw results from our search before any alignment between the promoters based on TATA box positioning or multiple TSSs of the genes.”

Collectively these details are important to clarify, but do not change the findings of the paper with regards to promoter similarities, co-expression data, migration measurements and observations, and the control of migration with drug treatment.

10. Fig 3, Fig 3 legend, plus text lines 128 and 138 still refer to reactivated latently infected cells dying in 1-2 days. As discussed in reviewer comments for the original manuscript, the author rebuttal and their new paragraph in the discussion, this is a complex issue. The 1-2 day estimated lifespan referenced is for productively infected cells on patients beginning ART. It is not clear how long it will take for reactivated, latently infected cells to die and may depend on the reactivation stimulus and immune cell status. Therefore it is suggested to remove the “1-2 day” time frame from Fig 3, Fig 3 legend and text lines 128 and 138. The main point is if reactivated cells can migrate to tissues with inefficient HAART penetration before they die, they can reseed infection irrespective of the time frame in which they die, which is currently unknown and could very well be longer than 1-2 days depending on the LRA.

We understand and agree with your explanation above. To address this concern, we have removed the “1-2 day” time frame from Figure 3, the legend of Figure 3, and in the main text (lines 183).

11. Text line 137 and Fig 3 legend: please provide references for the target-rich tissue sites with inefficient HAART penetration (eg. LN) that could allow migrating reactivated cells to reseed new infections.

Thank you for your observation of the need for these additional references. We found many publications about this topic (Schacker 2002 J Infect Dis 186 (8):1092-1097; Huster 2000 Am J Pathol 156 (6):1973-1986; and Tenner-Racz 1998 JEM 187 (6):949), but as we are limited in total references we chose to cite (Stellbrink 2002 AIDS 16 (11):1479-1487) in line 155 and in the legend of Figure 3 (line 813).

12. In Fig 4b legend, please mention that the untreated JLat samples were measured at 12hr for JLat clones 9.2 and 15.4 but 24hr for clone 6.3 as no 12hr time point was taken. In Supplementary Fig 4 legend, please add that no 12hr or 36hr time point was collected for JLat clone 6.3. In Supplementary

Fig 5 legend, please include that the values for the untreated JLats can be found in Supplementary Fig 4 as well as Fig 4b for the TNF-treated JLats.

We have changed the related sentences of the figure captions as follows:

Figure 4b:

“The untreated samples of the isoclones 9.2 and 15.4 were measured after 12h while the untreated sample of isoclone 6.3 was taken after 24h.”

Supplementary Figure 4:

“JLat 6.3 was not measured for 12h or 36h timepoints.”

Supplementary Figure 5:

“The values for migration of untreated JLats (12h to 48h) can be found in Supplementary Figure 4. The 12h and 24h time points of untreated JLats and all time points for TNF-treated JLats are found in Figure 4b of the main text.”

13. Please add to the discussion the possible reasons given in the author rebuttal on page 8 (Reviewer 1, question 2) as to why if Tat decreases CXCR4 expression from the cell surface and reduces the migration of HIV infected cells, why the migrating Tat expressing JLat and LTIG clones still have higher reactivation than the non-migrating cells. This is nicely explained in the rebuttal and would be helpful to readers if added to the discussion.

To emphasize why migrating and non-migrating cells show differences in reactivation levels, we have added the following to the main text to explain the heterogeneous transactivation of Tat among polyclonal populations resulting in delayed gene expression in non-migrating than migration cells (lines 299-303).

“Additionally, the consistent difference in reactivation levels of stationary and migrating cells may be also under the influence of the viral Tat protein. Tat transactivation is heterogeneous in clonal populations with single-cell transients stochastically initiating from the inactive state when reactivated, which could lead to asynchronous coupling of steady state levels^{30,31}”

14. Text lines 205-208: The text result that Cytarabine “enhanced” migration of latently infected cells is confusing as this sample had less migration than the untreated cells in Fig 5. Is it meant that these drugs still allowed migration of latently infected cells and thus may pose an issue if used to treat cancer in HIV+ patients? Please reword the text for clarity here.

To address this concern the text in lines 226-230 was changed as follows:

“Additionally, cancer treatments such as 17 β -Estradiol (E2) and Cytarabine display migration of latently infected cells (Fig. 5) and may pose a risk in HIV+ cancer patients treated with these drugs.

Custom drug strategies for HIV+ cancer patients may prove important given the overexpression of CXCR4 in more than 20 cancer types compared to non-cancerous cells⁴⁸⁻⁵¹.

15. Supplementary Table 2: Please add the bioactivities for: cytarabine (cytosine arabinoside incorporated into human DNA and kills cells), Valproic acid (HDACi), Panobinostat (also add that it is a HDACi), JQ1 (bromodomain inhibitor that reactivates HIV transcription) and 5-Aza-2-deoxycytidine (DNA methyltransferase inhibitor that can reactivate latent HIV).

We have updated Supplementary Table 2 with these additional bioactivities.

16. Results text lines 231-233: Line 231-The GFP+ infected and sorted population used in this study actually represents cells initially productively infected that are further stimulated by drugs, which is totally different to latent HIV being reactivated by drugs. Therefore please add “may” to the sentence “This GFP infected and sorted population may represent actively replicating latent cells ie. post reactivation”. For clarity on line 233, please also change “transient of reactivation to a fully activated state” to “transition of reactivation to a fully activated state”.

To address these comments, we have added the word “may” to line 307 and we also replaced the word “transient” to “transition” in line 324. Both sentences are described as follows:

Line 259 – *“This GFP+ infected and sorted population may represent actively replicating latent cells, i.e. post-reactivation.”*

Line 260 – *“Although deficient in observing the transition of reactivation to a fully activated state, the sorted infected population allows for observation of modulated migration and viral expression phenotypes.”*

17. Methods: Text line 401 - Please change “transfection” to “infection” of primary CD4+ T-cells with lentiviral supernatant. Text line 545 in the Flow analysis section - please define how the live gate was determined ie. FSC vs SSC.

As suggested, we have replaced the word “transfection” to “infection” in line 415 of the main text. To define how the live gate was determined we reworded a sentence in line 558 of the methods:

“10k cells in the live gate were collected for all measurements using FSC vs. SSC.”

In summary, the authors are to be congratulated on the additional data provided in the revised manuscript and their efforts to address all reviewer questions. The primary human CD4+ T-cell experiments in particular were difficult but have provided interesting new data despite being unable to test the impact of various LRAs on reactivation and migration of latently infected primary T-cells due to technical limitations. However, this new primary T-cell data confirms in productively infected, primary human CD4+ T-cells that HIV-1 LTR transcription and cell migration are both enhanced by

TNF in the absence of Tat, and confirm that this trend is lost when Tat-expressing, full length HIV is used for infection similar to the cell line data. They also confirm in productively infected, activated human CD4+ T-cells that certain LRAs enhance HIV expression but importantly impair cell migration. Such LRAs may be beneficial to reactivating latently infected cells in patients while minimising spread of these reactivated cells to tissue sites with inefficient HAART penetration to prevent reseeding of new infections. Given the novelty of these findings, their interest to the HIV latency field, virus-host factor field and opportunities to follow up other viral-human gene co-regulated promoters, the paper is recommended for publication provided the additional comments above are addressed.

We thank the reviewer for their support of this study along with their in-depth comments throughout the review process that have brought additional clarity and strength to the article. We hope that this revision has addressed these final comments satisfactorily.

Reviewer #2 (Remarks to the Author):

Bohn-Wippert et al have expanded their research study including new experiments using primary cells and more potent latency reversing compounds. These authors were extremely responsive to the reviewers comments and suggestions for improvement of their research study.

We thank the reviewer for their recognition of our extensive efforts to answer all of the initial reviews as comprehensively as possible, and for their comments towards improving the article.

Reviewer #3 (Remarks to the Author):

The authors have satisfactorily revised this manuscript to address my own comments and the comments of the other reviewers. This has substantially improved the manuscript, especially with regards to including some analysis of primary CD4+ T cells. A few minor details:

1. Enrichment analysis was done using DAVID, which is not updated very often. Although there was a recent update in October 2016. Can the authors confirm that all of the enrichment analysis was done using the most recent update?

The original analysis was performed October 17th, 2016, but to be on the safe side we performed the analysis again with DAVID v6.8 and saw slight changes in the following enrichment analysis to include more gene counts for various categories, although both results are very similar, and a large diversity of categories was still present. The new analysis has not changed our original conclusions or raises a need to introduce these results in the current manuscript. Below we have included both the previous and updated Functional Annotation Charts and updated Functional Annotation Clustering for side-by-side comparison.

Older version (DAVID 6.7) (from previous submission):

Selected	Category	Term	RT	Genes	Count	%	P-Value	Benjamini
[ ]	UP_SEQ_FEATURE	splice variant	RT		143	43.3	1.6E-2	8.0E-1
[x]	SP_PRR_KEYWORDS	alternative splicing	RT		143	43.3	1.6E-2	7.6E-1
[ ]	SP_PRR_KEYWORDS	phosphorylation	RT		140	42.4	1.3E-2	7.0E-1
[ ]	SP_PRR_KEYWORDS	cytosine	RT		96	28.1	2.9E-2	5.9E-1
[ ]	GOTERM_MF_FAT	ion binding	RT		96	28.1	4.8E-2	1.0E0
[ ]	GOTERM_MF_FAT	cation binding	RT		85	25.8	4.8E-2	1.0E0
[ ]	GOTERM_MF_FAT	metal ion binding	RT		83	25.2	6.7E-2	1.0E0
[x]	SP_PRR_KEYWORDS	metal-binding	RT		66	20.0	8.6E-2	7.9E-1
[ ]	GOTERM_SF_FAT	regulation of transcription	RT		61	18.5	1.9E-2	6.7E-1
[ ]	GOTERM_MF_FAT	transition metal ion binding	RT		58	17.6	7.0E-2	9.5E-1
[ ]	SP_PRR_KEYWORDS	acetylation	RT		57	17.3	2.5E-2	6.0E-1
[ ]	UP_SEQ_FEATURE	mutagenic site	RT		49	14.8	7.6E-2	6.7E-1
[ ]	SP_PRR_KEYWORDS	Transcription	RT		47	14.2	2.1E-2	5.8E-1
[ ]	GOTERM_SF_FAT	transcription	RT		47	14.2	6.1E-2	8.0E-1
[ ]	GOTERM_SF_FAT	regulation of RNA metabolic process	RT		46	13.9	1.3E-2	6.0E-1
[x]	SP_PRR_KEYWORDS	onc	RT		46	13.9	6.7E-2	7.6E-1
[ ]	GOTERM_SF_FAT	regulation of transcription, DNA-dependent	RT		45	13.6	1.4E-2	6.0E-1
[x]	SP_PRR_KEYWORDS	transcription regulation	RT		45	13.6	3.3E-2	5.9E-1
[ ]	SP_PRR_KEYWORDS	coiled coil	RT		44	13.3	4.6E-2	6.2E-1
[ ]	SP_PRR_KEYWORDS	DNA-binding	RT		42	12.7	3.4E-2	5.7E-1
[ ]	GOTERM_CC_FAT	organelle membrane	RT		28	8.5	7.2E-2	8.7E-1
[ ]	GOTERM_CC_FAT	mitochondrion	RT		25	7.6	3.6E-2	8.2E-1
[ ]	GOTERM_SF_FAT	cell cycle	RT		23	7.0	2.2E-2	7.0E-1
[ ]	GOTERM_SF_FAT	regulation of cell proliferation	RT		23	7.0	2.5E-2	7.3E-1
[ ]	GOTERM_CC_FAT	Golgi apparatus	RT		22	6.7	2.2E-2	9.1E-1
[ ]	GOTERM_SF_FAT	regulation of cell death	RT		22	6.7	5.9E-2	7.8E-1
[ ]	GOTERM_MF_FAT	identical protein binding	RT		21	6.4	7.1E-2	9.6E-1
[ ]	SP_PRR_KEYWORDS	developmental protein	RT		21	6.4	3.3E-2	6.1E-1
[ ]	INTERPRO	Zinc finger, C2H2-like	RT		21	6.4	5.2E-2	1.0E0
[x]	SMART	ZnF_C2H2	RT		21	6.4	5.4E-2	9.2E-1
[ ]	GOTERM_SF_FAT	regulation of programmed cell death	RT		21	6.4	9.1E-2	8.2E-1
[ ]	GOTERM_SF_FAT	positive regulation of molecular function	RT		20	6.1	9.1E-2	6.8E-1
[ ]	GOTERM_CC_FAT	endomembrane system	RT		20	6.1	2.6E-2	9.2E-1
[ ]	GOTERM_SF_FAT	cell death	RT		20	6.1	5.7E-2	7.8E-1
[ ]	GOTERM_SF_FAT	death	RT		20	6.1	6.0E-2	7.7E-1
[x]	INTERPRO	Zinc finger, C2H2-type	RT		20	6.1	7.5E-2	1.0E0

Updated version (DAVID 6.8):

Subset	Category	Term	RT	Genes	Count	%	P-Value	Bonferroni
[ ]	UP_KEYWORDS	Alternative splicing	RT		206	64.0	4.5E-8	1.5E-5
[ ]	UP_KEYWORDS	Polymorphism	RT		200	62.1	1.6E-2	4.9E-1
[ ]	GOTERM_MF_DIRECT	protein binding	RT		168	52.2	2.6E-3	7.1E-1
[ ]	UP_KEYWORDS	Phosphoprotein	RT		155	48.1	2.4E-4	3.9E-2
[ ]	UP_SEQ_FEATURE	splice variant	RT		143	44.4	3.3E-3	5.7E-1
[ ]	GOTERM_CC_DIRECT	nucleus	RT		105	32.6	2.7E-2	9.3E-1
[ ]	UP_KEYWORDS	Nucleus	RT		100	31.1	4.9E-3	3.4E-1
[ ]	GOTERM_CC_DIRECT	cytoplasm	RT		100	31.1	4.9E-2	8.8E-1
[ ]	UP_KEYWORDS	Cytoplasm	RT		90	28.0	1.4E-2	5.6E-1
[ ]	UP_KEYWORDS	Metal binding	RT		77	23.9	1.2E-3	1.2E-1
[ ]	UP_KEYWORDS	Acetylation	RT		67	20.8	1.6E-2	4.5E-1
[ ]	GOTERM_MF_DIRECT	metal ion binding	RT		51	15.8	2.9E-3	4.9E-1
[ ]	UP_SEQ_FEATURE	mutagenesis site	RT		49	15.2	6.4E-3	6.0E-1
[ ]	UP_KEYWORDS	Transcription	RT		47	14.6	4.6E-2	5.7E-1
[ ]	UP_KEYWORDS	Transcription regulation	RT		46	14.3	4.5E-2	5.7E-1
[ ]	UP_KEYWORDS	Zinc	RT		46	14.3	4.9E-2	5.7E-1
[ ]	UP_KEYWORDS	DNA binding	RT		44	13.7	1.5E-3	5.1E-1
[ ]	GOTERM_BP_DIRECT	transcription, DNA-templated	RT		41	12.7	1.0E-1	9.9E-1
[ ]	UP_KEYWORDS	Ubl conjugation	RT		35	10.9	5.2E-2	5.8E-1
[ ]	UP_KEYWORDS	Hydrolase	RT		34	10.6	6.2E-2	5.8E-1
[ ]	GOTERM_CC_DIRECT	intracellular	RT		32	9.9	2.8E-2	8.8E-1
[ ]	UP_KEYWORDS	Developmental protein	RT		24	7.5	1.6E-2	4.2E-1
[ ]	UP_KEYWORDS	Isopeptide bond	RT		24	7.5	8.6E-2	6.4E-1
[ ]	GOTERM_MF_DIRECT	nucleic acid binding	RT		23	7.1	8.5E-2	9.9E-1
[ ]	GOTERM_MF_DIRECT	protein homodimerization activity	RT		22	6.8	6.9E-3	7.5E-1
[ ]	INTERPRO	Zinc finger, C2H2-type/Integrase DNA-binding domain	RT		20	6.2	1.7E-2	9.8E-1
[ ]	INTERPRO	Zinc finger, C2H2-like	RT		20	6.2	3.2E-3	9.9E-1
[ ]	SMART	ZnF_C2H2	RT		20	6.2	4.1E-2	9.0E-1
[ ]	INTERPRO	Zinc finger, C2H2	RT		20	6.2	4.7E-2	9.9E-1
[ ]	GOTERM_CC_DIRECT	intracellular membrane-bounded organelle	RT		19	5.9	4.8E-3	7.5E-1
[ ]	UP_SEQ_FEATURE	zinc finger region:C2H2-type 7	RT		18	5.6	8.0E-4	5.6E-1
[ ]	UP_SEQ_FEATURE	zinc finger region:C2H2-type 6	RT		18	5.6	1.9E-3	6.1E-1
[ ]	UP_SEQ_FEATURE	zinc finger region:C2H2-type 5	RT		18	5.6	4.8E-3	6.2E-1
[ ]	UP_SEQ_FEATURE	zinc finger region:C2H2-type 3	RT		18	5.6	1.9E-2	8.5E-1
[ ]	UP_KEYWORDS	Cell cycle	RT		18	5.6	1.9E-2	4.4E-1

Older version (DAVID 6.7) (from previous submission):

Annotation Cluster 1		Enrichment Score: 1.81			Count	P-Value	Benjamini
[ ]	GOTERM_BP_FAT	positive regulation of protein kinase activity	RT		13	6.9E-4	4.5E-1
[ ]	GOTERM_BP_FAT	positive regulation of kinase activity	RT		13	9.3E-4	3.3E-1
[ ]	GOTERM_BP_FAT	positive regulation of transferase activity	RT		13	1.3E-3	3.1E-1
[ ]	GOTERM_BP_FAT	activation of protein kinase activity	RT		8	4.4E-3	5.7E-1
[ ]	GOTERM_BP_FAT	regulation of kinase activity	RT		15	5.0E-3	5.5E-1
[ ]	GOTERM_BP_FAT	regulation of transferase activity	RT		15	7.2E-3	6.5E-1
[ ]	GOTERM_BP_FAT	positive regulation of molecular function	RT		20	9.1E-3	6.8E-1
[ ]	GOTERM_BP_FAT	regulation of protein kinase activity	RT		14	9.3E-3	6.6E-1
[ ]	GOTERM_BP_FAT	regulation of phosphorylation	RT		17	9.7E-3	6.5E-1
[ ]	GOTERM_BP_FAT	positive regulation of catalytic activity	RT		18	1.2E-2	6.7E-1
[ ]	GOTERM_BP_FAT	regulation of phosphate metabolic process	RT		17	1.4E-2	6.0E-1
[ ]	GOTERM_BP_FAT	regulation of phosphorus metabolic process	RT		17	1.4E-2	6.0E-1
[ ]	GOTERM_BP_FAT	activation of MAPK activity	RT		3	7.3E-2	8.1E-1
[ ]	GOTERM_BP_FAT	MAPK cascade	RT		6	2.3E-1	9.2E-1
[ ]	GOTERM_BP_FAT	JNK cascade	RT		3	2.8E-1	9.4E-1
[ ]	GOTERM_BP_FAT	stress-activated protein kinase signaling pathway	RT		3	3.1E-1	9.5E-1
[ ]	GOTERM_BP_FAT	protein kinase cascade	RT		8	4.9E-1	9.8E-1
Annotation Cluster 2		Enrichment Score: 1.82			Count	P-Value	Benjamini
[ ]	GOTERM_BP_FAT	cell morphogenesis involved in differentiation	RT		12	4.5E-3	5.4E-1
[ ]	GOTERM_BP_FAT	axogenesis	RT		10	8.0E-3	6.5E-1
[ ]	GOTERM_BP_FAT	cell morphogenesis	RT		14	1.2E-2	7.0E-1
[ ]	GOTERM_BP_FAT	cellular component morphogenesis	RT		15	1.2E-2	6.6E-1
[ ]	GOTERM_BP_FAT	neuron differentiation	RT		16	1.2E-2	6.4E-1
[ ]	GOTERM_BP_FAT	cell morphogenesis involved in neuron differentiation	RT		10	1.3E-2	6.2E-1
[ ]	GOTERM_BP_FAT	neuron projection morphogenesis	RT		10	1.5E-2	5.9E-1
[ ]	GOTERM_BP_FAT	cell projection morphogenesis	RT		10	3.2E-2	7.6E-1
[ ]	GOTERM_BP_FAT	cell out morphogenesis	RT		10	4.1E-2	7.8E-1
[ ]	GOTERM_BP_FAT	neuron projection development	RT		10	4.1E-2	7.8E-1
[ ]	GOTERM_BP_FAT	neuron development	RT		12	4.2E-2	7.8E-1
[ ]	GOTERM_BP_FAT	axon guidance	RT		6	4.3E-2	7.9E-1
[ ]	GOTERM_BP_FAT	cell projection organization	RT		11	1.2E-1	8.5E-1
[ ]	GOTERM_BP_FAT	cell motion	RT		13	1.4E-1	8.6E-1

Annotation Cluster 3		Enrichment Score: 1.61			Count	P_Value	Benjamini
[ ]	GOTERM_BP_FAT	branching morphogenesis of a tube	RT		7	1.1E-3	3.0E-1
[ ]	GOTERM_BP_FAT	tube morphogenesis	RT		9	2.0E-3	3.9E-1
[ ]	GOTERM_BP_FAT	morphogenesis of a branching structure	RT		7	2.1E-3	3.6E-1
[ ]	GOTERM_BP_FAT	angiogenesis	RT		7	5.0E-2	7.6E-1
[ ]	GOTERM_BP_FAT	blood vessel morphogenesis	RT		8	8.5E-2	8.2E-1
[ ]	GOTERM_BP_FAT	blood vessel development	RT		8	1.5E-1	8.7E-1
[ ]	GOTERM_BP_FAT	vasculature development	RT		8	1.6E-1	8.9E-1
[ ]	SP_PIR_KEYWORDS	angiogenesis	RT		3	3.0E-1	9.4E-1
Annotation Cluster 4		Enrichment Score: 1.61			Count	P_Value	Benjamini
[ ]	UP_SEQ_FEATURE	zinc finger region:C2H2-type 7	RT		19	5.3E-4	4.2E-1
[ ]	UP_SEQ_FEATURE	domain:KRAB	RT		15	1.3E-3	4.8E-1
[ ]	UP_SEQ_FEATURE	zinc finger region:C2H2-type 6	RT		19	1.3E-3	3.6E-1
[ ]	INTERPRO	kruppel-associated box	RT		15	3.0E-3	8.4E-1
[ ]	SMART	KRAB	RT		15	3.1E-3	3.6E-1
[ ]	UP_SEQ_FEATURE	zinc finger region:C2H2-type 5	RT		19	3.7E-3	6.2E-1
[ ]	UP_SEQ_FEATURE	zinc finger region:C2H2-type 8	RT		16	4.1E-3	5.7E-1
[ ]	PIR_SUPERFAMILY	PIRSP005559:zinc finger protein ZFP-36	RT		8	9.9E-3	7.9E-1
[ ]	GOTERM_BP_FAT	regulation of RNA metabolic process	RT		46	1.3E-2	6.0E-1
[ ]	GOTERM_BP_FAT	regulation of transcription, DNA-dependent	RT		45	1.4E-2	6.0E-1
[ ]	UP_SEQ_FEATURE	zinc finger region:C2H2-type 4	RT		18	1.5E-2	8.6E-1
[ ]	UP_SEQ_FEATURE	zinc finger region:C2H2-type 3	RT		19	1.5E-2	8.3E-1
[ ]	GOTERM_BP_FAT	regulation of transcription	RT		61	1.9E-2	6.7E-1
[ ]	INTERPRO	Zinc finger, C2H2-type/intercase, DNA-binding	RT		19	1.9E-2	1.0E0
[ ]	SP_PIR_KEYWORDS	Transcription	RT		47	2.1E-2	5.8E-1
[ ]	UP_SEQ_FEATURE	zinc finger region:C2H2-type 9	RT		13	2.3E-2	8.8E-1
[ ]	UP_SEQ_FEATURE	zinc finger region:C2H2-type 10	RT		12	2.3E-2	8.7E-1
[ ]	UP_SEQ_FEATURE	zinc finger region:C2H2-type 2	RT		18	2.4E-2	8.6E-1
[ ]	SP_PIR_KEYWORDS	nucleus	RT		86	2.9E-2	5.9E-1
[ ]	SP_PIR_KEYWORDS	transcription regulation	RT		45	3.3E-2	5.9E-1
[ ]	SP_PIR_KEYWORDS	dna-binding	RT		42	3.4E-2	5.7E-1
[ ]	INTERPRO	Zinc finger, C2H2-like	RT		21	5.2E-2	1.0E0
[ ]	SMART	ZnF_C2H2	RT		21	5.4E-2	9.3E-1
[ ]	SP_PIR_KEYWORDS	zinc	RT		46	6.7E-2	7.0E-1
[ ]	INTERPRO	Zinc finger, C2H2-type	RT		20	7.5E-2	1.0E0
[ ]	GOTERM_BP_FAT	transcription	RT		47	8.1E-2	8.1E-1
[ ]	UP_SEQ_FEATURE	zinc finger region:C2H2-type 1	RT		14	1.1E-1	1.0E0
[ ]	UP_SEQ_FEATURE	zinc finger region:C2H2-type 13	RT		7	1.2E-1	1.0E0
[ ]	UP_SEQ_FEATURE	zinc finger region:C2H2-type 12	RT		8	1.2E-1	1.0E0
[ ]	GOTERM_MF_FAT	DNA binding	RT		48	1.3E-1	1.0E0

Updated version (DAVID 6.8):

Annotation Cluster 1		Enrichment Score: 1.88			Count	P_Value	Benjamini
UP_KEYWORDS	Cell division	RT	▬		13	1.4E-2	6.2E-1
GOTERM_BP_DIRECT	cell division	RT	▬		13	1.5E-2	9.8E-1
UP_KEYWORDS	Cell cycle	RT	▬		18	1.9E-2	4.4E-1
GOTERM_BP_DIRECT	mitotic nuclear division	RT	▬		10	2.4E-2	9.8E-1
UP_KEYWORDS	Mitosis	RT	▬		9	4.4E-2	5.9E-1
Annotation Cluster 2		Enrichment Score: 1.68			Count	P_Value	Benjamini
UP_SEQ_FEATURE	zinc finger region: C2H2-type 7	RT	▬		18	8.0E-4	5.8E-1
UP_KEYWORDS	Metal-binding	RT	▬		77	1.2E-3	1.2E-1
INTERPRO	Kruppel-associated box	RT	▬		16	1.8E-3	6.9E-1
UP_SEQ_FEATURE	zinc finger region: C2H2-type 6	RT	▬		18	1.9E-3	6.1E-1
UP_SEQ_FEATURE	domain: KRAB	RT	▬		14	2.5E-3	5.6E-1
GOTERM_MF_DIRECT	metal ion binding	RT	▬		51	2.9E-3	4.9E-1
SMART	KZAF	RT	▬		15	4.0E-3	5.3E-1
UP_SEQ_FEATURE	zinc finger region: C2H2-type 5	RT	▬		18	4.8E-3	6.2E-1
UP_KEYWORDS	Nucleus	RT	▬		100	4.9E-3	3.4E-1
UP_SEQ_FEATURE	zinc finger region: C2H2-type 8	RT	▬		15	6.2E-3	6.5E-1
UP_KEYWORDS	C2H2-binding	RT	▬		44	1.5E-2	5.1E-1
UP_SEQ_FEATURE	zinc finger region: C2H2-type 10	RT	▬		12	1.6E-2	8.4E-1
INTERPRO	Zinc finger C2H2-type/increase DNA-binding domain	RT	▬		20	1.7E-2	9.8E-1
UP_SEQ_FEATURE	zinc finger region: C2H2-type 3	RT	▬		18	1.9E-2	8.5E-1
UP_SEQ_FEATURE	zinc finger region: C2H2-type 4	RT	▬		17	1.9E-2	8.3E-1
UP_SEQ_FEATURE	zinc finger region: C2H2-type 2	RT	▬		17	2.7E-2	8.8E-1
GOTERM_CC_DIRECT	intracellular	RT	▬		32	2.8E-2	8.8E-1
INTERPRO	Zinc finger, C2H2-like	RT	▬		20	3.2E-2	9.9E-1
UP_SEQ_FEATURE	zinc finger region: C2H2-type 9	RT	▬		12	3.6E-2	9.1E-1
SMART	ZnF_C2H2	RT	▬		20	4.1E-2	9.0E-1
UP_KEYWORDS	Transcription regulation	RT	▬		46	4.5E-2	5.7E-1
UP_KEYWORDS	Transcription	RT	▬		47	4.6E-2	5.7E-1
INTERPRO	Zinc finger, C2H2	RT	▬		20	4.7E-2	9.9E-1
UP_KEYWORDS	Zinc	RT	▬		46	4.9E-2	5.7E-1
GOTERM_MF_DIRECT	nucleic acid binding	RT	▬		23	8.5E-2	9.9E-1
UP_SEQ_FEATURE	zinc finger region: C2H2-type 13	RT	▬		7	9.2E-2	1.0E0
UP_SEQ_FEATURE	zinc finger region: C2H2-type 12	RT	▬		8	9.8E-2	1.0E0
GOTERM_BP_DIRECT	transcription, DNA-templated	RT	▬		41	1.0E-1	9.9E-1
GOTERM_BP_DIRECT	regulation of transcription, DNA-templated	RT	▬		32	1.3E-1	1.0E0
UP_SEQ_FEATURE	zinc finger region: C2H2-type 1	RT	▬		13	1.4E-1	1.0E0
GOTERM_MF_DIRECT	DNA binding	RT	▬		34	1.5E-1	9.9E-1

Annotation Cluster 3			Enrichment Score: 1.67	Count	P_Value	Benjamini
[ ]	GOTERM_BP_DIRECT	regulation of transcription from RNA polymerase II promoter in response to hypoxia	RT	5	1.1E-3	8.4E-1
[ ]	KEGG_PATHWAY	HIF-1 signaling pathway	RT	5	8.8E-2	1.0E0
[ ]	KEGG_PATHWAY	Renal cell carcinoma	RT	4	1.0E-1	9.9E-1
Annotation Cluster 4			Enrichment Score: 1.64	Count	P_Value	Benjamini
[ ]	INTERPRO	Coaxial region factor 2/3 C-terminal type domain	RT	4	6.8E-3	8.0E-1
[ ]	UP_SEQ_FEATURE	domain:F3/B type C	RT	3	2.1E-2	8.4E-1
[ ]	INTERPRO	Galactose-binding domain-like	RT	5	4.0E-2	9.9E-1
[ ]	SMART	F3BFC	RT	3	4.5E-2	8.5E-1
Annotation Cluster 5			Enrichment Score: 1.42	Count	P_Value	Benjamini
[ ]	UP_KEYWORDS	Bromodomain	RT	4	2.0E-2	4.4E-1
[ ]	INTERPRO	Bromodomain	RT	4	3.0E-2	5.0E-1
[ ]	SMART	BRQND	RT	4	3.1E-2	9.2E-1
[ ]	INTERPRO	Bromodomain, conserved site	RT	3	6.2E-2	9.9E-1
[ ]	UP_SEQ_FEATURE	domain:Bromo	RT	3	7.2E-2	9.9E-1
Annotation Cluster 6			Enrichment Score: 1.41	Count	P_Value	Benjamini
[ ]	GOTERM_BP_DIRECT	regulation of transcription from RNA polymerase II promoter in response to hypoxia	RT	5	1.1E-3	8.4E-1
[ ]	GOTERM_BP_DIRECT	DNA damage response, signal transduction by p53 class mediator results in cell cycle arrest	RT	5	2.0E-2	9.7E-1
[ ]	GOTERM_BP_DIRECT	Notch signaling pathway	RT	6	4.3E-2	9.7E-1
[ ]	GOTERM_BP_DIRECT	regulation of signal transduction by p53 class mediator	RT	6	5.7E-2	9.8E-1
[ ]	GOTERM_BP_DIRECT	transcription-coupled nucleotide excision repair	RT	4	1.3E-1	1.0E0
[ ]	GOTERM_BP_DIRECT	stimulatory C-type lectin receptor signaling pathway	RT	3	5.3E-1	1.0E0
Annotation Cluster 7			Enrichment Score: 1.14	Count	P_Value	Benjamini
[ ]	UP_KEYWORDS	Transit peptide	RT	15	3.1E-2	5.4E-1
[ ]	UP_SEQ_FEATURE	transit peptide:Mitochondrion	RT	14	3.4E-2	9.2E-1
[ ]	GOTERM_CC_DIRECT	mitochondrial matrix	RT	11	4.3E-2	9.2E-1
[ ]	UP_KEYWORDS	Mitochondrion	RT	22	1.7E-1	8.1E-1
[ ]	GOTERM_CC_DIRECT	mitochondrion	RT	20	2.6E-1	9.8E-1

2. Regarding comment 4, reviewer 3, the authors bring up several possible explanations of why there are some genes that rank higher than CXCR4 in figure 2e. While it is outside the scope of this manuscript, some acknowledgement of these genes in the text would be helpful, including the possible explanations as to how they are regulated given their differing promoter architectures. The main text leaves this issue unaddressed and even just a sentence will add clarity for many readers as to why the other hits are not addressed.

We agree with the reviewer that some acknowledgement of the existence of these genes is needed along with possible non-genetic venues of co-regulation (or genetic with a different set of *cis* elements not searched for in the present study). We have now added the following to the main text (line 114-118):

“Genes with higher co-expression than CXCR4 or F3 to the LTR promoter exist and most likely occur through alternate modes of cis, trans, and post-transcriptional regulation, and are not co-regulated through the cis regulatory binding elements searched for determining genetic coupling in this study.”

If these minor issues are addressed I believe this manuscript should be accepted for publication.

We thank the reviewer for their ongoing support and insight throughout the review process and believe their comments have contributed to raising the scientific level of the original reported study.